# Resource allocation accounts for the large variability of rate-yield phenotypes across bacterial strains

**Valentina Baldazzi[1,2]\*, Delphine Ropers[3], Jean-Luc Gouzé[1], Tomas Gedeon[4], Hidde de Jong[3]\***

[1]Université Côte d'Azur, Inria, INRAE, CNRS, Sorbonne Université, Sophia Antipolis, France; [2]Université Côte d'Azur, INRAE, CNRS, Institut Sophia-Agrobiotech, Sophia Antipolis, France; [3]Université Grenoble Alpes, Inria, Grenoble, France; [4]Montana State University, Bozeman, United States

**Abstract** Different strains of a microorganism growing in the same environment display a wide variety of growth rates and growth yields. We developed a coarse-grained model to test the hypothesis that different resource allocation strategies, corresponding to different compositions of the proteome, can account for the observed rate-yield variability. The model predictions were verified by means of a database of hundreds of published rate-yield and uptake-secretion phenotypes of *Escherichia coli* strains grown in standard laboratory conditions. We found a very good quantitative agreement between the range of predicted and observed growth rates, growth yields, and glucose uptake and acetate secretion rates. These results support the hypothesis that resource allocation is a major explanatory factor of the observed variability of growth rates and growth yields across different bacterial strains. An interesting prediction of our model, supported by the experimental data, is that high growth rates are not necessarily accompanied by low growth yields. The resource allocation strategies enabling high-rate, high-yield growth of *E. coli* lead to a higher saturation of enzymes and ribosomes, and thus to a more efficient utilization of proteomic resources. Our model thus contributes to a fundamental understanding of the quantitative relationship between rate and yield in *E. coli* and other microorganisms. It may also be useful for the rapid screening of strains in metabolic engineering and synthetic biology.

**\*For correspondence:**
valentina.baldazzi@inria.fr (VB);
Hidde.de-Jong@inria.fr (HJ)

**Competing interest:** The authors declare that no competing interests exist.

## Editor's evaluation

This study develops a rigorous resource allocation model for *E. coli* growing under steady-state conditions. Validated by comparison with a compiled data set, the model highlights the complex nature of the relationship between metabolites, growth rate, and yield which is significantly more complex than the one-to-one-one relationship that has generally been assumed. The work will be of interest not only to investigators interested in basic questions of bacterial physiology but also to those working on applied problems in biotechnology.

## Introduction

Microbial growth consists of the conversion of nutrients from the environment into biomass. This flux of material is coupled with a flux of energy from the substrate to small energy cofactors (ATP, NADH, NADPH, etc.) driving biomass synthesis forward and releasing energy in the process (*Schaechter et al., 2006*). The growth of microorganisms has been profitably analyzed from the perspective of resource allocation, that is, the assignment of limiting cellular resources to the different biochemical

processes underlying growth (*Scott et al., 2010*; *Scott et al., 2014*; *Molenaar et al., 2009*; *Giordano et al., 2016*; *Weiße et al., 2015*; *Reimers et al., 2017*; *Bosdriesz et al., 2015*; *Towbin et al., 2017*; *Maitra and Dill, 2015*; *Dourado and Lercher, 2020*; *Metzl-Raz et al., 2017*). It is often considered that proteins, the main component of biomass, are also the bottleneck resource for growth. Proteins function as enzymes in carbon and energy metabolism and they constitute the molecular machines responsible for the synthesis of macromolecules, in particular proteins themselves. The composition of the proteome in a given growth condition can therefore be interpreted as the resource allocation strategy adopted by the cells to exploit available nutrients.

Two macroscopic criteria for characterizing microbial growth are growth rate and growth yield. The former refers to the rate of conversion of substrate into biomass, and the latter to the efficiency of the process, that is, the fraction of substrate taken up by the cells that is converted into biomass. Several empirical relations between proteome composition on the one hand, and growth rate and growth yield on the other, have been established. A linear relation between growth rate and the ribosomal protein fraction of the proteome holds over a large range of growth rates and for a variety of microbial species (*Scott et al., 2010*; *Neidhardt and Magasanik, 1960*; *Forchhammer and Lindahl, 1971*; *Bremer and Dennis, 1996*). Variants of this so-called growth law have been found for cases of reduced translation capacities (*Scott et al., 2010*) or different temperatures (*Herendeen et al., 1979*; *Mairet et al., 2021*). While the ribosomal protein fraction increases with the growth rate, the proteome fraction allocated to energy metabolism decreases (*Basan et al., 2015a*; *Schmidt et al., 2016*). Moreover, within this decreasing fraction, *Escherichia coli* and other microorganisms move resources from respiration to fermentation pathways (*Basan et al., 2015a*). Simple mathematical models have been proposed to account for the above relations in terms of the requirements of self-replication of the proteome and the relative protein costs and ATP yields of respiration and fermentation (*Scott et al., 2010*; *Molenaar et al., 2009*; *Giordano et al., 2016*; *Weiße et al., 2015*; *Bosdriesz et al., 2015*; *Dourado and Lercher, 2020*; *Mairet et al., 2021*; *Basan et al., 2015a*; *Mori et al., 2019*).

Most of these relations have been studied in experiments in which the same strain exhibits a range of growth rates in different environments, with different carbon sources. Even for a fixed environment, however, different strains of the same species may grow at very different rates and yields. For example, in a comparative study of seven *E. coli* strains, growth rates ranging from 0.61 to 0.97 hr$^{-1}$, and (carbon) growth yields between 0.52 and 0.66, were observed during aerobic growth on glucose (*Monk et al., 2016*). Since the genes encoding enzymes in central carbon and energy metabolism are largely shared across the strains (*Monk et al., 2016*), the yield differences are not due to different metabolic capacities but rather to different regulatory strategies, that is, different usages of the metabolic pathways of the cell. As another example, evolution experiments with *E. coli* have given rise to evolved strains that grow more than 40% faster, sometimes with higher growth yields, than the ancestor strain in the same environment (*LaCroix et al., 2015*). Analysis of the underlying mutations reveals that the higher rates and yields of the evolved strains are not due to new metabolic capacities, but rather to modified regulatory strategies (*LaCroix et al., 2015*; *Utrilla et al., 2016*).

Can the large variability of rate-yield phenotypes observed across different strains of the same species be explained by different resource allocation strategies, that is, different compositions of the proteome? In order to answer this question, we developed a coarse-grained resource allocation model that couples the fluxes of carbon and energy underlying microbial growth. The model was calibrated by means of existing data in the literature, without any parameter fitting, and its predictions were compared with a database of several hundreds of pairs of rates and yields of *E. coli* strains reported in the literature. The database includes wild-type strains as well as mutant strains obtained through directed mutagenesis or adaptive laboratory evolution (ALE).

We found that, in different growth conditions, the predicted variability of rate-yield phenotypes corresponds very well with the observed range of phenotypes. This also holds for the variability of substrate uptake and acetate secretion rates. Whereas in the literature, a high rate is often associated with a low yield, due to a shift of resources from respiration to fermentation, many of the *E. coli* strains in our database grow at a high rate and a high yield. The model predicts that strains with a high-rate, high-yield phenotype require resource allocation strategies that increase metabolite concentrations in order to allow for the more efficient utilization of proteomic resources, in particular enzymes in metabolism and ribosomes in protein synthesis. This prediction is confirmed by experimental data for a

high-rate, high-yield strain. A resource allocation strategy matching the observed strategy could only be found, however, when taking into account enzyme activities in addition to enzyme concentrations.

These results are interesting for both fundamental research and biotechnological applications. They show that the application of coarse-grained models can be used to predict multivariate phenotypes, without making any assumptions on optimality criteria, and reveal unexpected relations confirmed by the experimental data. The model is capable of predicting quantitative bounds on growth rates and yields within a specific environment, which can be exploited for rapidly screening performance limits of strains developed in synthetic biology and metabolic engineering.

## Results

### Coarse-grained model with coupled carbon and energy fluxes

Coarse-grained resource allocation models describe microbial growth by means of a limited number of macroreactions converting nutrients from the environment into proteins and other macromolecules. Several such models have been proposed, usually focusing on either carbon or energy fluxes (*Scott et al., 2010*; *Molenaar et al., 2009*; *Giordano et al., 2016*; *Weiße et al., 2015*; *Maitra and*

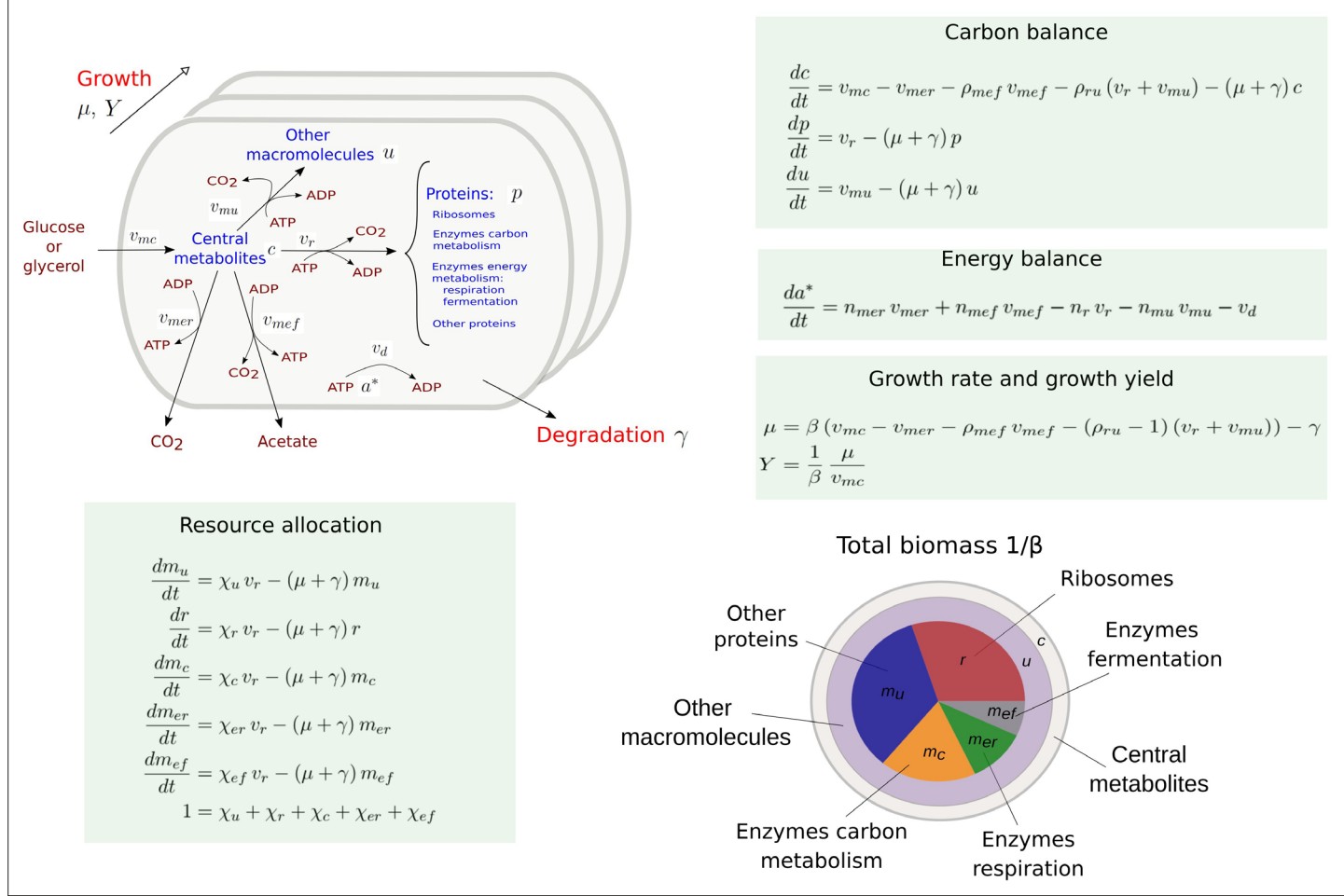

**Figure 1.** Coarse-grained model of microbial growth with coupled carbon and energy fluxes. Upper left figure: schematic outline of the model, showing the biomass constituents and the macroreactions, as well as the growth and degradation of biomass. Green boxes: system of differential equations describing the carbon and energy balances, growth rate and growth yield, and resource allocation. The kinetic expressions for the reaction rates can be found in Appendix 1. The growth rate and growth yield are defined in terms of the fluxes of the macroreactions. Lower right figure: biomass composition, including the protein categories considered in resource allocation. The fluxes $v_r$, $v_{mu}$, $v_{mc}$, $v_{mer}$, $v_{mef}$, $v_d$ [Cmmol or mmol gDW$^{-1}$ hr$^{-1}$], the variables $p$, $r$, $m_u$, $m_c$, $m_{er}$, $m_{ef}$, $c$, $u$, $a^*$ [Cmmol or mmol gDW$^{-1}$], the resource allocation parameters $\chi_u$, $\chi_r$, $\chi_c$, $\chi_{er}$, $\chi_{ef}$ [dimensionless], the degradation rate constant $\gamma$ [hr$^{-1}$], the biomass density $\beta$ [Cmmol gDW$^{-1}$], the ATP yield and cost factors $n_{mer}$, $n_{mef}$, $n_r$, $n_{mu}$ [mmol Cmmol$^{-1}$], and the correction factors for CO$_2$ loss $\rho_{mef}$, $\rho_{ru}$ [dimensionless] are formally defined in Appendix 1. The values of the parameters are derived in Appendix 2.

*Dill, 2015*; *Bosdriesz et al., 2015*; *Towbin et al., 2017*; *Mairet et al., 2021*). Few models have taken into account both, that is, the use of substrate as a carbon source for macromolecules and as a source of free energy to fuel the synthesis of macromolecules. This coupling of carbon and energy fluxes is essential, however, for understanding the relation between growth rate and growth yield. Among the notable exceptions, we cite the model of *Basan et al., 2015a* (see also *Mori et al., 2019*), which couples carbon and energy fluxes while abstracting from the reaction kinetics, and the model of *Zavřel et al., 2019*, which does provide such a kinetic view but ignores macromolecules other than proteins and focuses on photosynthetic growth (see Appendix 1 for a discussion of existing coarse-grained resource allocation models).

*Figure 1* presents a coarse-grained kinetic model that takes inspiration from and generalizes this previous work. While the model is generic, it has been instantiated for aerobic growth of *E. coli* in minimal medium with glucose or glycerol as the limiting carbon source. The model variables are intensive quantities corresponding to cellular concentrations of proteins ($p$) and other macromolecules (DNA, RNA, and lipids forming cell membranes) ($u$), as well as central carbon metabolites ($c$) and ATP ($a^*$). The central carbon metabolites notably comprise the 13 precursor metabolites from which the building blocks for macromolecules (amino acids, nucleotides, etc.) are produced (*Schaechter et al., 2006*). All concentrations have units Cmmol gDW$^{-1}$, except for ATP [mmol gDW$^{-1}$]. Five macroreactions are responsible for carbohydrate uptake and metabolism, ATP production by aerobic respiration and fermentation, and the synthesis of proteins and other macromolecules. The rates of the reactions, denoted by $v_{mc}$, $v_{mer}$, $v_{mef}$, $v_r$, and $v_{mu}$ [Cmmol gDW$^{-1}$ hr$^{-1}$], respectively, are defined by kinetic expressions involving protein, precursor metabolite, and ATP concentrations. Details of the rate equations and the derivation of the model from basic assumptions on microbial growth can be found in Appendix 1. *Appendix 1—table 1* summarizes the definition of variables, reaction rates, and parameters.

The carbon entering the cell is included in the different biomass components or released in the form of $CO_2$ and acetate. $CO_2$ is produced by respiration and macromolecular synthesis, while acetate overflow is due to aerobic fermentation (*Basan et al., 2015a*; *Gottschalk, 1986*). The carbon balance also includes the turnover of macromolecules, which is responsible for a large part of cellular maintenance costs (*van Bodegom, 2007* and Appendix 1).

The energy balance is expressed in terms of the production and consumption of ATP. While energy metabolism also involves other energy cofactors (NADP, NADPH, etc.), the latter can be converted into ATP during aerobic growth (*Basan et al., 2015a*; *Gottschalk, 1986*). We call the ATP fraction $a^*/(a^* + a)$, where $a^*$ and $a$ denote the ATP and ADP concentrations, respectively, the energy charge of the cell, by analogy with the concept of adenylate energy charge (*Atkinson, 1968*). The ATP yields of respiration and fermentation ($n_{mer}$ and $n_{mef}$) as well as the ATP costs of the synthesis of proteins and other macromolecules ($n_r$ and $n_{mu}$) are determined by the stoichiometry of the underlying metabolic pathways and the biomass composition (*Basan et al., 2015a*; *Kaleta et al., 2013* and Appendix 2). When total ATP production and consumption in growing microbial cells are computed from $n_{mer} v_{mer} + n_{mef} v_{mef}$ and $n_r v_r + n_{mu} v_{mu}$, respectively, the former usually largely exceeds the latter (*Feist et al., 2007*; *Russell and Cook, 1995*). This so-called uncoupling phenomenon is explicitly accounted for by an energy dissipation term $v_d$ in the energy balance (Appendix 1).

Like in other resource allocation models, the proteome is subdivided into categories (*Scott et al., 2010*; *Basan et al., 2015a*). We distinguish ribosomes and other translation-affiliated proteins, enzymes in central carbon metabolism, enzymes in respiration and fermentation metabolism, and a residual category of other proteins, with concentrations $r$, $m_c$, $m_{er}$, $m_{ef}$, and $m_u$, respectively. The latter category includes proteins involved in the synthesis of RNA and DNA as well as in a variety of housekeeping functions. Each category of protein catalyzes a different macroreaction in *Figure 1*: ribosomes are responsible for protein synthesis, enzymes for carbon and energy metabolism, and residual proteins for the synthesis of macromolecules other than proteins. Note that the proteins in the residual category may thus catalyze a macroreaction, contrary to what is assumed in other models in the literature (Appendix 1).

The protein synthesis capacity of the cell, given by the total protein synthesis rate $v_r$, is distributed over the protein categories using five fractional resource allocation parameters that sum to 1: $\chi_u$, $\chi_r$, $\chi_c$, $\chi_{er}$, and $\chi_{ef}$. Fixing the resource allocation parameters determines the model dynamics and therefore the growth phenotype (*Dourado and Lercher, 2020*; *Zavřel et al., 2019*; *de Groot et al., 2020*). During balanced growth, when the system is at steady state, the resource allocation

parameters equal the corresponding protein fractions, for example, $\chi_r^* = r^*/p^*$, where the asterisk ($*$) denotes the steady-state value (Appendix 1 and *Erickson et al., 2017*).

Contrary to most models of microbial growth, the biomass includes other cellular components (DNA, RNA, metabolites, etc.) in addition to proteins (Appendix 1). The growth rate $\mu$ [hr$^{-1}$] directly follows from the biomass definition, under the assumption that the total biomass concentration $1/\beta$ is constant (Appendix 1 and *de Jong et al., 2017*). The growth rate captures the specific accumulation of biomass corrected for degradation:

$$\mu = \beta \left( v_{mc} - v_{mer} - \rho_{mef} v_{mef} - (\rho_{ru} - 1)(v_r + v_{mu}) \right) - \gamma, \qquad (1)$$

where $\rho_{mef}$ and $\rho_{ru} - 1$ denote the fractional loss of carbon by fermentation and macromolecular synthesis, respectively. More precisely, $\rho_{mef}$ and $\rho_{ru}$, both greater than 1, express that $CO_2$ is a by-product of the synthesis of acetate and of proteins and other macromolecules, respectively, adding to the total flux of carbon through these macroreactions (*Basan et al., 2015a*; *Gottschalk, 1986*). In the growth rate definition of *Equation 1*, the total macromolecular synthesis rate $v_r + v_{mu}$ is multiplied with $\rho_{ru} - 1$, because only the associated $CO_2$ flux is lost to biomass production (Appendix 1).

The growth yield is defined as the ratio of the net biomass synthesis rate ($\mu/\beta$) and the substrate uptake rate $v_{mc}$:

$$Y = \frac{1}{\beta} \frac{\mu}{v_{mc}}. \qquad (2)$$

Yields are dimensionless and vary between 0 and 1. They express the fraction of carbon taken up by the cells that is included in the biomass, a definition often used in ecology and biotechnology (*Morin et al., 2016*; *Roller and Schmidt, 2015*). The definitions of *Equations 1 and 2* provide a rigorous statement of the carbon balance and thus enable the comparison of different resource allocation strategies.

The model in *Figure 1* was calibrated using data from the literature for batch or continuous growth of *E. coli* in minimal medium with glucose or glycerol. In brief, for the *E. coli* reference strain BW25113, we collected for each growth medium the growth rate and metabolite uptake and secretion rates (*Peebo et al., 2015*; *Haverkorn van Rijsewijk et al., 2011*; *Gerosa et al., 2015*), as well as protein and metabolite concentrations (*Schmidt et al., 2016*; *Gerosa et al., 2015*). Using additional assumptions based on literature data (*Bennett et al., 2009*; *Dourado et al., 2021*), we fixed a unique set of parameters for each condition (batch vs. continuous growth, glucose vs. glycerol), without parameter fitting (Appendix 2). The resulting set of quantitative models provides a concise but comprehensive representation of the growth of *E. coli* in different environments.

## Predicted rate-yield phenotypes for *E. coli*

The reference strain used for calibrating the model has, for each of the conditions considered, a specific resource allocation strategy defined by the values of the resource allocation parameters: $(\chi_u, \chi_r, \chi_c, \chi_{er}, \chi_{ef})$. We ask the question how the growth rate and growth yield change, during balanced growth, when the resource allocation strategy is different from the one adopted by the reference strain. In other words, we consider the range of possible rate-yield phenotypes for strains with the same metabolic capacities as the reference strain, but different regulation of the allocation of protein resources to the macroreactions of *Figure 1*. The same parameter values for the kinetic constants are used as for the reference strain. This allows us to focus on differences in growth rate and growth yield that can be unambiguously attributed to differences in resource allocation.

In order to predict the variability of rate-yield phenotypes, we uniformly sampled the space of possible resource allocation strategies. Except for the parameter $\chi_u$, expressing the fraction of resources attributed to housekeeping and other proteins, the parameters defining a resource allocation strategy were allowed to vary over the entire range from 0 to 1, subject to the constraint that they sum to 1 (*Figure 1*). The allowed range of values for $\chi_u$ was limited to the observed variation in the reference strain over a large variety of growth conditions (different limiting carbon sources, different stresses, etc.) (*Schmidt et al., 2016* and *Figure 2—figure supplement 1*). For every resource allocation strategy, we numerically simulated the system until a steady state was reached, corresponding to balanced growth of the culture (Materials and methods). From the steady-state values of the fluxes

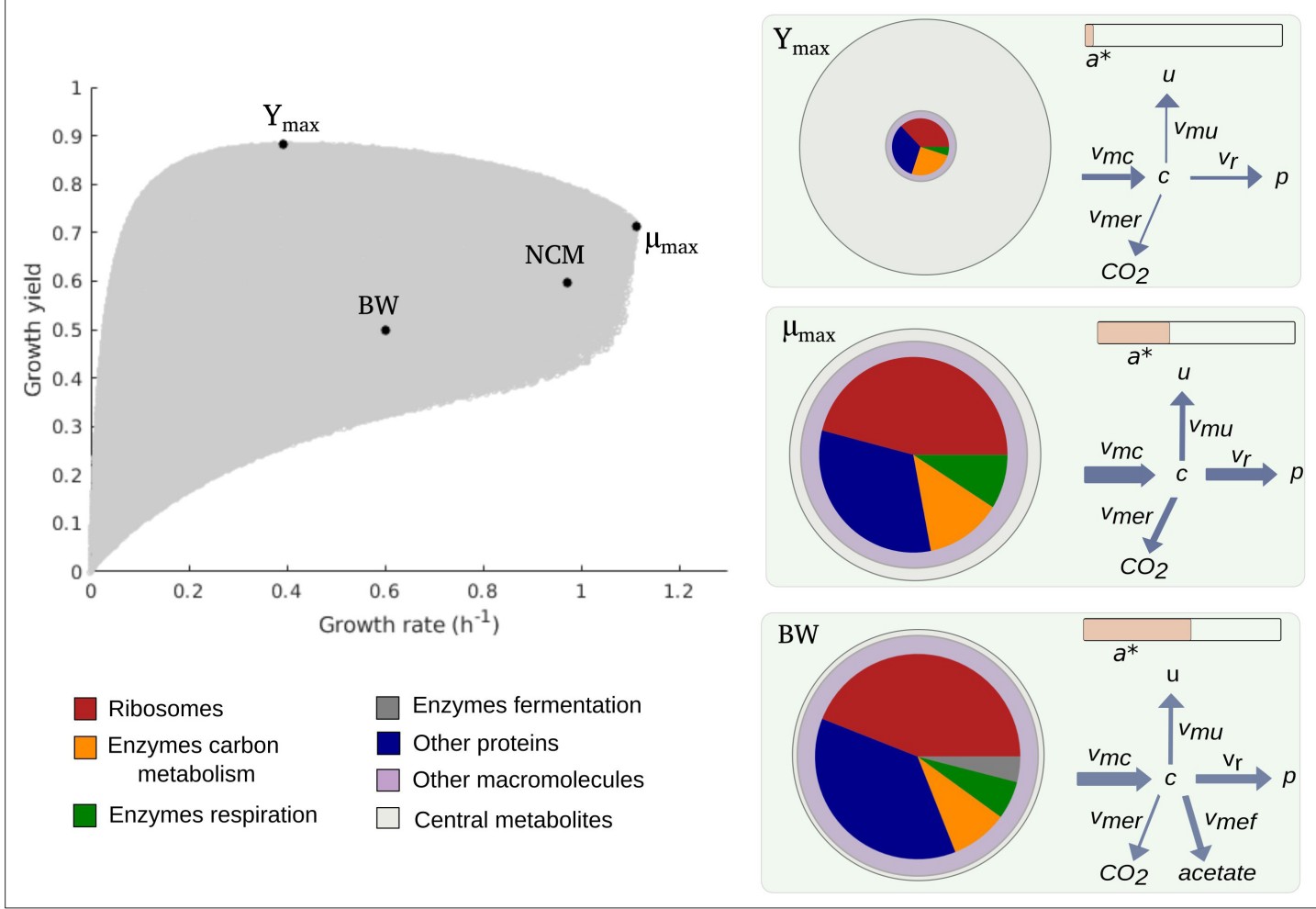

**Figure 2.** Predicted rate-yield phenotypes and underlying resource allocation strategies. Predicted rate-yield phenotypes during balanced growth of *E. coli* on minimal medium with glucose (gray dots). The resource allocation strategy and growth physiology underlying the rate-yield phenotypes are shown for selected points, corresponding to the BW25113 reference strain (BW), predicted maximum growth rate ($\mu_{max}$), and predicted maximum growth yield ($Y_{max}$). The pictograms show the biomass composition, flux distribution, and energy charge. Note that by calibration, the predicted and observed resource allocation strategies for the reference strain are identical. We also indicate, for later reference, the rate-yield phenotype of the NCM3722 strain (NCM).

The online version of this article includes the following figure supplement(s) for figure 2:

**Figure supplement 1.** Observed allocation of resources to the category of residual proteins in different growth conditions.

**Figure supplement 2.** Relation between resource allocation strategies and rate-yield phenotypes.

**Figure supplement 3.** Schematic overview of the computation of growth rate and growth yield from resource allocation strategies.

**Figure supplement 4.** Predicted fluxes, concentrations, and resource allocation along the Pareto frontier of growth rate and growth yield.

and concentrations, the growth rate and growth yield can then be computed by means of *Equations 1 and 2* (*Figure 2—figure supplement 3*).

*Figure 2* shows the cloud of predicted rate-yield phenotypes for batch growth on glucose. A first observation is that the possible combinations of rate and yield are bounded. The growth rate does not exceed 1.1 hr$^{-1}$, and for all but the lowest growth rates, the growth yield is larger than 0.3. The existence of an upper bound on the growth rate can be intuitively understood from *Equation 1*. The maximum growth rate is limited by the substrate uptake rate, which provides the carbon included in the biomass. In turn, the uptake rate is bounded by the concentration of enzymes responsible for substrate uptake and metabolism, a concentration that is ultimately limited by the total biomass concentration. The existence of a lower bound on the biomass yield is a direct consequence of the autocatalytic nature of microbial growth: the different growth-supporting functions are sustained by

enzymes and ribosomes, which need to be continually produced to counter the effect of growth dilution and degradation.

A second observation is that, for low growth rates, the maximum growth yield increases with the rate, whereas it decreases for high growth rates, above 0.4 hr$^{-1}$. The initial maximum yield increase can be attributed to the proportionally lower burden of the maintenance costs (**Pirt, 1965**). In particular, considering that a higher growth rate comes with a higher substrate uptake rate (**Equation 1**), the term $\gamma/v_{mc}$ appearing in the definition of the yield when substituting the growth rate expression (**Equation 2**) rapidly diminishes in importance when the growth rate increases (**Figure 4—figure supplement 1A**). The decrease of the maximum yield at higher growth rates reflects a trade-off that has been much investigated in microbial physiology and ecology (**Lipson, 2015**; **Beardmore et al., 2011**) and to which we return below.

Every point within the cloud of rate-yield phenotypes corresponds to a specific underlying resource allocation strategy. The mapping from resource allocation strategies to rate-yield phenotypes is far from straightforward due to the feedback loops in the model, which entail strong mutual dependencies between carbon and energy metabolism, protein synthesis, and growth. Useful insights into the nature of this mapping can be gained by visualizing the physiological consequences of a strategy in the form of a pictogram showing (i) the biomass composition, (ii) the flux map, and (iii) the energy charge. The pictogram summarizes how the incoming carbon flux is distributed over the biosynthesis, respiration, and fermentation fluxes, and how the concentrations of proteins, metabolites, and energy cofactors sustain these fluxes (**Figure 2**).

Due to model calibration, the fluxes, concentrations, and energy charge for the point corresponding to the growth of the reference strain, labeled BW in **Figure 2**, agree with the experimental data. At steady state, the resource allocation parameters coincide with the protein fractions (**Erickson et al., 2017** and Appendix 1), so that the relative sizes of the protein concentrations in the pictogram correspond to the resource allocation strategy adopted by the cells. As can be seen, the reference strain highly invests in ribosomal and other translation-oriented proteins, which take up almost 50% of the proteome. The pictogram also shows that the reference strain generates ATP by a combination of respiration and fermentation: both $v_{mer}$ and $v_{mef}$ are non-zero, and so are the corresponding enzyme concentrations $m_{er}$ and $m_{ef}$. Although proteins dominate the biomass, a non-negligible proportion of the latter consists of other macromolecules (25%) and central metabolites (1%) (Appendix 2).

How does the reference point compare with other notable points in the cloud of predicted rate-yield phenotypes, in particular the points at which the growth rate and growth yield are maximal, denoted by $\mu_{max}$ and $Y_{max}$? While the physiology of $\mu_{\max}$ is not radically different from that for the reference strain, it does have a number of distinctive features. The higher growth rate comes with a higher glucose uptake rate and a higher protein synthesis rate. The total protein concentration is lower though, due to increased growth dilution at the higher growth rate. Investment in energy metabolism has shifted from fermentation to respiration, in order to allow for more efficient ATP production at a lower enzyme concentration. The energy charge is slightly lower than in the reference strain. This is compensated for by a higher metabolite concentration, however, which leads to a higher saturation of ribosomes and allows protein synthesis to increase even at a lower ribosome concentration. In other words, bearing in mind the kinetic expression for protein synthesis from Appendix 1,

$$v_r(r, c, a^*) = k_r \, r \, \frac{c}{c + K_r} \, \frac{a^*}{a^* + K_{ar}},\tag{3}$$

where $k_r$ is a catalytic constant corresponding to the maximum protein synthesis rate and $K_r, K_{ar}$ half-saturation constants, $v_r$ can increase at $\mu_{\max}$ despite the decrease of $r$ and $a^*$, thanks to the increase of $c$.

The rate-yield phenotype corresponding to $Y_{max}$ has a predicted physiology that is strikingly different from the reference strain. The high yield is obtained by a strong reduction of protein synthesis and therefore lower concentrations of enzymes and ribosomes (**Figure 2**). Protein synthesis is the principal ATP-consuming process in microbial growth, so its reduction diminishes the need for ATP synthesis and decreases the associated loss of carbon (**Figure 1**). The net effect is a decrease of the growth rate, but an increase of the growth yield (**Equations 1 and 2**).

The strong reduction of the concentration of proteins and other macromolecules at $Y_{max}$ implies, by the assumption of constant biomass density (Appendix 1), that the metabolite concentration

increases. This may correspond to the formation of glycogen, a glucose storage compound, which occurs when excess glucose cannot be used for macromolecular synthesis due to other limiting factors. Glycogen concentrations in wild-type *E. coli* cells are low, but there exist mutants which accumulate high amounts of glycogen, on the order of 25–30% of biomass (*Morin et al., 2016*). The biomass percentage of carbohydrates and lipids in other microorganisms, such as microalgae, reaches even higher levels (*Finkel et al., 2016*; *Reitan et al., 2021*).

The upper boundary of the cloud of predicted rate-yield phenotypes in *Figure 2*, between $Y_{max}$ and $\mu_{max}$, is a Pareto frontier. It corresponds to a trade-off between growth rate and growth yield, which cannot be simultaneously increased in this region. How can this trade-off be explained? By making appropriate assumptions, the model can be simplified along the Pareto frontier, which allows the decrease in growth yield with the increase in growth rate to be traced back to changes in the resource allocation strategy (Appendix 1 and *Figure 2—figure supplement 4*). In summary, the analysis shows that an increase in growth rate requires protein synthesis to be increased, which comes with a higher loss of carbon, and therefore a lower (maximum) yield. The increase in protein synthesis leads to a higher protein concentration, reflected in a resource allocation strategy shifting resources to the synthesis of enzymes in energy metabolism and ribosomes, and a correspondingly lower concentration of central carbon metabolites. That is, on the physiological level, the trade-off between growth rate and growth yield corresponds to a trade-off between protein and metabolite concentrations.

Some caution should be exercised in the biological interpretation of the points $\mu_{max}$ and $Y_{max}$, as they are located on the upper boundary of the cloud of predicted rate-yield phenotypes. They represent extreme phenotypes that may be counterselected in the environment in which *E. coli* evolves or that may violate basic biophysical constraints not included in the model. Nevertheless, the bounds do put a quantitative limit on the variability of rate-yield phenotypes that can be confronted with the available experimental data.

## Comparison of predicted and observed rate-yield phenotypes for *E. coli*

We predicted the variability of rate-yield phenotypes of *E. coli* during batch growth in minimal medium with glucose or glycerol, and during continuous growth at different dilution rates in minimal medium with glucose. The resource allocation strategies were varied in each condition with respect to the strategy observed for the BW25113 strain used for model calibration (*Figure 3A*). In order to compare the predicted variability of rate-yield phenotypes with experimental data, we compiled a database of measured rates and yields reported in the literature (*Supplementary files 1 and 2*), and plotted the measurements in the phenotype spaces (*Figure 3B–D*). The database includes the reference wild-type strain, other *E. coli* wild-type strains, strains with mutants in regulatory genes, and strains obtained from ALE experiments. Apart from the rate and yield of the reference strain (*Haverkorn van Rijsewijk et al., 2011*), none of the data points plotted in *Figure 3* were used for calibration.

The variability of the measured rates and yields during batch growth on glucose corresponds very well with the predicted variability: all data points fall inside the predicted cloud of phenotypes and much of the cloud is covered by the data points (*Figure 3B*). Interestingly, the highest growth rates on glucose attained in ALE experiments, just above 1 hr⁻¹ (*LaCroix et al., 2015*; *Monk et al., 2017*), approach the highest predicted growth rates (1.1 hr⁻¹). The range of high growth rates is enriched in data points, which may reflect the bias that *E. coli* wild-type and mutant strains grow relatively fast on glucose and glycerol, and that in most ALE experiments the selection pressure is tilted toward growth rate.

The BW25113 strain has a low growth yield on glucose (equal to 0.50, *Haverkorn van Rijsewijk et al., 2011*). Many mutants of this strain with deletions of regulatory genes somewhat increase the yield (*Haverkorn van Rijsewijk et al., 2011*), but still fall well below the maximally predicted yield. The growth yield of some other wild-type strains is significantly higher, for example the W strain achieves a yield of 0.66 at a growth rate of 0.97 hr⁻¹ (*Monk et al., 2016*). The highest growth yield is achieved by an evolved strain (0.81, *Schuetz et al., 2012*), agreeing quite well with the maximum predicted growth yield for that growth rate. The latter strain does not secrete any acetate while growing on glucose (*Schuetz et al., 2012*), which contributes to the higher yield.

Similar observations can be made for growth of *E. coli* on glycerol, although in this case less experimental data points are available (*Figure 3D*). The model predicts that the highest growth rate on

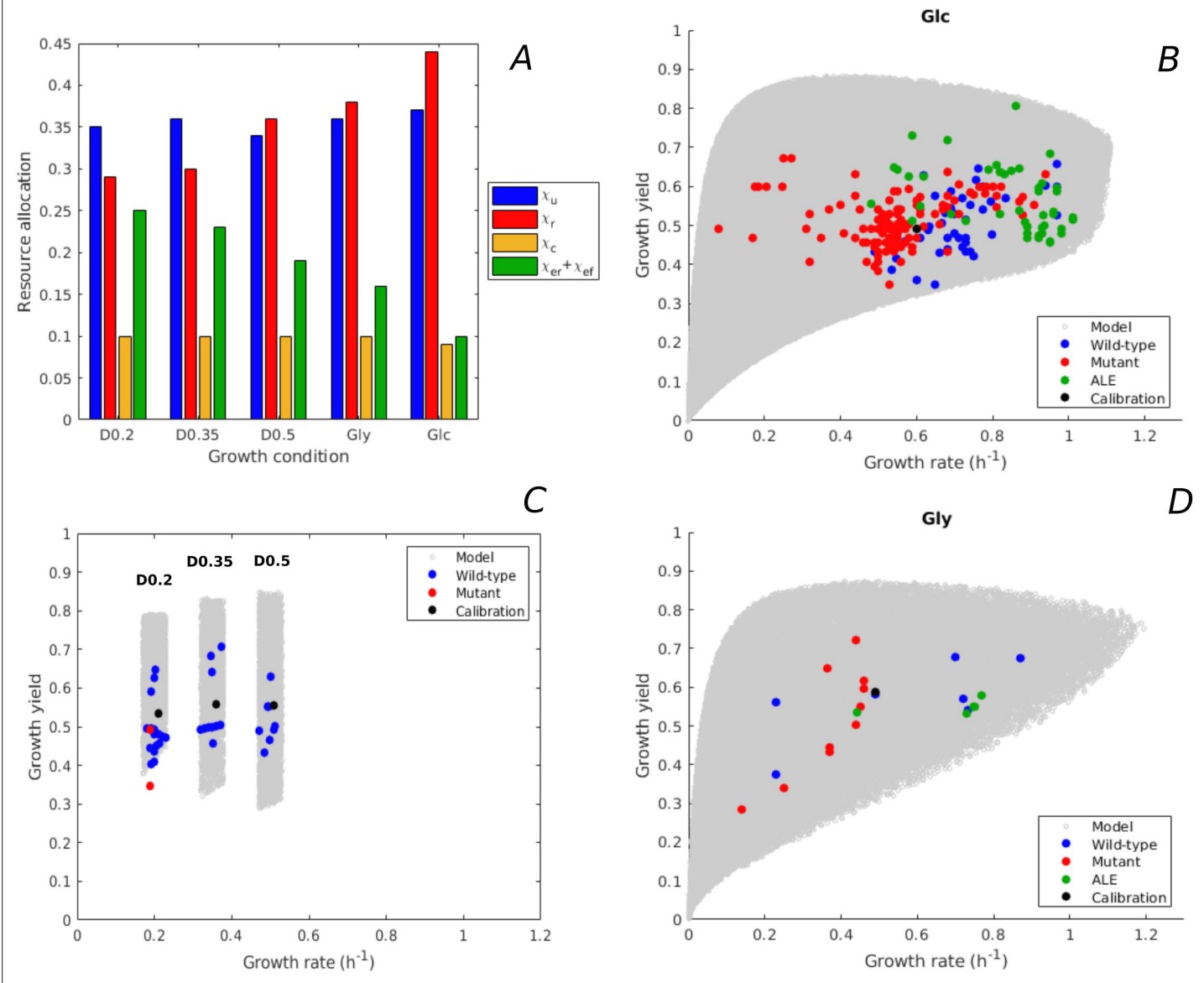

**Figure 3.** Predicted rate-yield phenotypes and comparison with experimental data. (**A**) Measured proteome fractions of the protein categories in the model, corresponding to resource allocation strategies during balanced growth, for the BW25113 reference strain used for model calibration (***Schmidt et al., 2016*** and Appendix 2). (**B**) Predicted and observed combinations of growth rate and growth yield for balanced batch growth of *E. coli* in minimal medium with glucose. The rate-yield phenotypes concern the reference strain, other wild-type strains, mutant strains obtained by directed mutagenesis, and mutant strains from adaptive laboratory evolution (ALE) experiments. (**C**) Idem for continuous growth in a chemostat in minimal medium with glucose at different dilution rates (0.2, 0.35, and 0.5 hr$^{-1}$). The predicted yields are shown for the indicated dilution rates ±10%. (**D**) Idem for batch growth of *E. coli* in minimal medium with glycerol. All predictions were made using the model in ***Figure 1***, calibrated for the different growth conditions, and varying the resource allocation parameters as described in the text (90,000–160,000 samples). The measurements of rate and yield reported in the source literature have been converted to units hr$^{-1}$ (growth rate) and a dimensionless unit corresponding to $\mathrm{Cmmol_{biomass}\ Cmmol_{substrate}^{-1}}$ (growth yield) (see Materials and methods and ***Supplementary files 1 and 2*** for details).

The online version of this article includes the following figure supplement(s) for figure 3:

**Figure supplement 1.** Robustness of rate-yield predictions for alternative model calibration and alternative model assumption.

glycerol is similar to the highest growth rate on glucose, which is confirmed by experimental data (***Andersen and von Meyenburg, 1980***). In addition to batch growth, we also considered continuous growth in a chemostat. This required a recalibration of the model, since the environment is not the same as for batch growth (Appendix 2). ***Figure 3C*** shows the predicted rate-yield phenotype space for dilution rates around 0.2, 0.35, and 0.5 hr$^{-1}$, as well as the observed rates and yields. Again, there is

good correspondence between the predicted and observed variability of growth yield. Most chemostat experiments reported in the literature have been carried out with the BW25113 and MG1655 wild-type strains. This absence of mutants and evolved strains may lead to an underestimation of the range of observed growth yields.

In the above comparisons of the model with the data, we made the assumption that the strains considered have the same metabolic capacities as the reference strain. This assumption was satisfied by restricting the database to wild-type strains with essentially the same central carbon and energy metabolism (*Monk et al., 2016*), mutant strains with deletions of genes encoding regulators instead of enzymes (*Haverkorn van Rijsewijk et al., 2011*), and short-term ALE mutants which have not had the time to develop new metabolic capacities (*Monk et al., 2017*). We also made the assumption that the parameter values are the same for all strains, so that differences in resource allocation strategies are the only explanatory variable. It is remarkable that, despite these strong assumptions, the model predicts very well the observed variability of rate-yield phenotypes in *E. coli*.

## Predicted and observed uptake-secretion phenotypes for *E. coli*

Growth rate and growth yield are defined in terms of carbon and energy fluxes through the population (*Equations 1 and 2*). Like rate and yield, some of these fluxes, in particular uptake and secretion rates, have been found to vary substantially across *E. coli* strains growing in minimal medium with glucose (*Monk et al., 2016*; *LaCroix et al., 2015*). Can our model also reproduce the observed variability of uptake-secretion phenotypes? We projected the model predictions in the space of uptake-secretion phenotypes, and crossed the latter with rate-yield phenotypes. Moreover, we compared the predicted variability with measurements from studies in which not only growth rate and growth yield, but also uptake and secretion rates were measured (*Supplementary file 1*).

*Figure 4A and B* relates the predicted range of glucose uptake rates to the growth rates and growth yields, respectively. The model predicts an overall positive correlation between growth rate and glucose uptake rate, which is an obvious consequence of the fact that glucose provides the carbon included in the biomass. The glucose uptake rate does not unambiguously determine the growth rate though. Depending on the resource allocation strategy, the bacteria can grow at different yields for a given glucose uptake rate (*Equation 2* and *Figure 4—figure supplement 1B*). Note that the trade-off between growth rate and maximum growth yield previously observed in *Figure 3* reappears here in the form of a trade-off between glucose uptake rate and maximum growth yield, for uptake rates above 20 Cmmol gDW$^{-1}$ hr$^{-1}$.

The predicted variability of glucose uptake rates vs growth rates and growth yields corresponds to the observed variability. Almost all data points fall within the predicted cloud of phenotypes and the data points cover much of the cloud. The strains resulting from ALE experiments cluster along the predicted upper bound of not only rate but also yield, suggesting that part of the increase in growth rate of ALE strains is obtained through the more efficient utilization of glucose.

Another observable flux is the acetate secretion rate, which is an indicator of the functioning of energy metabolism. In aerobic conditions, *E. coli* has two different modes of ATP production: respiration and fermentation. Glucose and glycerol are taken up by the cells and degraded in the glycolysis pathway, eventually producing acetyl-CoA. Whereas acetyl-CoA enters the tricarboxylic acid (TCA) cycle in the case of respiration, it is secreted in the form of acetate during fermentation. In both cases, NADP and other reduced compounds are produced along the way and their recycling is coupled with the generation of a proton gradient across the membrane, enabling the production of ATP. Respiration is the more efficient of the two ATP production modes: in *E. coli*, respiration yields 26 ATP molecules per molecule of glucose and fermentation only 12 (*Basan et al., 2015a*).

*Figure 4C and D* shows the predicted relation between acetate secretion rates and growth rates and growth yields. The plots reveal a clear trade-off between maximum growth yield and acetate secretion rate, due to the fact that fermentation is less efficient than respiration in producing ATP. The model predicts no apparent relation between growth rate and acetate secretion. In particular, high growth rates can be attained with a continuum of ATP production modes: from pure respiration to combinations of respiration and fermentation. Similar conclusions can be drawn when plotting the acetate secretion rate relative to the glucose uptake rate ($v_{mef}/v_{mc}$), that is, when considering the fraction of carbon taken up that is secreted as acetate (*Figure 4—figure supplement 1C–D*). Maximum yield requires respiration without fermentation, whereas minimum yield is attained for

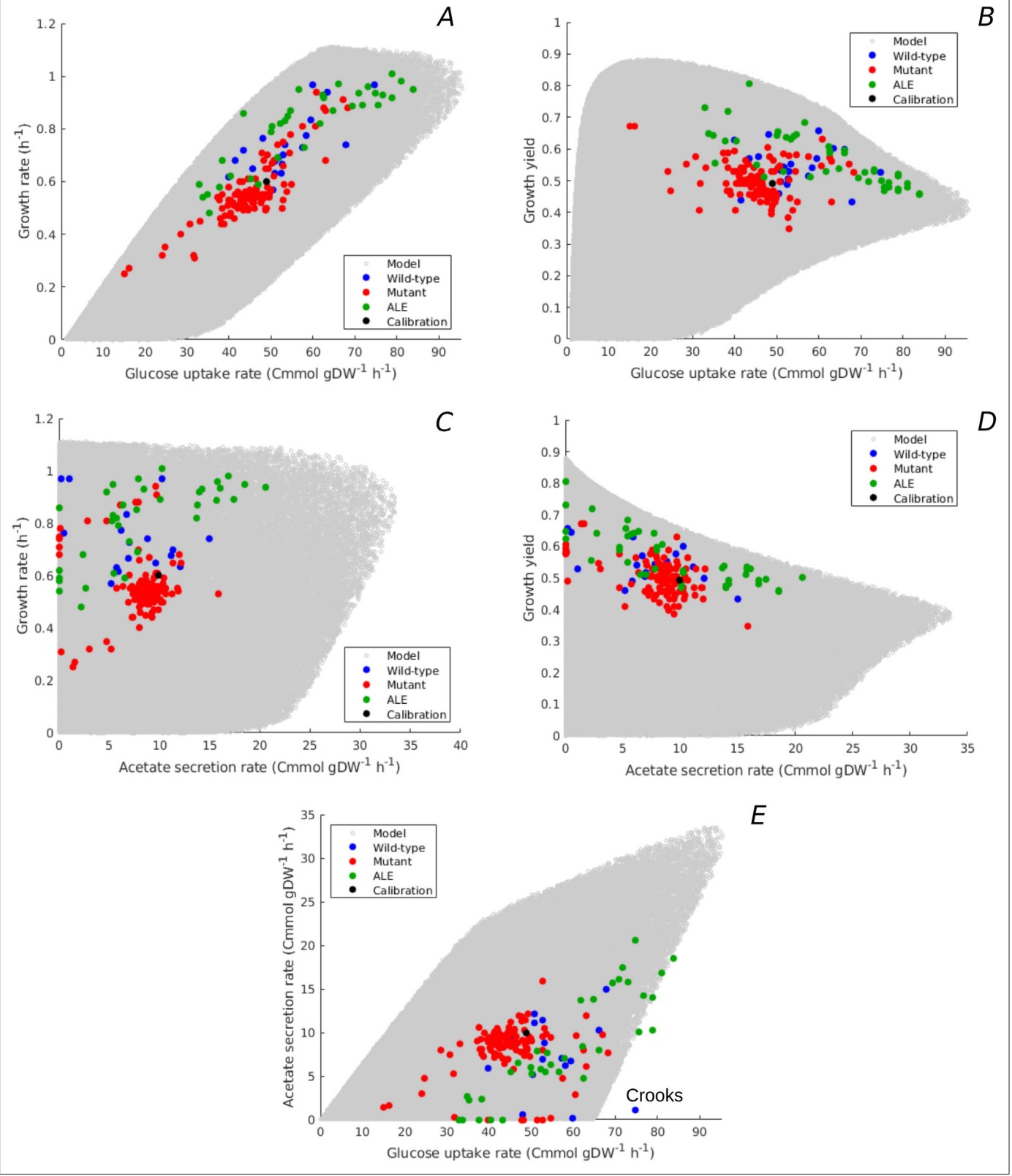

**Figure 4.** Predicted uptake-secretion phenotypes and comparison with experimental data. (**A**) Predicted and observed glucose uptake rates and growth rates for the case of batch growth of *E. coli* on minimal medium with glucose. (**B**) Idem for glucose uptake rates and growth yields. (**C**) Idem for acetate secretion rates and growth rates. (**D**) Idem for acetate secretion rates and growth yields. (**E**) Idem for glucose uptake and acetate secretion rates. The predicted uptake-secretion phenotypes $v_{mc}$ and $v_{mef}$ were taken from the simulations giving rise to *Figure 3B*. The measurements of glucose

*Figure 4 continued on next page*

*Figure 4 continued*

uptake and acetate secretion rates reported in the source literature have been converted to units Cmmol gDW$^{-1}$ hr$^{-1}$ (see Materials and methods and *Supplementary files 1 and 2* for details). The Crooks strain, labeled in panel E, shows an uptake-secretion phenotype deviating from the range of predicted phenotypes.

The online version of this article includes the following figure supplement(s) for figure 4:

**Figure supplement 1.** Additional model predictions of rate-yield and uptake-secretion phenotypes and their comparison with experimental data.

**Figure supplement 2.** Variation of normalized acetate secretion rate with growth rate in experiments with a single *E. coli* strain growing in different environments.

maximum fermentation, where more than 50% of the carbon entering the cell is lost due to acetate overflow.

The measured combinations of acetate secretion rate vs growth rate or growth yield entirely fall within the bounds predicted by the model (*Figure 4C–D*). The data notably show that as the growth yield increases, fermentation phenotypes give way to respiration phenotypes. The measurements further confirm that it is possible for *E. coli* to grow fast without acetate secretion. In particular, some of the fastest growing *E. coli* wild-type strains have no acetate overflow, like the W strain (*Monk et al., 2016*), and some of the evolved strains grow very fast but with little acetate overflow as compared to their ancestors (*Schuetz et al., 2012*). The observed relative acetate secretion rates also fall almost entirely within the predicted bounds (*Figure 4—figure supplement 1C–D*).

Another view on the uptake-secretion data is obtained when plotting, for each resource allocation strategy, the predicted glucose uptake rate against the predicted acetate secretion rate (*Figure 4E*). Not surprisingly, the maximum acetate secretion rate increases with the glucose uptake rate, since acetate is a by-product of glucose metabolism. The plot also emphasizes, however, that the increase of acetate secretion with glucose uptake is not a necessary constraint of the underlying growth physiology: *E. coli* is predicted to be able to grow without acetate overflow over almost the entire range of glucose uptake rates, from 0 to 65 Cmmol gDW$^{-1}$ hr$^{-1}$.

Again, the observed variability of uptake-secretion phenotypes falls well within the predicted bounds, although a few outliers occur. In particular, the Crooks strain has a phenotype that is significantly deviating from the predicted combinations of acetate secretion and glucose uptake rates (*Monk et al., 2017*). This suggests that resource allocation alone cannot fully explain the observed phenotype and other regulatory effects need to be taken into account in this case. High acetate secretion rates, above 20 Cmmol gDW$^{-1}$ hr$^{-1}$, are mostly absent from the database of observed uptake-secretion phenotypes. This is another manifestation of the over-representation of strains with a high growth rate on glucose (*Figure 3B*): the secretion of a large fraction of the glucose taken up in the form of acetate does not make it possible to attain high growth rates (*Equation 1*).

Given the higher ATP yield of respiration, it is not surprising that the highest growth yields are attained when respiration is preferred to fermentation. What might not have been expected, however, is that some strains achieve a growth rate on glucose close to the predicted maximum without resorting to fermentation. It is well known that when growing an *E. coli* strain in minimal medium with glucose at increasingly higher growth rates, the contribution of fermentation to ATP production increases at the expense of respiration, as witnessed by the increase of acetate secretion (*Basan et al., 2015a*; *Nanchen et al., 2006*; *Peebo et al., 2015*; *Valgepea et al., 2010* and *Figure 4—figure supplement 2*). This shift of resources from respiration to fermentation has been explained in terms of constraints on available protein resources, trading costly but efficient respiration enzymes against cheap but inefficient fermentation enzymes. The existence of strains capable of attaining the highest growth rates without fermentation suggests that this proteome constraint can be bypassed and raises the question which resource allocation strategies allow the bacteria to do so.

## Strategies enabling fast and efficient growth of *E. coli*

The analysis of the model predictions in *Figure 2*, notably the point $\mu_{max}$, provided some indications of the strategies enabling high-rate, high-yield growth of *E. coli*. Unfortunately, no data for $\mu_{max}$ are available. However, the NCM3722 strain (*Brown and Jun, 2015*) attains a growth rate approaching the maximally observed rate for *E. coli* in minimal medium with glucose (0.97 hr$^{-1}$), and has a significantly higher growth yield than the BW25113 reference strain (0.6) (*Schmidt et al., 2016*; *Cheng*

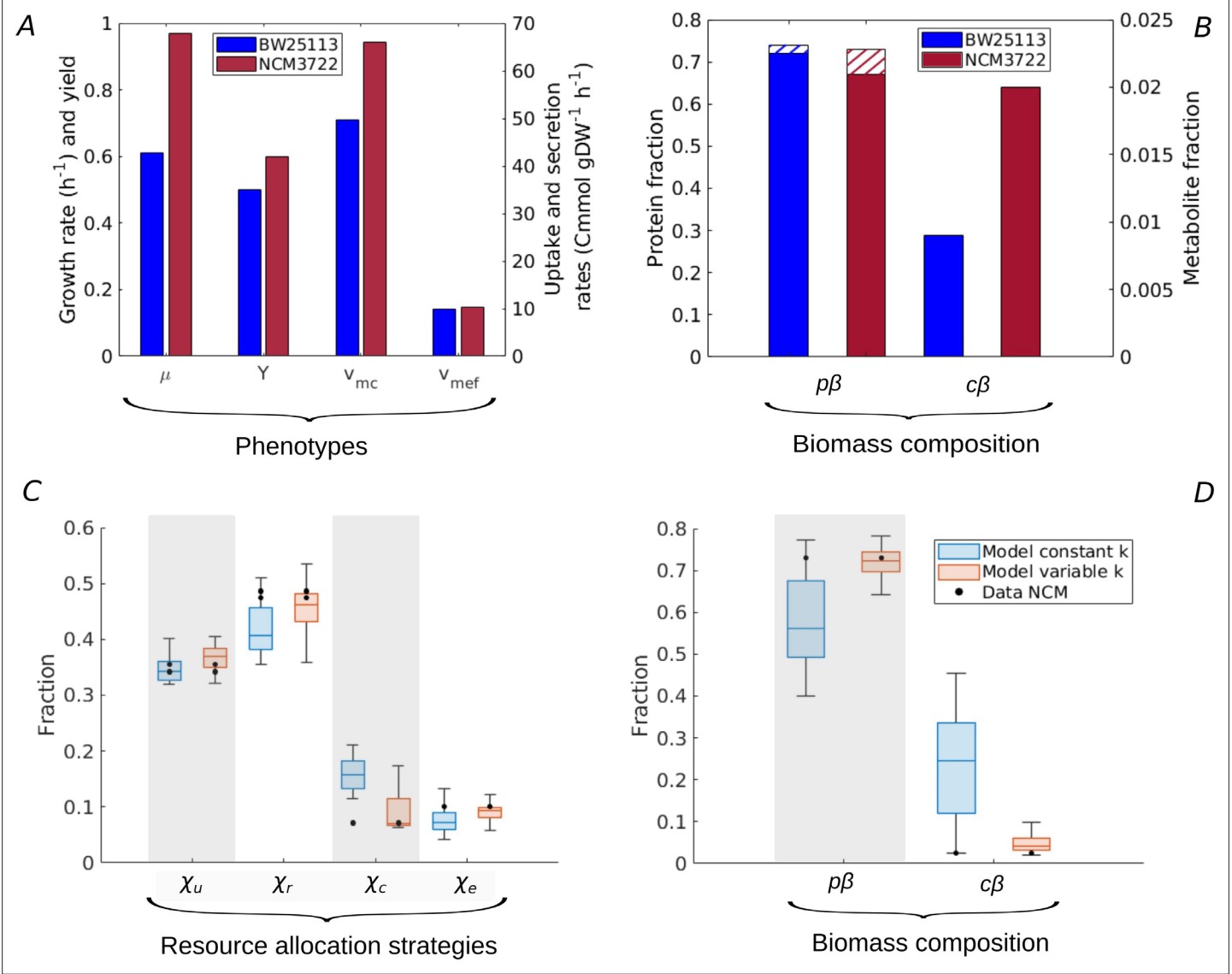

**Figure 5.** Resource allocation strategies underlying high-rate, high-yield phenotypes. (**A**) Characterization of the physiology of the NCM3722 strain in comparison with the BW25113 strain during batch growth on glucose (data from ***Appendix 2—table 1*** and ***Appendix 2—table 5***). (**B**) Comparison of total protein and metabolic fractions in NCM and BW. The total protein fraction includes amino acids (***Figure 1***), which is indicated by the hatched pattern. (**C**) Predicted resource allocation strategies for a strain with the NCM phenotype, in the case of the model with fixed catalytic constants (blue boxplot) or a model variant in which catalytic constants are allowed to vary twofold (red boxplot). The observed resource allocation strategy for NCM (***Schmidt et al., 2016***, black dots) corresponds with the strategies predicting the NCM phenotype when catalytic constants are allowed to vary, that is, when metabolic regulation in addition to resource allocation is taken into account. The model predictions summarized in the boxplot concern strategies with simulated rate-yield and uptake-secretion phenotypes within 5% of the observed values for NCM. The black dots correspond to three independent replicates of the proteomic measurements (***Schmidt et al., 2016***). (**D**) Predicted and observed biomass composition for high-rate, high-yield growth of *E. coli*, with data for NCM (***Appendix 2—table 5***). Regulation of enzyme activity leads to a very good match of predicted and observed total protein and metabolite concentrations, here indicated as fractions of the total biomass ($p\,\beta$ and $c\,\beta$).

The online version of this article includes the following figure supplement(s) for figure 5:

**Figure supplement 1.** Relative changes in kinetic parameters for resource allocation strategies reproducing the observed phenotypes of the NCM3722 strain during minimal growth on glucose.

*et al., 2019*). The glucose uptake and acetate secretion rates of NCM have been measured in the growth conditions considered here (***Basan et al., 2015a***; ***Cheng et al., 2019***) and proteomics data are available from the same experiment as used for calibration of the model (***Schmidt et al., 2016***, ***Figure 5A***). How does the observed resource allocation strategy for NCM compare with the strategies

that, according to the model, predict the rate-yield and uptake-secretion phenotypes of NCM? And how do these strategies enable fast and efficient growth of this strain?

Whereas every resource allocation strategy gives rise to a unique rate-yield phenotype, the inverse is not true: several strategies can in principle predict an observed combination of growth rate, growth yield, glucose uptake rate, and acetate secretion rate (Materials and methods and *Figure 2—figure supplement 2*). The boxplots in *Figure 5C* show the resource allocation strategies that, according to the model, give rise to a growth physiology consistent with that observed for NCM. That is, every individual strategy predicts a growth rate, growth yield, glucose uptake rate, and acetate secretion rate within 5% of the observed value. The same figure also shows the observed resource allocation strategy for NCM, consisting of the values of $\chi_u$, $\chi_r$, $\chi_c$, and $\chi_e = \chi_{er} + \chi_{ef}$ during balanced growth on glucose, derived from the proteomics data (Materials and methods).

Whereas the strategies reproducing the rate-yield and uptake-secretion phenotypes of NCM partially overlap with the measured strategy, the predicted $\chi_c$ values are significantly higher than those observed. In other words, the model requires a higher protein fraction for enzymes in central carbon metabolism ($m_c/p$) than observed in the proteomics data. The underlying problem is that in our model the carbon uptake and metabolization rate is directly proportional to the enzyme concentration (Appendix 1):

$$v_{mc} = k_{mc}\, m_c\, \frac{S}{S + K_{mc}} \approx e_m\, m_c, \qquad (4)$$

where $S \gg K_{mc}$ during balanced growth in batch and $e_m$ [hr$^{-1}$] is an apparent catalytic constant (Appendix 1). Therefore, the high value glucose uptake rate necessary for the high growth rate of NCM requires a high enzyme concentration, and therefore a high protein fraction $m_c/p$. This is contradicted by the measured protein fraction for NCM, which is slightly lower than the one observed for BW (0.07 as compared to 0.09 for BW), for a glucose uptake rate that is much higher (66.0 Cmmol gDW$^{-1}$ hr$^{-1}$ as compared to 49.6 Cmmol gDW$^{-1}$ hr$^{-1}$ for BW). Note that a less pronounced, but opposite divergence of model and data is seen in the case of the protein fractions of ribosomal proteins and enzymes in energy metabolism (*Figure 5C*). That is, the predicted over-investment in central metabolism comes with a corresponding under-investment in protein synthesis and energy metabolism.

The discrepancies between predicted and observed resource allocation strategies suggest that bacteria exploit additional regulatory factors to achieve high-rate, high-yield growth. This conclusion agrees with the view that the regulation of fluxes in central metabolism involves not only enzyme concentrations, but also regulation of enzyme activity (*Davidi and Milo, 2017*; *Donati et al., 2018*). While little is known about the mechanisms allowing NCM to grow much faster than BW, genomic changes and their physiological impact have been identified for ALE strains (*LaCroix et al., 2015*; *Utrilla et al., 2016*; *Cheng et al., 2014*). In an ALE mutant evolved in glycerol, the change in growth rate was attributed to a change in activity of the GlpK enzyme (*Cheng et al., 2014*), leading to higher glycerol uptake rates. In the model, the latter mutation would translate to an increase in the catalytic constant $k_{mc}$ (Appendix 1).

In order to verify the hypothesis that an additional layer of regulation, acting upon enzyme activity, plays a role in high-rate, high-yield growth, we modified the analysis of the model. Instead of varying only resource allocation parameters ($\chi_u, \chi_r, \chi_c, \chi_{er}, \chi_{ef}$), we also allowed the catalytic constants ($k_{mc}, k_{mer}, k_{mef}$), representing the (apparent) enzyme turnover rates in central carbon and energy metabolism (Appendix 1), to increase or decrease by at most a factor of 2. The results of the simulations are shown in *Figure 5C*. They reveal that there now exist resource allocation strategies capable of reproducing the observed NCM growth phenotypes within a 5% margin. Most notably, these strategies require an increased value of $k_{mc}$ (*Figure 5—figure supplement 1*). That is, the model predicts that glycolytic enzymes are more active in NCM as compared to BW during growth on glucose. This allows resources to be shifted from glycolytic enzymes to other growth-supporting functions. Whereas no experimental data exist to specifically test the above prediction, it is known that the activity of pyruvate kinase, regulated by fructose-1,6-bisphosphate (*Valentini et al., 2000*), increases with a higher glycolytic flux and therefore higher growth rate (*Kochanowski et al., 2013*; *Kremling et al., 2007*).

Our model thus allows the accurate reconstruction of resource allocation strategies underlying high-rate, high-yield growth of the *E. coli* NCM strain on glucose, when the repertoire of available strategies is enlarged from resource allocation to the regulation of enzyme activity. In addition to

the rate-yield and uptake-secretion phenotypes, the strategies also reproduce the total protein and metabolite concentrations (*Figure 5D* and *Basan et al., 2015b*; *Park et al., 2016*). Importantly for the question how the strategies enable high-rate, high-yield growth, NCM is seen to maintain a higher metabolite concentration than BW (*Figure 5B*). As a consequence, the estimated ratio of central metabolites and half-saturation constants rises from 1.2 for BW to 3.0 for NCM (Appendix 2). The resulting increased saturation of enzymes and ribosomes sustains higher metabolic fluxes, without an additional investment in proteins (*Figure 5B*). This observation, together with the higher activity of enzymes in central carbon metabolism, suggests that the more efficient utilization of proteomic resources is key to high-rate, high-yield growth of *E. coli*. This strategy is reminiscent of the proposed existence of a trade-off between enzyme and metabolite concentrations in central carbon metabolism in other recent studies (*Dourado et al., 2021*; *Fendt et al., 2010*; *O'Brien et al., 2016*).

## Discussion

Analysis of the resource allocation strategies adopted by microbial cells can explain a number of phenomenological relations between growth rate, growth yield, and macromolecular composition (*Scott et al., 2010*; *Scott et al., 2014*; *Molenaar et al., 2009*; *Giordano et al., 2016*; *Weiße et al., 2015*; *Reimers et al., 2017*; *Bosdriesz et al., 2015*; *Towbin et al., 2017*; *Maitra and Dill, 2015*; *Dourado and Lercher, 2020*; *Metzl-Raz et al., 2017*). We have generalized this perspective to account for a striking observation: the large variability of rate-yield phenotypes across different strains of a bacterial species grown in the same environment. We constructed a coarse-grained resource allocation model (*Figure 1*), which was calibrated using literature data on batch and continuous growth of the *E. coli* BW25113 strain in minimal medium with glucose or glycerol. In each of the conditions, we considered the rate-yield phenotypes predicted by the model when allowing resource allocation to vary over the entire range of possible strategies, while keeping the kinetic parameters constant.

This approach is based on a number of strong assumptions. The coarse-grained nature of the model reduces microbial metabolism and protein synthesis to a few macroreactions, instead of accounting for the hundreds of enzyme-catalyzed reactions involved in these processes (*Cheng et al., 2019*; *Adadi et al., 2012*; *Mori et al., 2016*; *Reimers et al., 2017*; *Wortel et al., 2018*). Resource allocation is reduced to constraints on protein synthesis capacity, whereas other constraints such as limited solvent capacity and membrane space may also play a role (*Adadi et al., 2012*; *Beg et al., 2007*; *Zhuang et al., 2011*; *Szenk et al., 2017*). All possible combinations of resource allocation parameters were considered, limited only by the constraint that they must sum to 1. Observed variations in protein abundance are less drastic (*Schmidt et al., 2016*; *Hui et al., 2015*), and coupled through shared regulatory mechanisms (*Scott et al., 2014*; *Chubukov et al., 2014*). The kinetic parameters in the model have apparent values absorbing unknown regulatory effects, specific to each growth condition. This contrasts with strain-specific kinetic models with an explicit representation of the underlying regulatory mechanisms (*Weiße et al., 2015*; *Erickson et al., 2017*; *Millard et al., 2017*), and does not allow our model as such to be used for transitions between growth conditions.

Despite these limitations, we observed a very good quantitative correspondence between the predicted and observed variability of rate-yield phenotypes of different *E. coli* strains grown in the same environment (*Figure 3*). This correspondence also holds when the comparison with the experimental data is extended to glucose uptake and acetate secretion rates associated with the measured growth rates and growth yields (*Figure 4*). The results suggest that differences in resource allocation are a major explanatory factor for the observed rate-yield variability. We verified the robustness of this conclusion by testing alternative ways to calibrate the model (Appendix 1 and Appendix 2). In particular, we used data for another commonly used laboratory strain, MG1655, to determine the kinetic parameters, and we interpreted the proteomics data differently by introducing an additional category of growth-rate-independent proteins that do not carry a flux (*Scott et al., 2010*; *Hui et al., 2015*). In both cases, the predicted rate-yield variability largely overlaps with that obtained for the reference model (*Figure 3—figure supplement 1*).

Many studies of microbial growth have provided evidence for a trade-off between growth rate and growth yield (see *Lipson, 2015*; *Beardmore et al., 2011*, for reviews). One particularly telling manifestation of this trade-off is the relative increase of acetate overflow, and thus decrease of the growth yield, when an *E. coli* strain is grown on glucose at increasingly higher growth rates, by setting the dilution rate in a chemostat or by genetically modifying the glucose uptake rate (*Figure 4—figure*

*supplement 2*). This shift of resources from respiration to fermentation has been explained in terms of a trade-off between energy efficiency and protein cost (*Molenaar et al., 2009*; *Basan et al., 2015a*; *Pfeiffer et al., 2001*). In the experimental condition considered here, batch growth on glucose of different *E. coli* strains with the same metabolic capacities, we found no straightforward relation between growth rate and growth yield. Neither the model nor the data show a correlation between growth rate and acetate overflow (*Figure 4C* and *Figure 4—figure supplement 1*), as was also previously observed by *Cheng et al., 2019*, for a selection of ALE mutant strains. In particular, the data show that some of the fastest growing strains secrete little or no acetate and therefore have a high growth yield.

These findings raise the question which resource allocation strategies allow *E. coli* to grow on glucose both rapidly and efficiently. Our model predicts that a high-rate, high-yield phenotype, as exemplified by $\mu_{max}$ in *Figure 2*, can be obtained by increasing the concentration of central carbon metabolites in comparison with the concentration observed for the BW25113 strain used for calibration. While no data are available for the $\mu_{max}$ phenotype, a higher concentration of central carbon metabolites is indeed observed for the well-characterized NCM3722 strain, which also exhibits high-rate, high-yield growth (*Figure 5B*). The increased concentration of metabolites leads to a higher saturation of enzymes and ribosomes, and allows an increase of biosynthetic fluxes without a higher investment in proteins. When comparing the resource allocation strategies that predict the NCM phenotype with experimental data (*Figure 5*), we found some discrepancies that cannot be solely attributed to the uncertainty in the proteomics data. We therefore allowed the apparent catalytic constants of the macroreactions to vary as well, contrary to the initial model assumption, in order to account for genetic differences between strains or for regulatory mechanisms responding to physiological changes. This fine-tuning of the adaptation repertoire made it possible to quantitatively reproduce the high-rate, high-yield phenotype of NCM by means of resource allocation strategies consistent with the proteomics data (*Figure 5*). In comparison with the BW reference strain, a higher value of the catalytic constant corresponding to glucose uptake and metabolism was required, that is, a higher activity of glycolytic enzymes (*Figure 5—figure supplement 1*). Both higher enzyme saturation and higher enzyme activity point at a more efficient utilization of proteomic resources as a requirement for high rate, high-yield growth.

A strategy consisting of the more efficient utilization of enzymes and ribosomes cannot be predicted by most existing models. For example, with constant metabolite concentrations and some additional simplifying assumptions, our model reduces to the well-known model of *Basan et al., 2015a*, which predicts that high growth rates can only be attained at the expense of low growth yields (Appendix 1). In other words, in the absence of the possibility of a trade-off between proteins and metabolites, our simplified model also predicts that an increase in growth rate requires a shift from energy-efficient but costly respiration to energy-inefficient but cheap fermentation. The model presented in this work is thus general enough to accommodate different strategies to increase the growth rate, some of which lead to a decrease in growth yield whereas others may afford an increase in growth yield by exploiting available degrees of freedom in the space of resource allocation strategies.

The main finding of this study is that the observed variability of growth rates and growth yields across different strains of a bacterial species can, to a large extent, be accounted for by a coarse-grained resource allocation model. The capability to predict the range of rates and yields achievable by a microbial species, and the possibility to relate these to underlying resource allocation strategies, is of great interest for a fundamental understanding of microbial growth. In addition, by extending the model with a macroreaction for the production of a protein or a metabolite of interest (*Yegorov et al., 2019*), this provides rapidly exploitable guidelines for metabolic engineering and synthetic biology, by pointing at performance limits of specific strains and suggesting improvements. While instantiated for growth of *E. coli*, the model equations are sufficiently generic to apply to other microorganisms. The calibration of such model variants can benefit from the same hierarchical procedure as developed here, exploiting largely available proteomics and metabolomics datasets.

# Materials and methods

## Simulation studies

The resource allocation models were derived from a limited number of assumptions on the processes underlying microbial growth, as explained in Appendix 1. The parameters in the models were determined from literature data, as described in Appendix 2. In order to produce the plots with rate, yield, uptake, and secretion phenotypes (*Figures 2–4*), we uniformly sampled combinations of resource allocation parameters $\chi_r$, $\chi_c$, $\chi_{er}$, and $\chi_{ef}$ such that their sum equals 1-$\chi_u$, where $\chi_u$ was sampled from a reduced interval determined from the data (*Figure 2—figure supplement 1*). Starting from initial conditions, the system was simulated for each combination of resource allocation parameters until a steady state was reached, and rate and yield were computed from the fluxes and concentrations at steady state (*Figure 2—figure supplement 3*).

When sampling the space of initial conditions for a given resource allocation strategy, the system was found to always reach the same steady state. Whereas every strategy thus gives rise to a unique rate-yield phenotype, the inverse is not true: different strategies can account for a given growth rate and growth yield. An intuitive explanation can be obtained from inspection of *Equations 1 and 2*. A given rate-yield phenotype fixes the substrate uptake rate $v_{mc}$ and the sum $v_{mer} + \rho_{mef} v_{mef} + ((\rho_{ru} - 1)(v_r + v_{mu}))$, representing the loss of carbon due to $CO_2$ outflow and acetate secretion. Different resource allocation strategies, and hence different protein and metabolite concentrations, can lead to fluxes that add up to the latter sum, and thus enable the cells to grow at the specified rate and yield (*Figure 2—figure supplement 3*). The same argument generalizes to combined rate-yield and uptake-secretion phenotypes.

All simulations were carried out by means of Matlab R2020b. The models and the simulation code used for generating all figures in the paper are available at https://gitlab.inria.fr/baldazzi/coliallocation.

## Computation of rates and yields from published experimental data

The rate-yield database was compiled from the experimental literature (*Supplementary files 1 and 2*). Growth rates have unit $hr^{-1}$ and growth yields were converted to the dimensionless quantity $Cmmol_{substrate}\ Cmmol_{biomass}^{-1}$ by means of appropriate conversion constants. Most publications report yields with unit $gDW\ mmol_{substrate}^{-1}$, that is, as the ratio of the growth rate with unit $hr^{-1}$ and the substrate uptake rate with unit $mmol_{substrate}\ gDW^{-1}\ hr^{-1}$. If yields are not explicitly reported, then they were computed in this way from the reported growth rate and substrate uptake rate. In order to convert $mmol_{substrate}$ to $Cmmol_{substrate}$, we multiplied the former with the number of carbon atoms in the substrate molecule (six for glucose, three for glycerol). In order to convert gDW to $Cmmol_{biomass}$, we used the consensus value for the biomass density $1/\beta$, 40.65 $Cmmol_{biomass}\ gDW^{-1}$ (Appendix 2). Some substrate uptake rates, in particular for the NMC3722 strain, were expressed in units $mM_{substrate}\ OD^{-1}\ hr^{-1}$. We used strain-specific and when possible laboratory-specific conversion constants from optical density (OD) to $gDW\ L^{-1}$, notably the value 0.49 $gDW\ L^{-1}\ OD^{-1}$ for NMC3722 (*Basan et al., 2015a*). Acetate secretion rates reported in $mmol_{acetate}\ gDW^{-1}\ hr^{-1}$ or $mM_{acetate}\ OD^{-1}\ hr^{-1}$ were converted to unit $Cmmol\ gDW^{-1}\ hr^{-1}$ using the same procedure.

## Computation of resource allocation strategies from proteomics data

The observed resource allocation strategies for the BW25113, MG1655, and NCM3722 strains were computed by means of the proteomics data in Table S11 of *Schmidt et al., 2016*. We computed the mass fraction for each protein category distinguished in the model by associating the latter with specific COG groups ($r/p$ amino acid transport and metabolism and translation; $m_c/p$ carbohydrate transport and metabolism; $(m_{er} + m_{ef})/p$ energy production and conversion; $m_u/p$ all other COG groups). The mass fraction of enzymes in energy metabolism was further subdivided into fractions attributed to respiration and fermentation, $m_{er}/p$ and $m_{ef}/p$, in the same way as for model calibration, by distinguishing enzymes specific to fermentation, enzymes specific to respiration, and enzymes shared between respiration and fermentation (*Basan et al., 2015a*, and *Supplementary file 4*). The resource allocation strategy during balanced growth ($\chi_u, \chi_r, \chi_c, \chi_{er}, \chi_{ef}$) was equated with the corresponding mass fractions.

## Acknowledgements

This work was supported by the ANR project Maximic (ANR-17-CE40-0024). The authors would like to thank Francis Mairet and Antrea Pavlou for comments on a previous version of the manuscript, and Achille Fraisse for help with the simulation studies.

## Additional information

### Funding

| Funder | Grant reference number | Author |
| --- | --- | --- |
| French National Research Agency | Maximic project (ANR-17-CE40-0024) | Delphine Ropers<br>Jean-Luc Gouzé<br>Hidde de Jong |

The funders had no role in study design, data collection and interpretation, or the decision to submit the work for publication.

### Author contributions

Valentina Baldazzi, Conceptualization, Data curation, Software, Validation, Investigation, Methodology, Writing - original draft, Writing - review and editing; Delphine Ropers, Conceptualization, Data curation, Writing - review and editing; Jean-Luc Gouzé, Conceptualization, Funding acquisition, Writing - review and editing; Tomas Gedeon, Conceptualization, Methodology, Writing - review and editing; Hidde de Jong, Conceptualization, Software, Funding acquisition, Validation, Investigation, Writing - original draft, Writing - review and editing

### Author ORCIDs

Valentina Baldazzi ⓘ http://orcid.org/0000-0001-9734-9759
Delphine Ropers ⓘ http://orcid.org/0000-0003-2659-3003
Hidde de Jong ⓘ http://orcid.org/0000-0002-2226-650X

### Decision letter and Author response

Decision letter https://doi.org/10.7554/eLife.79815.sa1
Author response https://doi.org/10.7554/eLife.79815.sa2

## Additional files

### Supplementary files

• Supplementary file 1. Database with reported rate-yield pairs for *Escherichia coli* grown on glucose minimal medium (excel file).

• Supplementary file 2. Database with reported rate-yield pairs for *Escherichia coli* grown on glycerol minimal medium (excel file).

• Supplementary file 3. Half-saturation constants for reactions in central carbon metabolism of *Escherichia coli* (excel file).

• Supplementary file 4. Classification of energy proteins (excel file).

• MDAR checklist

### Data availability

The current manuscript is a computational study, so no data have been generated for this manuscript. Models and simulation code are available at https://gitlab.inria.fr/baldazzi/coliallocation (copy archived at *Baldazzi and de Jong, 2023*). Literature data used for model calibration and validation are included in the manuscript as Supplementary files 1-4.

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

## Appendix 1

## Model equations

### Modeling assumptions

The coarse-grained resource allocation model of coupled carbon and energy fluxes generalizes and elaborates upon previous models of microbial growth (*Scott et al., 2010*; *Giordano et al., 2016*; *Basan et al., 2015a*; *Zavřel et al., 2019*). It is based on a partitioning of the cellular proteome into five major categories:

**Ribosomes** and translation-affiliated proteins, including enzymes in amino acid metabolism, that are necessary for protein synthesis.

**Enzymes in central carbon metabolism** that are responsible for carbohydrate uptake and metabolism, leading to central carbon metabolites that fuel biosynthesis and ATP production pathways.

**Enzymes in energy metabolism** that are responsible for transferring (free) energy from carbohydrate substrates to small energy cofactors like ATP, NADH, and NADPH. This category is further subdivided into enzymes for **aerobic respiration** and **fermentation**, respectively.

**Other proteins** that do not fall within one of the above-mentioned categories. This category includes, for example, proteins involved in the synthesis of RNA and DNA, cell-cycle proteins, and a variety of housekeeping functions.

The partitioning is different from that found in some other coarse-grained models of microbial growth, as discussed in the section Model variant with an additional growth-rate-independent protein category below.

In addition to the above proteins, we distinguish two intracellular metabolite categories:

**Central carbon metabolites**, that is, catabolic products of the carbohydrate substrate (glucose, glycerol, etc.) taken up from the medium. Central carbon metabolites include intermediates of the glycolysis pathway, the TCA cycle, and the pentose phosphate pathway, notably the 13 precursor metabolites from which the building blocks for macromolecules (amino acids, nucleotides, etc.) are produced (*Schaechter et al., 2006*). Central carbon metabolites can be stored in the form of glycogen or other storage compounds.

**Energy cofactors** driving the synthesis of proteins and other macromolecules, occurring in both their higher-energy form (ATP, NADH, NADPH, etc.) and lower-energy form (ADP, NAD+, NADP+, etc.). Here, we restrict ourselves to the principal energy cofactors ATP and ADP, exploiting the fact that in aerobic conditions NADH and NADPH can be converted to ATP (*Basan et al., 2015a*; *Gottschalk, 1986*).

In addition to proteins and metabolites, we have:

**Other macromolecules**, notably including RNA, DNA, and lipids forming the cell membrane.

The cellular biomass consists of the sum of the above categories, that is, it includes proteins, metabolites, and other macromolecules, contrary to most other models which equate biomass with proteins. For reasons of simplicity, energy cofactors are not included as a separate category in the biomass. This is motivated by the fact that the total biomass fraction of ATP, ADP, NADH, NAD+, etc. is negligible (<1%, Appendix 2). As a consequence, the model does not explicitly account for their synthesis from central carbon intermediates, but only represents their role in the flow of energy through the different macroreactions.

The following macroreactions interconverting the above biomass categories are distinguished in the model:

**Carbon uptake and central carbon metabolism**, responsible for the uptake of the carbohydrate substrate from the medium and its conversion into metabolic precursors for amino acid biosynthesis and energy metabolism.

**Energy metabolism** for the regeneration of energy cofactors (conversion of ADP into ATP) through the respiration or fermentation of central carbon intermediates. In the former case, carbon leaves the cell in the form of $CO_2$, whereas both acetate and $CO_2$ are produced in the second case.

**Protein synthesis** involving the biosynthesis and polymerization of amino acids, a process driven by ATP and releasing $CO_2$.

**Synthesis of other macromolecules**, like RNA and DNA, which consumes precursors from central metabolism and ATP, and releases $CO_2$.

The total protein synthesis rate is divided over the different protein categories enumerated above, according to fractional resource allocation parameters. Together, these parameters define the resource allocation strategy of the cell and determine the growth rate and growth yield in a given environmental condition.

The model includes two macroreactions producing ATP (respiration and fermentation) and two macroreactions consuming ATP (synthesis of proteins and other macromolecules). The ATP produced and consumed in central carbon metabolism is accounted for in the ATP balance of the other macroreactions. For example, the net ATP consumption attributed to protein synthesis does not only include the ATP costs of amino acid polymerization, but also ATP consumption and production required for amino acid synthesis (*Kaleta et al., 2013*). The same holds for the production of ATP by energy metabolism (*Basan et al., 2015a*).

Much of the carbon taken up and the ATP produced by microbial cells does not directly contribute to growth but is used for maintenance. Maintenance is a broad concept that includes, among other things, the turnover of macromolecules, osmoregulation, motility, and energy spilling (*van Bodegom, 2007*). The first type of maintenance costs distinguished in the model are the resources needed to compensate for the degradation of biomass, in particular macromolecules. As a consequence of biomass degradation, cells require a minimal substrate uptake rate above which net growth of the population starts. In Appendix 2, we show that biomass degradation in our model is structurally equivalent to the so-called maintenance coefficient in the Pirt model (*Pirt, 1965*). The second form of maintenance considered is energy dissipation. This refers to the sizable fraction of ATP that is not consumed for macromolecular synthesis but invested in other cellular processes that are not explicitly modeled, such as motility and the regulation of osmotic pressure, or that is apparently spilled (*Russell and Cook, 1995*).

## Derivation of model equations

A schematic representation of microbial growth is shown in *Appendix 1—figure 1*, illustrating the modeling assumptions discussed above. Here, we derive a mathematical model from these assumptions following a number of basic steps outlined previously (*de Jong et al., 2017*). We first define extensive variables for quantities and rates, then normalize these with respect to the mass of the growing microbial population, assuming that the biomass density is constant (*Basan et al., 2015a*). This will lead to intensive variables denoting concentrations and specific reaction rates, as well as matching expressions of growth rate and growth yield in terms of these rates.

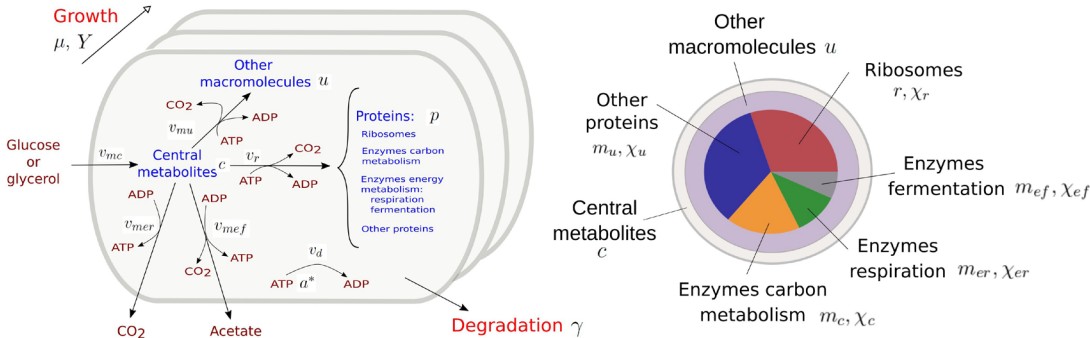

**Appendix 1—figure 1.** Resource allocation model of coupled carbon and energy fluxes in microorganisms. The figure shows the biomass categories and macroreactions, together with the concentration variables, reaction rates, and growth and degradation rates.

Carbohydrates in the medium are taken up and metabolized by the cellular population at a rate $V_{mc}$, a macroreaction that is controlled by enzymes with a total quantity equal to $M_c$. The resulting central carbon metabolites having a quantity $C$ are used to produce ATP and synthesize proteins and other macromolecules. More specifically, two alternative ATP-producing pathways are considered: respiration at a rate $V_{mer}$, catalyzed by enzymes with a quantity $M_{er}$, and fermentation at a rate $V_{mef}$, catalyzed by enzymes with a quantity $M_{ef}$. Synthesis of proteins and other macromolecules occurs at

rates $V_r$ and $V_{mu}$, respectively, and are catalyzed by ribosomes and other proteins with quantities $R$ and $M_u$, respectively. The protein and metabolite quantities are expressed in units mmol of carbon (Cmmol) and the rates in units Cmmol hr$^{-1}$.

ADP and ATP, at total quantities $A$ and $A^*$ [mmol], respectively, are permanently recycled through the ATP production and the biosynthesis pathways. $CO_2$ is released by the cell through respiration, but also as a by-product of the biosynthetic reactions and fermentation. The latter $CO_2$ outflux is accounted for in the carbon balance through the (dimensionless) correction factors $\rho_{ru}$ and $\rho_{mef}$, respectively. The correction factors express that $CO_2$ is a by-product of the synthesis of proteins and other macromolecules ($\rho_{ru}$) and acetate ($\rho_{mef}$). The loss of $CO_2$ adds to the total flux of carbon through these macroreactions, which makes $\rho_{ru} > 1$ and $\rho_{mef} > 1$. All biomass components are subjected to degradation at a rate $\gamma$ [hr$^{-1}$].

The time evolution of the total quantity of each biomass component in the growing population can now be written as follows:

$$\frac{dC}{dt} = V_{mc} - V_{mer} - \rho_{mef} V_{mef} - \rho_{ru} (V_r + V_{mu}) - \gamma C, \tag{5}$$

$$\frac{dU}{dt} = V_{mu} - \gamma U, \tag{6}$$

$$\frac{dM_u}{dt} = \chi_u V_r - \gamma M_u, \tag{7}$$

$$\frac{dR}{dt} = \chi_r V_r - \gamma R, \tag{8}$$

$$\frac{dM_c}{dt} = \chi_c V_r - \gamma M_c, \tag{9}$$

$$\frac{dM_{er}}{dt} = \chi_{er} V_r - \gamma M_{er}, \tag{10}$$

$$\frac{dM_{ef}}{dt} = \chi_{ef} V_r - \gamma M_{ef}, \tag{11}$$

where $\chi_u, \chi_r, \chi_c, \chi_{er}, \chi_{ef}$ are dimensionless resource allocation parameters, such that

$$\chi_u + \chi_r + \chi_c + \chi_{er} + \chi_{ef} = 1. \tag{12}$$

The time evolution of the total quantity of protein $P = M_u + R + M_c + M_{er} + M_{ef}$ is obtained by summing the differential equations for the different protein categories:

$$\frac{dP}{dt} = V_r - \gamma P. \tag{13}$$

We define the total cellular biomass $B$ [gDW] as

$$B = \beta (M_u + R + M_c + M_{er} + M_{ef} + C + U), \tag{14}$$

where $1/\beta$ is the biomass carbon content [Cmmol gDW$^{-1}$]. Recall that ATP and ADP are not included in the biomass.

Assuming that the volume of the growing microbial population is proportional to the biomass (**Basan et al., 2015a**), we transform the above quantities into concentrations by dividing by the total biomass $B$ : $m_u = M_u/B$, $m_c = M_c/B$, $m_{er} = M_{er}/B$, $m_{ef} = M_{ef}/B$, $r = R/B$, $c = C/B$, $u = U/B$. Accordingly, the concentration variables have units Cmmol gDW$^{-1}$ and the total biomass concentration is given by $1/\beta$.

The dynamics of the concentration variables is described by the following system of differential equations:

$$\frac{dc}{dt} = \frac{V_{mc}}{B} - \frac{V_{mer}}{B} - \rho_{mef} \frac{V_{mef}}{B} - \rho_{ru} (\frac{V_r}{B} + \frac{V_{mu}}{B}) - \gamma c - \frac{1}{B}\frac{dB}{dt} c, \tag{15}$$

$$\frac{du}{dt} = \frac{V_{mu}}{B} - \gamma\,u - \frac{1}{B}\frac{dB}{dt}\,u, \tag{16}$$

$$\frac{dm_u}{dt} = \chi_u\,\frac{V_r}{B} - \gamma\,m_u - \frac{1}{B}\frac{dB}{dt}\,m_u, \tag{17}$$

$$\frac{dr}{dt} = \chi_r\,\frac{V_r}{B} - \gamma\,r - \frac{1}{B}\frac{dB}{dt}\,r, \tag{18}$$

$$\frac{dm_c}{dt} = \chi_c\,\frac{V_r}{B} - \gamma\,m_c - \frac{1}{B}\frac{dB}{dt}\,m_c, \tag{19}$$

$$\frac{dm_{er}}{dt} = \chi_{er}\,\frac{V_r}{B} - \gamma\,m_{er} - \frac{1}{B}\frac{dB}{dt}\,m_{er}, \tag{20}$$

$$\frac{dm_{ef}}{dt} = \chi_{ef}\,\frac{V_r}{B} - \gamma\,m_{ef} - \frac{1}{B}\frac{dB}{dt}\,m_{ef}, \tag{21}$$

The (specific) growth rate $\mu$ [hr$^{-1}$] is defined as the relative biomass increase of the cell,

$$\mu = \frac{1}{B}\frac{dB}{dt}, \tag{22}$$

so that the last term in the preceding equations describes dilution by growth. Furthermore, defining $v_{mc} = V_{mc}/B$, $v_{me} = V_{me}/B$, $v_r = V_r/B$, and $v_{mu} = V_{mu}/B$ as the reaction rates per unit of biomass (volume) [Cmmol hr$^{-1}$ gDW$^{-1}$], we obtain

$$\frac{dc}{dt} = v_{mc} - v_{mer} - \rho_{mef}\,v_{mef} - \rho_{ru}\,(v_r + v_{mu}) - (\mu + \gamma)\,c, \tag{23}$$

$$\frac{du}{dt} = v_{mu} - (\mu + \gamma)\,u, \tag{24}$$

$$\frac{dm_u}{dt} = \chi_u\,v_r - (\mu + \gamma)\,m_u, \tag{25}$$

$$\frac{dr}{dt} = \chi_r\,v_r - (\mu + \gamma)\,r, \tag{26}$$

$$\frac{dm_c}{dt} = \chi_c\,v_r - (\mu + \gamma)\,m_c, \tag{27}$$

$$\frac{dm_{er}}{dt} = \chi_{er}\,v_r - (\mu + \gamma)\,m_{er}, \tag{28}$$

$$\frac{dm_{ef}}{dt} = \chi_{ef}\,v_r - (\mu + \gamma)\,m_{ef}. \tag{29}$$

In addition to the flow of carbon through the system, two equations describe energy transfer due to the production and consumption of ATP. We define, analogously to the other concentration variables, $a^* = A^*/B$ and $a = A/B$, with units mmol gDW$^{-1}$. The energy and mass flows are coupled via the following balance equations

$$\frac{da^*}{dt} = n_{mer}\,v_{mer} + n_{mef}\,v_{mef} - n_r\,v_r - n_{mu}\,v_{mu} - v_d, \tag{30}$$

$$\frac{da}{dt} = -n_{mer}\,v_{mer} - n_{mef}\,v_{mef} + n_r\,v_r + n_{mu}\,v_{mu} + v_d, \tag{31}$$

where $n_{mer}$ and $n_{mef}$ represent the ATP yield of the two ATP production pathways (with $n_{mer} > n_{mef}$, i.e. respiration has a higher yield than fermentation), and $n_{mu}$ and $n_r$ the ATP costs of biomass and protein synthesis, respectively. The reaction rate $v_d$ accounts for energy dissipation, that is, the fact

that around half of the ATP produced is not utilized for macromolecular synthesis but dissipated in other cellular processes (**Russell and Cook, 1995**; **Feist et al., 2007**).

Since $da^*/dt = -da/dt$, the total concentration of the energy cofactors (pool of $a$ and $a^*$) is equal to some constant $a_0$ [mmol gDW$^{-1}$],

$$a_0 = a + a^*, \tag{32}$$

in agreement with experiments in which usually little variation in the concentration of energy cofactors is observed (**Petersen and Møller, 2000**; **Schneider and Gourse, 2004**). Given the dependency between $a^*$ and $a$, we omit the differential equation of the latter.

The model variables and rates are summarized in **Appendix 1—table 1**.

**Appendix 1—table 1.** Model variables and rates.
The units Cmmol and gDW refer to mmol carbon and gram dry weight, respectively.

| Model | Description | Unit |
|---|---|---|
| | Macromolecule concentrations | |
| $p$ | Total proteins | Cmmol gDW$^{-1}$ |
| $r$ | Ribosomes | Cmmol gDW$^{-1}$ |
| $m_c$ | Enzymes in central carbon metabolism | Cmmol gDW$^{-1}$ |
| $m_{er}$ | Enzymes in energy metabolism (respiration) | Cmmol gDW$^{-1}$ |
| $m_{ef}$ | Enzymes in energy metabolism (fermentation) | Cmmol gDW$^{-1}$ |
| $m_u$ | Other proteins | Cmmol gDW$^{-1}$ |
| $u$ | Other macromolecules | Cmmol gDW$^{-1}$ |
| | Metabolite concentrations | |
| $c$ | Central carbon metabolites | Cmmol gDW$^{-1}$ |
| $a$ | ADP | mmol gDW$^{-1}$ |
| $a^*$ | ATP | mmol gDW$^{-1}$ |
| | Reaction rates | |
| $v_{mc}$ | Carbon uptake and central metabolism | Cmmol gDW$^{-1}$ hr$^{-1}$ |
| $v_{mer}$ | Energy metabolism (respiration) | Cmmol gDW$^{-1}$ hr$^{-1}$ |
| $v_{mef}$ | Energy metabolism (fermentation) | Cmmol gDW$^{-1}$ hr$^{-1}$ |
| $v_r$ | Protein synthesis | Cmmol gDW$^{-1}$ hr$^{-1}$ |
| $v_{mu}$ | Synthesis of other macromolecules | Cmmol gDW$^{-1}$ hr$^{-1}$ |
| $v_d$ | Energy dissipation | mmol gDW$^{-1}$ hr$^{-1}$ |
| | Other rates and yield | |
| $\mu$ | Growth rate | hr$^{-1}$ |
| $\gamma$ | Degradation rate | hr$^{-1}$ |
| $Y$ | Growth yield | – |

**Appendix 1—table 2.** Model parameters.

| Model | Description | Unit |
|---|---|---|
| | Resource allocation parameters | |
| $\chi_r$ | Fraction of ribosomal proteins | – |
| $\chi_c$ | Fraction of enzymes in central carbon metabolism | – |
| $\chi_{er}$ | Fraction of enzymes in respiratory energy metabolism | – |

*Appendix 1—table 2 Continued on next page*

*Appendix 1—table 2 Continued*

| Model | Description | Unit |
|---|---|---|
| $\chi_{ef}$ | Fraction of enzymes in fermentation energy metabolism | – |
| $\chi_u$ | Fraction of other proteins | – |
| | ATP factors | |
| $n_{mer}$ | ATP yield from respiration | mmol Cmmol$^{-1}$ |
| $n_{mef}$ | ATP yield from fermentation | mmol Cmmol$^{-1}$ |
| $n_r$ | ATP cost of protein synthesis | mmol Cmmol$^{-1}$ |
| $n_{mu}$ | ATP cost of synthesis of other macromolecules | mmol Cmmol$^{-1}$ |
| | Correction factors | |
| $\rho_{mef}$ | Correction for $CO_2$ loss during fermentation | – |
| $\rho_{ru}$ | Correction for $CO_2$ loss during biosynthesis | – |
| $1/\beta$ | Total biomass concentration | Cmmol gDW$^{-1}$ |

Using the definition of total biomass (*Equation 14*), we can express the growth rate $\mu$ as a function of the reaction rates as follows:

$$\begin{aligned} \mu &= \frac{1}{B}\frac{dB}{dt} = \beta \frac{1}{B}\frac{d(M_u + R + M_c + M_{er} + M_{ef} + C + U)}{dt} \\ &= \beta\,(v_{mc} - v_{mer} - \rho_{mef}\,v_{mef} - (\rho_{ru} - 1)\,(v_r + v_{mu})) - \gamma. \end{aligned} \tag{33}$$

Note that the total macromolecular synthesis rate is multiplied by $\rho_{ru} - 1$ rather than $\rho_{ru}$, expressing that only the additional $CO_2$ outflux is lost to biomass synthesis.

The nondimensional growth yield is defined as the ratio between the net biomass synthesis rate ($\mu/\beta$) and the carbon uptake rate $v_{mc}$, which leads to the following expression:

$$Y = \frac{1}{\beta}\frac{\mu}{v_{mc}} = \frac{v_{mc} - v_{mer} - \rho_{mef}\,v_{mef} - (\rho_{ru} - 1)\,(v_r + v_{mu}) - \gamma/\beta}{v_{mc}}. \tag{34}$$

We use Michaelis-Menten kinetics to define the rates of the macroreactions:

$$v_{mc}(m_c, S) = m_c\,k_{mc}\,\frac{S}{S + K_{mc}}, \tag{35}$$

$$v_r(r, c, a^*) = r\,f_r(c, a^*) = r\,k_r\,\frac{c}{c + K_r}\,\frac{a^*}{a^* + K_{ar}}, \tag{36}$$

$$v_{mu}(m_u, c, a^*) = m_u\,f_{mu}(c, a^*) = m_u\,k_{mu}\,\frac{c}{c + K_{mu}}\,\frac{a^*}{a^* + K_{amu}}, \tag{37}$$

$$v_{mer}(m_{er}, c, a) = m_{er}\,f_{mer}(c, a) = m_{er}\,k_{mer}\,\frac{c}{c + K_{mer}}\,\frac{a}{a + K_{amer}}, \tag{38}$$

$$v_{mef}(m_{ef}, c, a) = m_{ef}\,f_{mef}(c, a) = m_{ef}\,k_{mef}\,\frac{c}{c + K_{mef}}\,\frac{a}{a + K_{amef}}, \tag{39}$$

where $S$ denotes the concentration of the substrate in the medium [Cmmol L$^{-1}$], $K_{mc}$, $K_r$, $K_{ar}$, $K_{mu}$, $K_{amu}$, $K_{mer}$, $K_{amer}$, $K_{mef}$, $K_{amef}$ half-saturation constants [Cmmol gDW$^{-1}$] and [mmol gDW$^{-1}$], and $k_{mc}$, $k_r$, $k_{mu}$, $k_{mer}$, $k_{mef}$ maximum catalytic rate constants [hr$^{-1}$]. As can be seen, rates are proportional to enzyme concentrations, but depend nonlinearly on metabolite concentrations. During balanced growth in batch, the external substrate concentration $S$ is much higher than the half-saturation constant $K_{mc}$ ($S \gg K_{mc}$), so that *Equation 35* can be approximated by $v_{mc}(m_c) = m_c\,e_s$, where $e_s = k_{mc}$ [hr$^{-1}$]. During continuous growth, the external substrate concentration $S$ is approximately constant, with the parameter $e_s$ now defined as

$$e_s = k_{mc}\,\frac{S}{S + K_{mc}}.$$

The energy dissipation rate is defined by first-order mass-action kinetics:

$$v_d(a^*) = k_d\, a^*, \tag{40}$$

where $k_d$ [hr$^{-1}$] is a catalytic rate constant.

The resource allocation model of microbial growth thus becomes

$$\frac{dc}{dt} = v_{mc}(m_c) - v_{mer}(m_{er}, c, a) - \rho_{mef}\, v_{mef}(m_{ef}, c, a) \\ - \rho_{ru}\,(v_r(r, c, a^*) + v_{mu}(m_u, c, a^*)) - (\mu + \gamma)\, c, \tag{41}$$

$$\frac{du}{dt} = v_{mu}(m_u, c, a^*) - (\mu + \gamma)\, u, \tag{42}$$

$$\frac{dm_u}{dt} = \chi_u\, v_r(r, c, a^*) - (\mu + \gamma)\, m_u, \tag{43}$$

$$\frac{dr}{dt} = \chi_r\, v_r(r, c, a^*) - (\mu + \gamma)\, r, \tag{44}$$

$$\frac{dm_c}{dt} = \chi_c\, v_r(r, c, a^*) - (\mu + \gamma)\, m_c, \tag{45}$$

$$\frac{dm_{er}}{dt} = \chi_{er}\, v_r(r, c, a^*) - (\mu + \gamma)\, m_{er}, \tag{46}$$

$$\frac{dm_{ef}}{dt} = \chi_{ef}\, v_r(r, c, a^*) - (\mu + \gamma)\, m_{ef}, \tag{47}$$

$$\frac{da^*}{dt} = n_{mer}\, v_{mer}(m_{er}, c, a) + n_{mef}\, v_{mef}(m_{ef}, c, a) \\ - n_r\, v_r(r, c, a^*) - n_{mu}\, v_{mu}(m_u, c, a^*) - v_d(a^*), \tag{48}$$

with

$$\mu = \beta\,(v_{mc}(m_c) - v_{mer}(m_{er}, c, a) - \rho_{mef}\, v_{mef}(m_{ef}, c, a) \\ - (\rho_{ru} - 1)\,(v_r(r, c, a^*) + v_{mu}(m_u, c, a^*))) - \gamma. \tag{49}$$

Since it holds by *Equation 14* that

$$1/\beta = u + c + m_c + m_{er} + m_{ef} + r + m_u, \tag{50}$$

we can omit the differential equations for one of the variables in the right-hand side. Given that $u$ is not playing a role in any of the kinetic rates, we usually eliminate *Equation 42*.

Note that in the above model, like in other resource allocation models (*Erickson et al., 2017*), resource allocation parameters and proteome fractions coincide at steady state. For example, from the steady-state equation for ribosomes, $\chi_r\, v_r = (\mu + \gamma)\, r$, and the steady-state equation for total proteins, $v_r = (\mu + \gamma)\, p$, it follows that $\chi_r = r/p$.

## Model variant with an additional growth-rate-independent protein category

The model described above includes a residual category of proteins, consisting of proteins other than ribosomes and translation-affiliated proteins ($R$), enzymes in central carbon metabolism ($M_c$), or enzymes in energy metabolism ($M_{er}$ and $M_{ef}$). This category $M_u$ carries a flux, because it includes the machinery for the synthesis of macromolecules other than proteins, in particular RNA and DNA. Moreover, we allow the fraction of the proteome occupied by this category to vary with the particular resource allocation strategy adopted, and therefore with the growth rate.

The fact that the proteome fraction of $M_u$ may change with the growth rate and that it carries a flux distinguishes it from a residual category of housekeeping proteins that is found in other models of microbial growth (*Scott et al., 2010*; *Mori et al., 2016*). The latter protein category (usually indicated by $Q$) is not accessible to growth-rate-dependent proteome adjustments and carries no flux. Its size can be determined in different ways, most rigorously as the sum of the offsets of the

linear relation between growth rate and proteome fraction of the individual protein categories (**Hui et al., 2015**).

We developed a variant of the model used in this study that includes such a growth-rate-independent category $Q$. First of all, for each of the other protein categories, we distinguished a growth-rate-independent and -dependent part, indicated by the superscripts 0 and $\mu$, respectively. For example, for ribosomes and translation-affiliated proteins, we have $R = R^0 + R^\mu$. Second, we defined $Q$ as consisting of the growth-rate-independent parts of the other protein categories:

$$Q = R^0 + M_c^0 + M_{er}^0 + M_{ef}^0 + M_u^0. \tag{51}$$

Following these notations, the total cellular biomass $B$ [gDW] is now defined as

$$B = \beta \left( Q + R^\mu + M_c^\mu + M_{er}^\mu + M_{ef}^\mu + M_u^\mu + C + U \right), \tag{52}$$

where in what follows we drop the superscripts for the growth-rate-dependent parts of the protein categories. Notice that, like in the reference model, ATP and ADP are not included in the biomass.

Following the same steps as for the reference model, a system of ordinary differential equations can be derived. The only differences with **Equations 41–49** are that an additional equation for the category $Q$ is added:

$$\frac{dq}{dt} = \chi_q \, v_r(r, c, a^*) - (\mu + \gamma) \, q. \tag{53}$$

Moreover, the sum of biomass components is given by

$$1/\beta = q + m_c + m_{er} + m_{ef} + r + m_u + u + c, \tag{54}$$

and the sum of resource allocation parameters is extended with $\chi_q$:

$$\chi_q + \chi_r + \chi_c + \chi_{er} + \chi_{ef} + \chi_u = 1. \tag{55}$$

Note that, while the model has a very similar structure as the reference model of **Equations 41–49**, the interpretation of the protein concentrations $m_c$, $r$, $m_{er}$, $m_{ef}$, and $m_u$ has changed: instead of denoting the total enzyme and ribosome concentrations, they now refer to the growth-rate-dependent part of these concentrations.

## Comparison with other coarse-grained resource allocation models

The model of **Figure 1** differs in several assumptions from previously proposed resource allocation models of microbial growth. We summarize these differences below, focusing the comparison on coarse-grained models. That is, we do not consider fine-grained models on the genome scale used in constraint-based analysis (**Cheng et al., 2019**; **Adadi et al., 2012**; **Mori et al., 2016**; **Reimers et al., 2017**; **Wortel et al., 2018**).

A first class of models takes into account either the carbon or energy balance, but not both (**Molenaar et al., 2009**; **Scott et al., 2010**; **Scott et al., 2014**; **Maitra and Dill, 2015**; **Giordano et al., 2016**; **Weiße et al., 2015**; **Bosdriesz et al., 2015**; **Erickson et al., 2017**; **Towbin et al., 2017**; **Dourado and Lercher, 2020**; **Mairet et al., 2021**). Typical examples are the classical model of **Scott et al., 2010**, which describes mass flow from substrate to different categories of proteins, and the model of **Maitra and Dill, 2015**, which provides a balance of ATP produced from the substrate and ATP consumed for protein synthesis. These models have successfully reproduced the ribosomal growth law, that is, the linear relation between growth rate and the ribosomal protein fraction, and other empirical regularities. However, apart from the presence of an occasional dissipation term, all substrate is used for biomass synthesis. Therefore, the growth yield as defined by **Equation 2** does not vary with resource allocation. For our purpose, we need to be able to take into account that the use of substrate for ATP production is accompanied by the outflow of $CO_2$ and the secretion of acetate, thus lowering the growth yield.

A second class of models takes into account the coupling of the carbon and energy balances, but describes the latter as fluxes of carbon and energy without specifying the underlying reaction kinetics (**Basan et al., 2015a**; **Mori et al., 2019**). For example, in the model of **Basan et al., 2015a**, fluxes in energy metabolism are modeled as the product of the proteome fraction of enzymes in respiration

or fermentation multiplied by a corresponding efficiency coefficient. The energy coefficients express the ATP yield per unit of protein in the respiration and fermentation pathways, respectively. The coefficients are constant and therefore cannot express differences in the utilization of enzymes depending on the concentrations of central carbon metabolites and energy cofactors. These concentrations may change with the resource allocation strategy and lead to a higher saturation of enzymes, which we hypothesized as an explanation for high-rate, high-yield growth of *E. coli*. In addition, this category of models equates biomass with proteins, like the other models cited above. This does not allow the total protein concentration to vary and a trade-off between protein and metabolite concentrations to occur. In the next section, we precisely define the additional modeling assumptions that allow our model to be reduced to the model of *Basan et al., 2015a*.

A third class of models does provide a kinetic description of all fluxes in the model and does include metabolites in the biomass definition, although ignoring other macromolecules (*Zavřel et al., 2019*; *Faizi et al., 2018*). The model of *Zavřel et al., 2019*, is closest to our model, but since it describes growth of cyanobacteria, it does not include alternative ATP production pathways and therefore does not account for differences in growth yield depending on the investment of cellular resources in respiration or fermentation. Moreover, the analysis of this model is focused on accounting for the experimentally observed growth rate of cyanobacteria under different light intensities. This has motivated the choice to look for resource allocation strategies optimizing the growth rate for each light intensity rather than scanning the space of possible resource allocation strategies in order to predict the variability of rate-yield phenotypes.

The model presented in this work could be further extended by taking into account additional features of some of the models cited above. For example, instead of treating resource allocation strategies as an input to the model (*Figure 2—figure supplement 2*), they could be defined as a function of the bacterial physiology, for example, translation activity (*Scott et al., 2014*; *Maitra and Dill, 2015*; *Giordano et al., 2016*; *Weiße et al., 2015*; *Bosdriesz et al., 2015*; *Erickson et al., 2017*; *Towbin et al., 2017*). This would allow, among other things, to account for the adaptation of resource allocation during dynamic transitions between states of balanced growth. As another example, our model could be extended to allow the uptake of alternative carbon sources (*Erickson et al., 2017*; *Towbin et al., 2017*), which would allow the modeling of diauxic growth behavior. The short summary in this section describes the main differences between the model of *Figure 1* and some major previous work, but cannot do complete justice to the rich diversity of results in the literature. We refer to article-length reviews on coarse-grained resource allocation models and microbial growth for more extensive information (*Scott et al., 2014*; *Kafri et al., 2016*; *de Jong et al., 2017*; *Bruggeman et al., 2020*).

## Simplified coarse-grained resource allocation models

In this section, we discuss two simplifications of the model that (i) allow its predictions to be analyzed along the Pareto frontier of growth rate and growth yield in *Figure 2*, in order to explore the relation between resource allocation and growth, and (ii) allow the predictions to be compared with previous work, in particular the model of *Basan et al., 2015a*.

### Model simplification and analysis along the Pareto frontier

We analyze the model of *Equations 41–50*, with the reaction rates given by *Equations 33–40*, at steady state, after making a number of simplifying assumptions that are valid along the Pareto frontier of growth rate and growth yield shown in *Figure 2*. Using this simplified model, the decrease of the maximum yield with increasing growth rate can be traced back to qualitative changes in resource allocation parameters.

First, we exploit the fact that the contribution to the carbon balance of $CO_2$ loss during macromolecular synthesis is negligible along the Pareto frontier, that is, $v_{mc} \gg (\rho_{ru} - 1)(v_r + v_{mu})$ (*Figure 2—figure supplement 4C*). Second, we exploit the fact that the degradation of macromolecules is negligible at high growth rates, that is, $\gamma \ll \mu$ (*Figure 4—figure supplement 1A*). Third, for the maximum yields along the Pareto frontier, the contribution of fermentation to energy production is negligible ($v_{mef} \approx 0$, *Figure 2—figure supplement 4*). This leads to a simplified definition of growth rate (*Equation 33*):

$$\mu = \beta(v_{mc} - v_{mer}). \tag{56}$$

Fourth, over most of the rate-yield phenotype space, and a fortiori along the Pareto frontier, the rate of synthesis of other macromolecules is strongly coupled to the rate of protein synthesis (*Figure 2—figure supplement 4C*). In other words, $v_{mu} \approx \alpha_1 v_r$, where $\alpha_1 < 1$ is a positive constant. Fifth, in a similar way, the rate of ATP spilling is strongly coupled to the rate of ATP production, that is, $v_d \approx \alpha_2 v_{mer}$, with $\alpha_2 < 1$ a positive constant. This leads to the following simplified energy balance (*Equation 30*):

$$(n_r + \alpha_1 n_{mu}) v_r = (n_{mer} + \alpha_2) v_{mer}. \tag{57}$$

Moreover, the assumptions lead to the following simplification of the biomass composition (*Equation 50*):

$$1/\beta = c + (1 + \alpha_1) p, \quad \text{where } p = m_c + m_{er} + r + m_u, \tag{58}$$

and the resource allocation constraint:

$$\chi_r + \chi_c + \chi_{er} + \chi_u = 1. \tag{59}$$

Sixth, we note that $\chi_u$, and therefore $m_u/p$, are approximately constant at their minimal possible value (*Figure 2—figure supplement 4D* and *Figure 2—figure supplement 1*). Finally, seventh, while the concentrations of central metabolites and ADP vary along the Pareto frontier, we observe that the Michaelian term in the rate equations in which $c$ and $a$ occur are approximately constant, contrary to the term for $a^*$ (*Figure 2—figure supplement 4B*). This leads to simplified expressions for the rate equations of ATP production and consumption (*Equation 36* and *Equation 38*):

$$v_r = k_r' r \frac{a^*}{a^* + K_{ar}}, \tag{60}$$

$$v_{mer} = k_{mer}' m_{er}, \tag{61}$$

where $k_r', k_{mer}'$ are lumped constants incorporating the effect of the metabolite and energy cofactor concentrations on the reaction rates.

With the above simplifications, it becomes possible to explicitly relate the observed increase in growth rate ($\mu \uparrow$) and decrease in growth yield ($Y \downarrow$) to underlying changes in the resource allocation parameters, due to the constraints on carbon and energy fluxes, biomass composition, and resource allocation. We first note that, by *Equation 1*, a decrease in growth yield ($Y \downarrow$) must be accompanied by a decrease of the ratio of the growth rate and the substrate uptake rate: ($\mu/v_{mc}$) $\downarrow$. Because $\mu \uparrow$ this necessarily implies $v_{mc} \uparrow$, that is, the substrate rate must increase along the Pareto frontier. Furthermore, by substituting the simplified growth rate expression of *Equation 56* into the yield definition, we obtain the expression

$$Y = 1 - \frac{v_{mer}}{v_{mc}}, \tag{62}$$

where $Y \downarrow$ implies ($v_{mer}/v_{mc}$) $\uparrow$, that is, the fraction of substrate dedicated to ATP production increases for higher growth rates along the Pareto frontier. Because $v_{mc} \uparrow$, it must also hold that $v_{mer} \uparrow$. With the simplified energy balance of *Equation 57*, $v_{mer} \uparrow$ implies $v_r \uparrow$. Moreover, from the proportionality of $v_{mer}$ and $v_d$, it follows that $v_d \uparrow$ too. In summary, the flux of carbon underlying microbial growth increases with higher growth rate along the Pareto frontier, as verified in *Figure 2—figure supplement 4C*.

Does the increase in protein synthesis rate lead to a higher (total) protein concentration? The answer is less straightforward than might be thought, because under conditions of balanced growth the protein synthesis rate equals growth dilution of proteins, that is, $v_r = \mu p$. Both $v_r$ and $\mu$ increase, so the direction of increase of $p$ is not obvious from this equation. However, note that with the simplified energy balance of *Equation 57*, the growth yield equation of *Equation 62* can be rewritten as

$$Y = 1 - \frac{n_r + \alpha_1 n_{mu}}{n_{mer} + \alpha_2} \frac{v_r}{v_{mc}}, \tag{63}$$

which with $Y \downarrow$ implies ($v_r/v_{mc}$) $\uparrow$. Now, because ($v_{mc}/\mu$) $\uparrow$, and ($v_r/v_{mc}$)($v_{mc}/\mu$) $= v_r/\mu$, it follows that ($v_r/\mu$) $\uparrow$, and therefore $p \uparrow$. That is, in order to facilitate the higher flux of carbon through the bacteria,

a higher protein concentration is needed. By the constant total biomass concentration (**Equation 58**), this directly implies that the concentration of central carbon metabolites must decrease ($c \downarrow$). In other words, the trade-off between rate and yield along the Pareto frontier is accompanied by a trade-off between proteins and metabolites. Because the concentration of central carbon metabolites remains largely saturating (**Figure 2—figure supplement 4B**), however, the decrease of the concentration does not much affect its driving force in the reactions of energy metabolism and macromolecular synthesis.

How do the concentrations of the individual protein classes change when the growth rate increases along the Pareto frontier? With the definition of the substrate uptake rate, $v_{mc} = m_c e_s$, we immediately find that $v_{mc} \uparrow$ implies $m_c \uparrow$. From the simplified rate equation for energy metabolism (**Equation 61**) it also follows that $m_{er} \uparrow$. Determining the direction of change of the ribosome concentration is less straightforward. Note that the simplified rate equation can be rewritten as follows:

$$\frac{v_r}{a^*} = k_r' \, r \, \frac{1}{a^* + K_{ar}},\tag{64}$$

Because $v_r \sim v_{mer} \sim v_d \sim a^*$, the ratio $v_r/a^*$ remains constant for increasing $\mu$ along the Pareto frontier. Because $v_d \uparrow$, we have $a^* \uparrow$, so that it follows that $r \uparrow$. In conclusion, not only the total protein concentration, but the concentrations of all enzymes and the ribosomes increase (**Figure 2—figure supplement 4A**).

The fact that the steady-state concentration of a protein category increases does not imply that the corresponding resource allocation parameter also increases. Since the total protein concentration $p$ increases, even for constant resource allocation, the concentration of the protein category increases. This is the case for the category of other proteins: $\chi_u$ is constant, so that with $m_u = \chi_u p$, it follows that $m_u \uparrow$. Dividing **Equations 60 and 61** by $p$, we obtain the following expressions:

$$\frac{v_{mer}}{p} = k_{mer}' \chi_{er},\tag{65}$$

$$\mu = k_r' \chi_r \frac{a^*}{a^* + K_{ar}}.\tag{66}$$

From the energy balance, we find that $v_{mer}/p$ must change in the same direction as $v_r/p = \mu$, that is, increase along the Pareto frontier. As a consequence, $\chi_{er} \uparrow$. Since both $\mu \uparrow$ and $a^* \uparrow$, the second equation does not unambiguously fix the direction of change of $\chi_r$, which depends on the ratio of $\mu$ and $a^*/(a^* + K_{ar})$. In particular, if this ratio remains constant, then $\chi_r$ also remains constant, whereas if the ratio increases, then $\chi_r \uparrow$. **Figure 2—figure supplement 4B** shows that these two cases both occur along the Pareto frontier. $\chi_r$ remains constant for a large range of growth rates: the ribosome concentration nevertheless increases due to the higher total protein concentration. This is not sufficient for the highest growth rates, however, where $\chi_r$ needs to increase as well to sustain the higher flux of carbon through the bacteria. In both cases, however, the resource allocation constraint of **Equation 59** forces $\chi_c$ to decrease (**Figure 2—figure supplement 4D**). That is, whereas the concentration of $m_c$ increases, the fraction of resources devoted to the uptake and metabolism of the carbon source decreases, so as to free resources for energy metabolism and protein synthesis at the higher growth rate. The higher investment in protein synthesis, and the corresponding higher energy demand and $CO_2$ loss through respiration, explain the lower growth yield.

The above analysis thus explicitly relates the observed change in rate and yield along the Pareto surface with the changes in fluxes, concentrations, and resource allocation parameters shown in **Figure 2—figure supplement 4**. We emphasize that some of the assumptions underlying the model simplifications are specific for the Pareto frontier, such as the restriction of energy metabolism to respiration. As a consequence, accounting for a change in rate and yield in terms of changes in resource allocation may be different in other regions of the rate-yield phenotype space.

### Reduction to resource allocation model of Basan et al.

We simplify the model of **Equations 41–50**, with the reaction rates given by **Equations 33–40**, to the model of **Basan et al., 2015a**, by making a number of additional assumptions.

First, assume that the concentrations of central carbon metabolites, energy cofactors, and other macromolecules are constant and that their contribution to the biomass balance can be ignored. This leads to the revised rate equations

$$v_r(r) = k'_r\, r, \quad v_{mer}(m_{er}) = k'_{mer}\, m_{er}, \quad v_{mef}(m_{ef}) = k'_{mef}\, m_{ef}, \ldots \tag{67}$$

where the constants $k'_r, k'_{mer}, k'_{mef}, \ldots$ lump the effects of the catalytic efficiency of the enzymes and the concentrations of central carbon metabolites and energy cofactors. Moreover, the assumption reduces the biomass to total *protein* mass:

$$B = \beta'\,(M_u + R + M_c + M_{er} + M_{ef}), \tag{68}$$

and consequently,

$$1/\beta' = m_c + m_{er} + m_{ef} + r + m_u, \tag{69}$$

where $1/\beta'$ is the total protein concentration. Note that, with this simplification, the total protein concentration is constant, independently from the resource allocation strategy adopted by the cell.

A second assumption is that energy dissipation and the degradation of macromolecules can be neglected, which means that $\gamma = k_d = 0$. The absence of protein degradation, together with the revised biomass definition, leads to the proportionality of growth rate and protein synthesis rate:

$$\mu = \frac{1}{B}\frac{dB}{dt} = \frac{\beta'}{B}\frac{dV_r}{dt} = \beta'\,v_r. \tag{70}$$

The absence of energy dissipation, in combination with the omission of other macromolecules, leads to a revised energy balance:

$$0 = n_{mer}\,v_{mer} + n_{mef}\,v_{mef} - n_r\,v_r, \tag{71}$$

which with **Equation 70** gives

$$n_{mer}\,v_{mer} + n_{mef}\,v_{mef} = \frac{n_r}{\beta'}\,\mu, \tag{72}$$

and hence

$$n_{mer}\,k'_{mer}\,m_{er} + n_{mef}\,k'_{mef}\,m_{ef} = \frac{n_r}{\beta'}\,\mu. \tag{73}$$

Third, assume that in the mass balance for the central carbon metabolites the contributions from growth dilution and spontaneous degradation can be neglected in comparison with the utilization of these metabolites for protein synthesis. Then **Equation 41** reduces to

$$v_{mc} = v_{mer} + \rho_{mef}\,v_{mef} + \rho_{ru}\,v_r, \tag{74}$$

which with the energy balance of **Equation 71** can be rewritten as

$$v_{mc} = \left(\rho_{mef} - \frac{n_{mef}}{n_{mer}}\right) v_{mef} + \left(\rho_{ru} - \frac{n_r}{n_{mer}}\right) v_r \approx \left(\rho_{ru} - \frac{n_r}{n_{mer}}\right) v_r, \tag{75}$$

bearing in mind that both $\rho_{mef}$ and $n_{mef}/n_{mer}$ assume values in the range 1–2. That is, the substrate uptake rate is approximately proportional to the protein synthesis rate.

Now, using the protein mass balance of **Equation 69**, we can express the total concentration of energy proteins as follows:

$$m_{er} + m_{ef} = (1/\beta' - m_u) - m_c - r, \tag{76}$$

which with **Equation 75** and the rate equations for the glucose uptake and protein synthesis rates give

$$m_{er} + m_{ef} = (1/\beta' - m_u) - \frac{1}{\beta'\,e_s}\left(\rho_{ru} - \frac{n_r}{n_{mer}} + \frac{e_s}{k'_r}\right)\mu \tag{77}$$

$$= m_e^{max} - \frac{\alpha}{\beta'} \mu, \tag{78}$$

where $m_e^{max} = 1/\beta' - m_u$, making the further assumption that $m_u = \beta'/\chi_u$ is constant, and $\alpha = (1/e_s)(\rho_{ru} - n_r/n_{mer} + e_s/k_r')$. *Equation 78* expresses that the concentration (or equivalently for constant $1/\beta'$, the fraction) of proteins involved in energy metabolism linearly decreases with the growth rate. *Basan et al., 2015a*, posit the same linear relationship, based on proteomics data for the NCM3722 strain (*Hui et al., 2015*).

When combining *Equations 72 and 78*, we can solve for the two unknowns $m_{ef}$ and $m_{er}$ as a function of $\mu$:

$$m_{ef} = \frac{\left(\frac{n_r}{\beta' n_{mer} k_{mer}'} + \frac{\alpha}{\beta'}\right) \mu - m_e^{max}}{\left(\frac{n_{mef}}{n_{mer}} \frac{k_{mef}'}{k_{mer}'} - 1\right)}, \tag{79}$$

$$m_{er} = \frac{\left(\frac{n_r}{\beta' n_{mef} k_{mef}'} + \frac{\alpha}{\beta'}\right) \mu - m_e^{max}}{\left(\frac{n_{mer}}{n_{mef}} \frac{k_{mer}'}{k_{mef}'} - 1\right)}. \tag{80}$$

The model is only valid in the range of growth rates where both concentrations are positive. By means of the simplified expressions for the respiration and fermentation fluxes (*Equation 67*), we can compute the total ATP production rate $n_{mer} k_{mer}' m_{er} + n_{mef} k_{mef}' m_{ef}$ using the above expressions. The ATP production rates of *Basan et al., 2015a*, are rescaled by using protein fractions instead of protein concentrations, which gives rise to $J_{E,f} \equiv n_{mef} k_{mef}' \beta' m_{ef}$ and $J_{E,r} \equiv n_{mer} k_{mer}' \beta' m_{er}$. Developing the expressions for $J_{E,f}$ and $J_{E,r}$ by means of *Equations 79 and 80* yields equations that are equivalent to Eqs S12 and S13, respectively, of *Basan et al., 2015a*, after appropriately renaming the parameters ($\epsilon_f = n_{mef} k_{mef}'$, $\epsilon_r = n_{mer} k_{mer}'$, $\sigma = n_r$, $b = \alpha$, and $\phi_{E,max} = \beta' m_e^{max}$).

The model of *Basan et al., 2015a*, predicts a trade-off between respiration and fermentation when the growth rate increases, because the protein cost of fermentation is lower than the protein cost of respiration, that is, $n_{mef} k_{mef}' > n_{mer} k_{mer}'$. This relation, which is preserved for the parameter values for growth on glucose in our model (*Appendix 2—table 2*), implies that when the growth rate increases, the concentration of fermentation enzymes increases at the expense of the concentration of respiration enzymes. Due to the lower protein cost of fermentation, however, the total ATP production rate increases.

As explained in the Discussion section, our model makes less stringent assumptions, which notably allows metabolite and total protein concentrations to vary with different resource allocation strategies. As a consequence, there are ways to increase the total ATP production rate without shifting resources from energy-efficient but costly respiration (high $n_{mer}$ but low $n_{mer} k_{mer}'$) to energy-inefficient but cheap fermentation (low $n_{mef}$ but high $n_{mef} k_{mef}'$). In particular, in our model, growth rate and growth yield can be simultaneously increased, by trading off proteins against metabolites, thus enabling a more efficient use of proteomic resources.

## Appendix 2

### Model calibration

#### Reference datasets and model calibration strategy

Model calibration was performed using published reference datasets with measurements of growth rates and fluxes (*Haverkorn van Rijsewijk et al., 2011*; *Gerosa et al., 2015*; *Peebo et al., 2015*), protein concentrations (*Schmidt et al., 2016*), and metabolite concentrations (*Gerosa et al., 2015*; *Bennett et al., 2009*; *Park et al., 2016*). The datasets concern the *E. coli* BW25113 strain: either batch growth in minimal medium with glucose or glycerol, or continuous growth in minimal medium with glucose. We also used auxiliary data for other strains at comparable growth rates, when necessary. Moreover, we adopted a top-down model calibration procedure, in order to enforce consistency across different data types.

#### Step 1

We used the total biomass density and measured biomass proportions of proteins and metabolites to derive total protein and metabolite concentrations.

#### Step 2

We used proteomics and metabolomics data to derive the concentrations of the different protein and metabolite categories distinguished in the model.

#### Step 3

We used published data to reconstruct the biomass degradation rate for growth on glucose and glycerol.

#### Step 4

We used the measured substrate uptake and acetate secretion rates, the growth rate, and the derived protein and metabolite concentrations to reconstruct the other metabolic fluxes from the carbon mass balance.

#### Step 5

We derived the kinetic parameters from literature data and from the fluxes and the concentrations obtained in the previous steps.

The above procedure does not require computational parameter fitting, since all parameters are unambiguously fixed by the data, literature information, and suitable hypotheses motivated by experimental results. We explain the procedure in detail for batch growth of the reference strain, and then summarize the results for continuous growth and for an alternative strain. In what follows, observed fluxes, growth rates, and concentrations, as well as kinetic parameters derived from this information, are denoted by a hat $\hat{\cdot}$ symbol.

#### Reconstruction of concentrations, rates, and fluxes for batch growth

##### Total biomass concentration $1/\beta$

The total concentration of biomass in the cell, in units Cmmol gDW$^{-1}$, is referred to in our model as $1/\beta$. Using the definition of yield (*Equation 2* in the main text), we have $1/\beta = Y v_{mc}/\mu$. With the values reported by *Morin et al., 2016*, for the MG1655 strain, we estimate

$$1/\hat{\beta} = 40.65 \text{ Cmmol gDW}^{-1}. \tag{81}$$

This value is close to the theoretical value obtained from the fact that the carbon mass fraction of biomass is approximately 0.5 (*Folsom and Carlson, 2015*):

$$1 \text{ gDW} = 0.5 \text{ CgDW} = \frac{0.5}{12.01 \cdot 10^{-3}} = 41.6 \text{ Cmmol}, \tag{82}$$

where CgDW refers to Cgram dry weight and the molecular weight of C equals 12.01 g mol$^{-1}$. Another way to determine the total biomass concentration is to use the estimated elementary biomass composition of *E. coli*. *von Stockar and Liu, 1999*, report $CH_{1.77}O_{0.49}N_{0.24}$, which with the

molecular weights of H, O, and N yields an estimate of 40.03 Cmmol gDW$^{-1}$, again close to the value proposed above.

## Metabolite concentrations $c$, $a$, $a^*$, and $a_0$

A recent quantification of 43 abundant metabolites in the *E. coli* BW25113 strain growing in minimal medium with glucose or glycerol learns that these metabolites sum up to a concentration of 0.89 Cmmol gDW$^{-1}$ and 0.69 Cmmol gDW$^{-1}$, respectively (*Gerosa et al., 2015*). When comparing the metabolites quantified by Gerosa et al. with those measured in a broader screen carried out by *Park et al., 2016*, we conclude that 56% of the metabolite mass is covered by the study of Gerosa et al. As a consequence, we estimate the total metabolite concentrations in growth on glucose and glycerol to be 1.6 Cmmol gDW$^{-1}$ and 1.2 Cmmol gDW$^{-1}$, respectively. With the biomass density value of $1/\hat{\beta}$, these concentrations correspond to 3.9% and 3.0% of the total biomass. The estimates correspond well to the older estimate that metabolites constitute 3.5% of the total biomass, obtained for the *E. coli* B/r strain growing at a rate of around 1 hr$^{-1}$ (*Neidhardt, 1996*), and a more recent estimate of 2.9% (*Feist et al., 2007*).

Analysis of the data of *Gerosa et al., 2015*, shows that central carbon metabolites account for 22% of the total free metabolite concentration during growth in minimal medium with glucose. We therefore estimate the concentration of the pool of central metabolites in this condition as

$$\hat{c} = 0.22 \cdot 1.6 = 0.35 \text{ Cmmol gDW}^{-1}. \tag{83}$$

For growth on glycerol, the fraction of central metabolites is 17%, so that

$$\hat{c} = 0.17 \cdot 0.92 = 0.20 \text{ Cmmol gDW}^{-1}. \tag{84}$$

As explained in Appendix 1, we consider pools of charged and discharged energy cofactors expressed as ATP equivalents. Following the arguments of *Basan et al., 2015b*, 1 NADH or 1 NADPH molecule can be converted into 2 ATP molecules. With these conversion factors, we obtain from the ATP/ADP, NADH/NAD+, NADPH/NADP+ concentrations reported by *Gerosa et al., 2015*, the following estimates of the concentrations of energy cofactors during growth on glucose:

$$\hat{a}^* = 0.009 \text{ mmol gDW}^{-1}, \quad \hat{a} = 0.011 \text{ mmol gDW}^{-1}. \tag{85}$$

The values for growth on glycerol are

$$\hat{a}^* = 0.005 \text{ mmol gDW}^{-1}, \quad \hat{a} = 0.010 \text{ mmol gDW}^{-1}. \tag{86}$$

Accordingly, $\hat{a}_0 = 0.020$ mmol gDW$^{-1}$ for growth on glucose, and $\hat{a}_0 = 0.015$ mmol gDW$^{-1}$ for growth on glycerol. Recall that ATP and ADP are not included in the mass balance (Appendix 1).

## Protein concentrations $m_u$, $r$, $m_c$, and $m_{er} + m_{ef}$

Estimates of the total protein concentration of *E. coli* reported in the literature vary significantly (*Milo, 2013*). For example, older values for the B/r strain indicate a mass fraction of 0.55 (*Neidhardt, 1996*), for cells growing with a doubling time of 40 min ($\mu = 1.04$ hr$^{-1}$). In their quantification of the NCM3722 strain, *Basan et al., 2015b*, report a value of 0.67 for the protein fraction of dry biomass of cells growing in batch in minimal medium with glucose at a rate of 0.99 hr$^{-1}$. For growth on other carbon sources at rates of 0.42–0.43, this fraction increases to 0.73–0.76. *Valgepea et al., 2013*, find that for glucose-limited growth in a bioreactor at a rate of 0.4 hr$^{-1}$, the MG1655 strain, another K-12 descendant, has a protein dry biomass fraction equal to 0.53. *Milo, 2013*, cites an old reference value of 0.24 g mL$^{-1}$, which with an estimated total (dry) biomass concentration of 0.33 g mL$^{-1}$ yields a protein mass fraction of 0.73, in agreement with the values of Basan et al.

We based our estimates on the data from *Basan et al., 2015b*, who report protein dry mass fractions for batch growth in different media at different growth rates. From within the range of reported values, we chose the dry mass fractions for growth rates corresponding to the observed growth rates of the BW25113 strain in minimal medium with glucose or glycerol (*Appendix 2—figure 1*). This resulted in protein dry mass fractions of 0.72 (glucose) and 0.73 (glycerol). Like the carbon mass fraction of biomass, the carbon mass fraction of protein is approximately 0.5 (Supplementary table 3 in *Feist et al., 2007*). As a consequence, the above protein dry mass fractions also denote the protein fractions of the total biomass concentration expressed in units Cmmol gDW$^{-1}$.

In our model, the process of protein synthesis includes the synthesis of amino acids from central metabolites (Appendix 1). For reasons of consistency, we therefore add the concentrations of free amino acids to the total protein concentration. Given that amino acids account for around 50% of metabolites (*Bennett et al., 2009*), and the total metabolite concentrations were estimated to take up 3.9% and 3.0% of the total biomass during growth on glucose and glycerol, respectively, the total protein concentrations amount to a fraction of 0.74 of the total biomass density, for both glucose and glycerol.

The proteomics data of *Schmidt et al., 2016*, provide information on the mass fractions of each of the protein categories distinguished in the model. This information, together with the total protein concentration established above, allows us to compute the concentrations $m_u$, $r$, $m_c$, and $m_{er} + m_{ef}$ (in units Cmmol gDW$^{-1}$). The use of mass fractions, instead of the absolute values also reported by Schmidt et al., has the advantage of ensuring the consistency of the protein concentrations with the uptake, secretion, and growth rates reconstructed below. In the case of growth in minimal medium with glucose, we thus estimate that

$$\hat{m}_u = 0.37 \cdot 0.74 \cdot 1/\hat{\beta} = 11.1 \text{ Cmmol gDW}^{-1}, \tag{87}$$

$$\hat{r} = 0.44 \cdot 0.74 \cdot 1/\hat{\beta} = 13.2 \text{ Cmmol gDW}^{-1}, \tag{88}$$

$$\hat{m}_c = 0.09 \cdot 0.74 \cdot 1/\hat{\beta} = 2.7 \text{ Cmmol gDW}^{-1}, \tag{89}$$

$$\hat{m}_{er} + \hat{m}_{ef} = 0.10 \cdot 0.74 \cdot 1/\hat{\beta} = 3.0 \text{ Cmmol gDW}^{-1}. \tag{90}$$

while for minimal medium with glycerol we obtain

$$\hat{m}_u = 0.36 \cdot 0.74 \cdot 1/\hat{\beta} = 10.9 \text{ Cmmol gDW}^{-1}, \tag{91}$$

$$\hat{r} = 0.38 \cdot 0.74 \cdot 1/\hat{\beta} = 11.5 \text{ Cmmol gDW}^{-1}, \tag{92}$$

$$\hat{m}_c = 0.10 \cdot 0.74 \cdot 1/\hat{\beta} = 3.0 \text{ Cmmol gDW}^{-1}, \tag{93}$$

$$\hat{m}_{er} + \hat{m}_{ef} = 0.16 \cdot 0.74 \cdot 1/\hat{\beta} = 4.8 \text{ Cmmol gDW}^{-1}, \tag{94}$$

The above mass fractions correspond to the following resource allocation parameters for growth on glucose:

$$\hat{\chi}_u = 0.37, \quad \hat{\chi}_r = 0.44, \quad \hat{\chi}_c = 0.09, \tag{95}$$

and growth on glycerol:

$$\hat{\chi}_u = 0.36, \quad \hat{\chi}_r = 0.38, \quad \hat{\chi}_c = 0.10. \tag{96}$$

We will discuss in a later section how to distribute the total concentration $\hat{m}_{er} + \hat{m}_{ef}$ over the respiration and fermentation protein classes (and thus determine the resource allocation parameters $\chi_{er}$ and $\chi_{ef}$).

## Concentration of other macromolecules $u$

The biomass definition in the model enforces the concentration $u$ of other macromolecules (RNA, DNA, lipids in the cell membrane) to equal the difference between the total biomass concentration and the sum of the total protein and metabolite concentrations. For growth on glucose, we thus find that

$$\hat{u} = 10.2 \text{ Cmmol gDW}^{-1}, \tag{97}$$

whereas for growth on glycerol, we also obtain

$$\hat{u} = 10.2 \text{ Cmmol gDW}^{-1}. \tag{98}$$

The estimated values, and all other concentration values derived above, are summarized in **Appendix 2—table 1**.

## Degradation rate $\gamma$

The model includes a degradation constant $\gamma$ that accounts for one of the main causes of so-called maintenance costs of the cell, the turnover of macromolecules and other biomass components. We show that the biomass degradation constant can be determined by means of the well-known Pirt model for maintenance, defined by

$$v_{mc} = \frac{\mu}{Y^{max}} + k_m, \tag{99}$$

where $v_{mc}$ [Cmmol gDW$^{-1}$ hr$^{-1}$] is the substrate uptake rate, $Y^{max}$ [gDW Cmmol$^{-1}$] the maximum biomass yield without maintenance, and $k_m$ [Cmmol gDW$^{-1}$ hr$^{-1}$] the so-called maintenance coefficient (**Pirt, 1965**).

By substituting expressions for $Y^{max}$ and $\mu$ from our model (Appendix 1) into **Equation 99**, we obtain

$$
\begin{aligned}
v_{mc} &= \frac{\beta \left( v_{mc} - v_{mer} - \rho_{mef} v_{mef} - (\rho_{ru} - 1)(v_r + v_{mu}) - \gamma/\beta \right)}{\beta \left( v_{mc} - v_{mer} - \rho_{mef} v_{mef} - (\rho_{ru} - 1)(v_r + v_{mu}) \right)} \cdot v_{mc} + k_m \\
&= v_{mc} - \frac{\gamma}{Y^{max}} + k_m,
\end{aligned}
\tag{100}
$$

or

$$\gamma = k_m \cdot Y^{max}. \tag{101}$$

Data for growth of the *E. coli* MG1655 strain in minimal medium with glucose, by **Esquerré et al., 2014**, indicate a maintenance coefficient of $k_m = 0.35$ mmol$_{glc}$ gDW$^{-1}$ hr$^{-1}$ and a maximal yield $Y_{max} = 76.2$ gDW mol$_{glc}^{-1}$, practically identical to the values reported for the same strain in the same medium by **Nanchen et al., 2006** ($k_m = 0.37$ mmol$_{glc}$ gDW$^{-1}$ hr$^{-1}$, $Y_{max} = 76$ gDW mol$_{glc}^{-1}$). Using the values from **Esquerré et al., 2014**, we find $\hat{\gamma} = 0.027$ hr$^{-1}$. By the same reasoning as above, the maintenance rate for growth in minimal medium with glycerol can be obtained. Classical experiments indicate that the rate is 1.2 times the rate for glucose (**Farmer and Jones, 1976**), so $\hat{\gamma} = 0.032$ hr$^{-1}$.

## Substrate uptake flux $v_{mc}$, fermentation flux $v_{mef}$, and biosynthesis fluxes $v_{mu}$, $v_r$

The datasets used from **Haverkorn van Rijsewijk et al., 2011**, and **Gerosa et al., 2015**, consist of measured fluxes and the growth rate of the *E. coli* BW25113 strain, during exponential growth in minimal medium with glucose and glycerol, respectively. In particular, the glucose or glycerol uptake rate $v_{mc}$ [mmol$_{glc/gly}$ gDW$^{-1}$ hr$^{-1}$], the acetate secretion rate $v_{mef}$ [mmol$_{ace}$ gDW$^{-1}$ hr$^{-1}$], and the growth rate $\mu$ [hr$^{-1}$] were measured. The values for glucose are $\hat{v}_{mc} = 8.26$ mmol$_{glc}$ gDW$^{-1}$ hr$^{-1}$, $\hat{v}_{mef} = 4.89$ mmol$_{ace}$ gDW$^{-1}$ hr$^{-1}$, and $\hat{\mu} = 0.61$ hr$^{-1}$. These values are very close to those reported by **Morin et al., 2016**, for the MG1655 strain. In the case of growth on glycerol, we have $\hat{v}_{mc} = 11.3$ mmol$_{gly}$ gDW$^{-1}$ hr$^{-1}$ and $\hat{\mu} = 0.49$ hr$^{-1}$, while the acetate secretion rate was found to be small: $\hat{v}_{mef} = 0.60$ mmol$_{ace}$ gDW$^{-1}$ hr$^{-1}$. (**Gerosa et al., 2015**, actually report a glycerol uptake rate of 10.14 mmol$_{glc}$ gDW$^{-1}$ hr$^{-1}$, but explain that uptake rates were computed by dividing the measured growth rates by the measured biomass yields [see *Extended Experimental Procedures*]. In the case of glycerol, the growth rate and the biomass yield were found to be 0.49 hr$^{-1}$ and 0.47 gDW g$^{-1}$, respectively (*Data S1*), which with a molecular weight of 92.09 g mol$^{-1}$ gives a value of $0.49/(0.47 \cdot 92.09 \cdot 0.001) = 11.3$ mmol gDW$^{-1}$ hr$^{-1}$ for the glycerol uptake rate).

In agreement with the biomass concentration units, we express mass fluxes in terms of the amount of carbon flowing through the system [Cmmol gDW$^{-1}$ hr$^{-1}$]. Bearing in mind that the carbon content of glucose is 6 C and that of acetate 2 C, we obtain the following rates:

$$\hat{v}_{mc} = 8.26 \cdot 6 = 49.6 \, \text{Cmmol gDW}^{-1} \, \text{hr}^{-1}, \tag{102}$$

$$\hat{v}_{mef} = 4.89 \cdot 2 = 9.8 \, \text{Cmmol gDW}^{-1} \, \text{hr}^{-1}. \tag{103}$$

Similarly, for growth on glycerol we have

$$\hat{v}_{mc} = 11.3 \cdot 3 = 33.9 \, \text{Cmmol gDW}^{-1} \, \text{hr}^{-1}, \tag{104}$$

$$\hat{v}_{mef} = 0.60 \cdot 2 = 1.2 \text{ Cmmol gDW}^{-1} \text{ hr}^{-1}, \tag{105}$$

where we have used the fact that the carbon content of glycerol is 3 C.

The measured fluxes, together with the growth and degradation rates and the total biomass concentration, fix the biosynthesis fluxes in the model. This can be shown by rewriting the equations in the model in the following way:

$$v_{mu} = (\mu + \gamma)\, u, \tag{106}$$

$$v_r = (\mu + \gamma)\,(m_u + r + m_c + m_{er} + m_{ef}). \tag{107}$$

Values for $v_{mu}$ and $v_r$ can be directly computed from the values for the concentrations and rates in the right-hand sides of **Equations 106 and 107** that were derived above. This yields for growth on glucose:

$$\hat{v}_{mu} = 6.5 \text{ Cmmol gDW}^{-1} \text{ hr}^{-1}, \tag{108}$$

$$\hat{v}_r = 19.2 \text{ Cmmol gDW}^{-1} \text{ hr}^{-1}, \tag{109}$$

and for growth on glycerol:

$$\hat{v}_{mu} = 5.3 \text{ Cmmol gDW}^{-1} \text{ hr}^{-1}, \tag{110}$$

$$\hat{v}_r = 15.8 \text{ Cmmol gDW}^{-1} \text{ hr}^{-1}. \tag{111}$$

## Respiration flux $v_{mer}$ and $CO_2$ correction factors $\rho_{ru}$ and $\rho_{mef}$

In the flux datasets mentioned above, $CO_2$ released by the cells was not directly measured. The $CO_2$ flux can be derived from the carbon mass balance, bearing in mind that almost all of the carbon not integrated into biomass leaves the cells as $CO_2$ or acetate (**Gerosa et al., 2015**; **Gottschalk, 1986**). The carbon mass balance is given by the definition of the growth rate, which provides an expression for the total $CO_2$ outflux $v_{CO_2}$. We have

$$v_{CO_2} = v_{mer} + (\rho_{mef} - 1)\, v_{mef} + (\rho_{ru} - 1)\,(v_r + v_{mu}) = v_{mc} - v_{mef} - \frac{\mu + \gamma}{\beta}, \tag{112}$$

where $\rho_{ru} - 1 > 0$ is the correction factor accounting for the release of $CO_2$ during the synthesis of amino acids, proteins, and other biomass components and $\rho_{mef} - 1 > 0$ the correction factor accounting for the $CO_2$ released during the conversion of glucose to acetate (Appendix 1). That is, the total $CO_2$ flux is composed of the $CO_2$ released during respiration ($v_{mer}$), fermentation ($(\rho_{mef} - 1)\, v_{mef}$), and the $CO_2$ released during macromolecular synthesis ($(\rho_{ru} - 1)\,(v_r + v_{mu})$). **Basan et al., 2015a**, argue that the latter $CO_2$ outflux is proportional to the growth rate over a wide range of conditions, with a proportionality constant $\eta$:

$$(\rho_{ru} - 1)\,(v_r + v_{mu}) = \eta\, \mu. \tag{113}$$

The value of $\eta$ is estimated at 7.2 Cmmol gDW$^{-1}$ (**Basan et al., 2015a**), so that for a growth rate of .61 hr$^{-1}$ in the case of minimal medium with glucose, the $CO_2$ outflux associated to biosynthesis equals 4.4 Cmmol gDW$^{-1}$ hr$^{-1}$. Moreover, with the values for $v_r$ and $v_{mu}$ derived above, we find

$$\hat{\rho}_{ru} = \frac{\hat{\eta}\, \hat{\mu}}{\hat{v}_r + \hat{v}_{mu}} + 1 = 1.17. \tag{114}$$

That is, 17% of the carbon flux toward macromolecular synthesis is lost as $CO_2$. The total $CO_2$ outflux can be directly computed from **Equation 112**, giving

$$\hat{v}_{CO_2} = 13.9 \text{ Cmmol gDW}^{-1} \text{ hr}^{-1}. \tag{115}$$

For each acetate molecule, one $CO_2$ is produced (**Basan et al., 2015a**), so that $\hat{\rho}_{mef} = 1.5$. The respiration-associated $CO_2$ outflux can now be reconstructed as

$$\hat{v}_{mer} = \hat{v}_{CO_2} - (\hat{\rho}_{mef} - 1)\, v_{mef} - (\hat{\rho}_{ru} - 1)\,(\hat{v}_r + \hat{v}_{mu}) = 4.6 \text{ Cmmol gDW}^{-1} \text{ h}^{-1}. \tag{116}$$

In the case of growth on glycerol, we find $\hat{v}_{CO_2} = 11.5$ Cmmol gDW$^{-1}$ hr$^{-1}$ and $\hat{v}_{mer} = 7.3$ Cmmol gDW$^{-1}$ hr$^{-1}$, while the value for $\rho_{ru}$ is the same as for glucose (1.17). The reconstructed flux measurements are summarized in *Appendix 2—table 1*, whereas the flux correction factors for $CO_2$ release are included in *Appendix 2—table 2*.

**Appendix 2—table 1.** Reconstruction of growth and degradation rates, fluxes, and concentrations from published datasets for the case of batch growth of *E. coli* in minimal medium with glucose or glycerol, as explained in the text.

The uncertainty intervals for the rates, fluxes, and metabolite concentrations are standard deviations reported in the source publications, after unit conversion. The uncertainty interval for the total biomass concentration was obtained by propagating the errors of the measurements in the right-hand side of $1/\beta = Yv_{mc}/\mu$ (*Morin et al., 2016*). The uncertainty interval for the total protein concentration was obtained by combining the latter error with the standard error of the mean for the total protein fraction predicted by the linear model fitted to the data in *Appendix 2—figure 1*. The resulting error was distributed over the individual protein categories according to their mass fractions. References: *a* **Haverkorn van Rijsewijk et al., 2011**, *b* **Gerosa et al., 2015**, *c* **Esquerré et al., 2014**, *d* **Farmer and Jones, 1976**, *e* **Morin et al., 2016**, *f* **Basan et al., 2015b**, *g* **Schmidt et al., 2016**, *h* **Park et al., 2016**.

| Rates | Unit | Glucose | Glycerol | Reference |
|---|---|---|---|---|
| $\hat{\mu}$ | hr$^{-1}$ | $0.61 \pm 0.01$ | $0.49 \pm 0.01$ | $a, b$ |
| $\hat{\gamma}$ | hr$^{-1}$ | 0.027 | 0.032 | $c, d$ |
| Uptake, secretion, biosynthesis fluxes | | | | |
| $\hat{v}_{mc}$ | Cmmol gDW$^{-1}$ hr$^{-1}$ | $49.6 \pm 5$ | $33.9 \pm 1.0$ | $a$ |
| $\hat{v}_{mer}$ | Cmmol gDW$^{-1}$ hr$^{-1}$ | 4.6 | 7.3 | Derived |
| $\hat{v}_{mef}$ | Cmmol gDW$^{-1}$ hr$^{-1}$ | $9.8 \pm 3.0$ | $1.2 \pm 0.4$ | $a$ |
| $\hat{v}_{mu}$ | Cmmol gDW$^{-1}$ hr$^{-1}$ | 6.5 | 5.3 | Derived |
| $\hat{v}_r$ | Cmmol gDW$^{-1}$ hr$^{-1}$ | 19.2 | 15.8 | Derived |
| Total biomass concentration | | | | |
| $1/\hat{\beta}$ | Cmmol gDW$^{-1}$ | $40.65 \pm 2.0$ | $40.65 \pm 2.0$ | $e$ |
| Protein concentrations | | | | |
| $\hat{m}_u$ | Cmmol gDW$^{-1}$ | $11.1 \pm 0.5$ | $10.9 \pm 0.5$ | $e, f, g$ |
| $\hat{r}$ | Cmmol gDW$^{-1}$ | $13.2 \pm 0.6$ | $11.5 \pm 0.6$ | $e, f, g$ |
| $\hat{m}_c$ | Cmmol gDW$^{-1}$ | $2.7 \pm 0.1$ | $3.0 \pm 0.1$ | $e, f, g$ |
| $\hat{m}_{er} + \hat{m}_{ef}$ | Cmmol gDW$^{-1}$ | $3.0 \pm 0.1$ | $4.8 \pm 0.2$ | $e, f, g$ |
| $\hat{m}_{er}$ | Cmmol gDW$^{-1}$ | 1.9 | 4.4 | Derived |
| $\hat{m}_{ef}$ | Cmmol gDW$^{-1}$ | 1.1 | 0.47 | Derived |
| Metabolite concentrations | | | | |
| $\hat{c}$ | Cmmol gDW$^{-1}$ | $0.35 \pm 0.002$ | $0.20 \pm 0.002$ | $b, h$ |
| $\hat{a}^*$ | mmol gDW$^{-1}$ | $0.009 \pm 0.0002$ | $0.005 \pm 0.0003$ | $b$ |
| $\hat{a}$ | mmol gDW$^{-1}$ | $0.011 \pm 0.0006$ | $0.010 \pm 0.0005$ | $b$ |

*Appendix 2—table 1 Continued on next page*

*Appendix 2—table 1 Continued*

| Rates | Unit | Glucose | Glycerol | Reference |
|---|---|---|---|---|
| $\hat{a}_0$ | mmol gDW$^{-1}$ | $0.020 \pm 0.0008$ | $0.015 \pm 0.0008$ | *b* |
| Concentration of other biomass | | | | |
| $\hat{u}$ | Cmmol gDW$^{-1}$ | 10.2 | 10.2 | Derived |

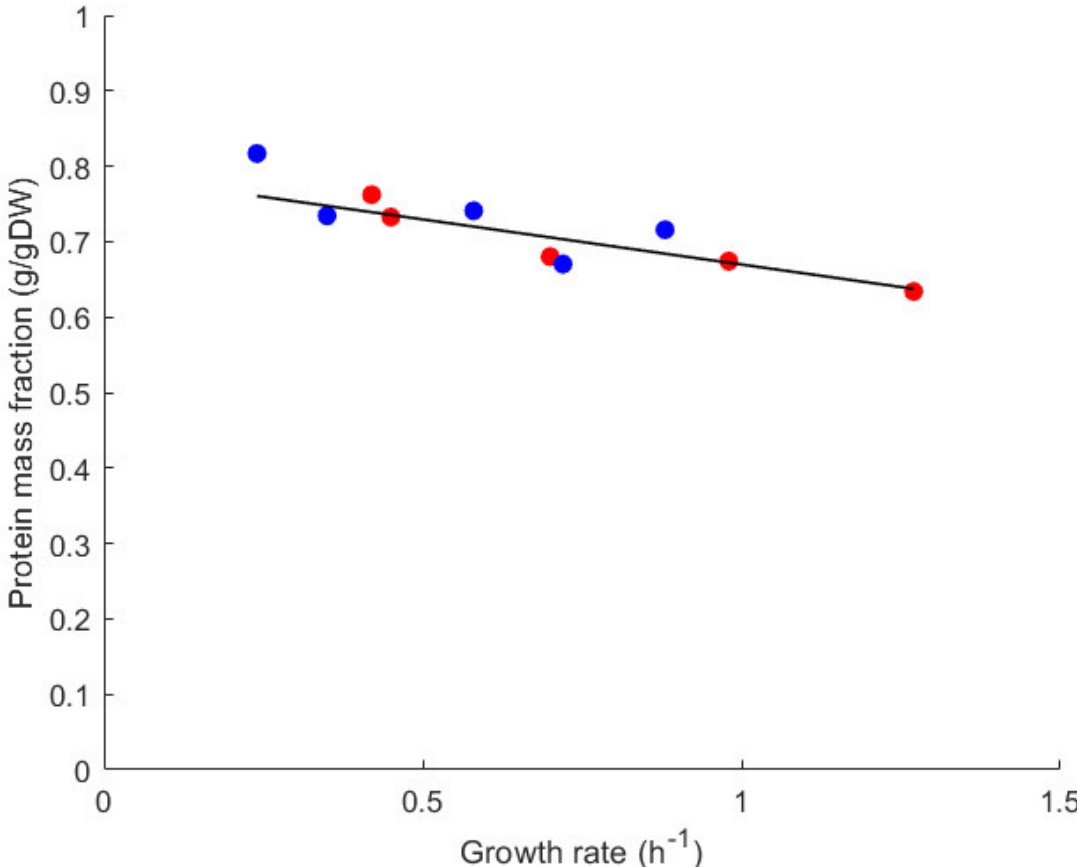

**Appendix 2—figure 1.** Protein dry mass fraction for different growth rates of *E. coli*. The protein dry mass fraction (g gDW$^{-1}$) as a function of the steady-state growth rate was computed from data for the NCM3722 wild-type strain grown in different media (red dots) or for a strain carrying a plasmid for the gratuitous overexpression of a protein (blue dots) (Appendix Table S4 in *Basan et al., 2015b*). We interpolated the data (black line) to provide an estimate of the protein dry mass fractions at the growth rates corresponding to batch growth of the BW25113 strain in minimal medium with either glucose or glycerol ($\mu$ = 0.61 hr$^{-1}$ or $\mu$ = 0.49 hr$^{-1}$, respectively).

**Appendix 2—table 2.** Estimation of the values of the kinetic parameters in the model, in the case of batch growth of *E. coli* in minimal medium with glucose or glycerol, as explained in the text.

| Parameter | Glucose | Glycerol | Unit |
|---|---|---|---|
| $\hat{\rho}_{ru}$ | 1.17 | 1.17 | – |
| $\hat{\rho}_{mef}$ | 1.5 | 1.5 | – |
| $\hat{k}_r$ | 2.9 | 3.6 | hr$^{-1}$ |
| $\hat{k}_{mu}$ | 1.2 | 1.3 | hr$^{-1}$ |
| $\hat{e}_s$ | 18.3 | 11.2 | hr$^{-1}$ |

*Appendix 2—table 2 Continued on next page*

*Appendix 2—table 2 Continued*

| Parameter | Glucose | Glycerol | Unit |
|---|---|---|---|
| $\hat{k}_{mer}$ | 5.0 | 4.4 | $\mathrm{hr}^{-1}$ |
| $\hat{k}_{mef}$ | 17.4 | 6.7 | $\mathrm{hr}^{-1}$ |
| $\hat{k}_a$ | 2279 | 6462 | $\mathrm{hr}^{-1}$ |
| $\hat{K}_r$ | 0.29 | 0.28 | $\mathrm{Cmmol\,gDW}^{-1}$ |
| $\hat{K}_{mu}$ | 0.29 | 0.28 | $\mathrm{Cmmol\,gDW}^{-1}$ |
| $\hat{K}_{mer}$ | 0.29 | 0.28 | $\mathrm{Cmmol\,gDW}^{-1}$ |
| $\hat{K}_{mef}$ | 0.29 | 0.28 | $\mathrm{Cmmol\,gDW}^{-1}$ |
| $\hat{K}_{ar}$ | 0.0009 | 0.0005 | $\mathrm{mmol\,gDW}^{-1}$ |
| $\hat{K}_{amer}$ | 0.0011 | 0.001 | $\mathrm{mmol\,gDW}^{-1}$ |
| $\hat{K}_{amef}$ | 0.0011 | 0.001 | $\mathrm{mmol\,gDW}^{-1}$ |
| $\hat{K}_{amu}$ | 0.0009 | 0.0005 | $\mathrm{mmol\,gDW}^{-1}$ |
| $\hat{n}_{mer}$ | 4.3 | 4.7 | $\mathrm{mmol\,Cmmol}^{-1}$ |
| $\hat{n}_{mef}$ | 2.0 | 2.3 | $\mathrm{mmol\,Cmmol}^{-1}$ |
| $\hat{n}_r$ | 0.77 | 0.09 | $\mathrm{mmol\,Cmmol}^{-1}$ |
| $\hat{n}_{mu}$ | 0.65 | 0.65 | $\mathrm{mmol\,Cmmol}^{-1}$ |

## Estimation of parameter values for batch growth

The model contains 20 kinetic parameters. Estimation of all of these values from the data in *Appendix 2—table 1* would lead to identifiability problems. However, as shown below, making appropriate assumptions based on experimental observations allows all parameters to be unambiguously fixed.

## Parameters in energy balance equation $n_{me}$, $n_{mer}$, $n_{mef}$, $n_r$, $n_{mu}$, $k_a$

We remind that the energy cofactor rate equation at steady state, or energy balance, is given by

$$0 = n_{mer}\,v_{mer} + n_{mef}\,v_{mef} - n_r\,v_r - n_{mu}\,v_{mu} - v_d, \tag{117}$$

where $v_d = k_a\,a^*$.

The ATP yield coefficients $n_{mer}$ and $n_{mef}$ describe how many energy cofactor molecules (ATP) can be regenerated from a molecule of substrate (glucose or glycerol), in units $\mathrm{mmol_{ATP}\,Cmmol}$. *Basan et al., 2015b*, describe a procedure for deriving the yield coefficients $n_{mer}$ and $n_{mef}$ from the reaction stoichiometry of the metabolic pathways used during growth on glucose. Aerobic respiration generates 4 ATP, 8 NADH, 2 NADPH, and 2 $FADH_2$ from one molecule of glucose, equivalent to 26 ATP, whereas aerobic fermentation (acetate overflow) leads to 4 ATP and 4 NADH, equivalent to 12 ATP. As a consequence,

$$\hat{n}_{mer} \quad = 26\,\mathrm{ATP/Glc} = 26/6 = 4.3\,\mathrm{mmol\,Cmmol}^{-1}, \tag{118}$$

$$\hat{n}_{mef} \quad = 12\,\mathrm{ATP/Glc} = 12/6 = 2\,\mathrm{mmol\,Cmmol}^{-1}, \tag{119}$$

bearing in mind that glucose contains 6 C atoms. Restricting central metabolism to the glycolysis and TCA pathways, like *Basan et al., 2015b*, and focusing on the main flux of glycerol catabolism through the lower part of the glycolysis pathway, the ATP yield of glycerol respiration can be determined as 2 ATP, 4 NADH, 1 NADPH, and 2 $FADH_2$, equivalent to 14 ATP. Similarly, for aerobic fermentation we find 2 ATP, 2 NADH, and 1 $FADH_2$, equivalent to 7 ATP. This yields

$$\hat{n}_{mer} \quad = 14 \,\text{ATP/Gly} = 14/3 = 4.7 \,\text{mmol Cmmol}^{-1}, \tag{120}$$

$$\hat{n}_{mef} \quad = 7 \,\text{ATP/Gly} = 7/3 = 2.3 \,\text{mmol Cmmol}^{-1}, \tag{121}$$

given that glycerol contains 3 C atoms.

The coefficient $n_r$ describes the ATP costs of protein synthesis. *Kaleta et al., 2013*, compute the amount of ATP needed for the elongation of a protein by one amino acid, including the net ATP costs of the synthesis of the amino acids from central metabolites and mRNA synthesis. They find that the ATP costs of the synthesis of many amino acids are negative (i.e. their synthesis yields ATP), while the ATP costs of mRNA synthesis are negligible in comparison with the translation costs. For glucose, the median total ATP costs are 3.7 ATP/amino acid. This equals 3.7/4.8=0.77 $\text{mmol}_{\text{ATP}}$ $\text{Cmmol}_{\text{aa}}^{-1}$, where the mean C content of amino acids, weighted for the amino acid composition of biomass, is estimated at 4.8 (data from *Feist et al., 2007*). That is,

$$\hat{n}_r = 0.77 \,\text{mmol Cmmol}^{-1}. \tag{122}$$

These theoretical costs are close to the value of 0.94 $\text{mmol}_{\text{ATP}}$ $\text{Cmmol}_{\text{aa}}^{-1}$ obtained from the review of Russell and Cook, who base their estimate on calculations by Stouthamer (*Russell and Cook, 1995*). (The value of 0.94 $\text{mmol}_{\text{ATP}}$ $\text{Cmmol}_{\text{aa}}^{-1}$ is obtained by converting the value given in Table 1 of *Russell and Cook, 1995*, bearing in mind that the calculations were done for a protein fraction of biomass equal to 0.52 and using a carbon mass fraction of protein equal to 0.5; *Feist et al., 2007*.) For glycerol, where the synthesis of many amino acids is energetically favorable (*Kaleta et al., 2013*), the median total ATP costs are much lower: 0.44 ATP/amino acid. This amounts to 0.44/4.8=0.09 $\text{mmol}_{\text{ATP}}$ $\text{Cmmol}_{\text{aa}}^{-1}$, and hence

$$\hat{n}_r = 0.09 \,\text{mmol Cmmol}^{-1}. \tag{123}$$

The coefficient $n_{mu}$ describes the ATP costs of the synthesis of other macromolecules (RNA, DNA, etc.). From the review of *Russell and Cook, 1995*, under the assumption that the average carbon mass fraction of other macromolecules is also equal to 0.5, we find that these ATP costs equal 0.65 $\text{mmol}_{\text{ATP}}$ $\text{Cmmol}_{\text{macromolecule}}^{-1}$, so that

$$\hat{n}_{mu} = 0.65 \,\text{mmol Cmmol}^{-1}. \tag{124}$$

This value applies to growth on glucose, but in the absence of information specific to growth on glycerol, we use the same value for the latter condition.

It has been well established that the estimated ATP production exceeds the estimated ATP consumption for macromolecular synthesis by a factor of 2–3 in the case of growth on minimal medium with glucose (*Feist et al., 2007*; *Russell and Cook, 1995*). This suggests a dissipation of energy which is also observed in our case: the ratio of $\hat{n}_{mer} \hat{v}_{mer} + \hat{n}_{mef} \hat{v}_{mef}$ and $\hat{n}_r \hat{v}_r + \hat{n}_{mu} \hat{v}_{mu}$ equals 2.1 in the case of glucose, and increases to 7.5 in the case of glycerol. The difference is due to the costs of osmoregulation, motility, and other maintenance processes (*van Bodegom, 2007*), but also to energy spilling, a factor that remains little understood (*Russell and Cook, 1995*). As explained in Appendix 1, we model all of the above forms of energy dissipation by a first-order reaction with constant $k_a$ whose value can be computed by closing the energy balance (*Equation 117*):

$$\hat{k}_a = \frac{\hat{n}_{mer} \hat{v}_{mer} + \hat{n}_{mef} \hat{v}_{mef} - \hat{n}_r \hat{v}_r - \hat{n}_{mu} \hat{v}_{mu}}{\hat{a}^*}. \tag{125}$$

In the case of batch growth on glucose, we thus find an approximate value

$$\hat{k}_a = 2279 \,\text{hr}^{-1}, \tag{126}$$

and for glycerol,

$$\hat{k}_a = 6426 \,\text{hr}^{-1}. \tag{127}$$

## Parameter in rate equation for central carbon metabolism $e_s$

As explained in Appendix 1, the macroreaction for central carbon metabolism simplifies to the following simple rate equation:

$$v_{mc} = e_s \, m_c. \tag{128}$$

With the value for $m_c$ derived in the previous section (**Appendix 2—table 1**), we obtain the following estimates for glucose:

$$\hat{e}_s = 18.3 \ \mathrm{hr}^{-1}, \tag{129}$$

and for glycerol:

$$\hat{e}_s = 11.2 \ \mathrm{hr}^{-1}. \tag{130}$$

## Parameters in the rate equations for the synthesis of proteins and other biomass components $K_r$, $K_{mu}$, $K_{ar}$, $K_{amu}$, $k_r$, and $k_{mu}$

The rate equations for the macroreactions corresponding to protein synthesis and the synthesis of other macromolecules are restated as a reminder:

$$v_r \quad = k_r \, r \, \frac{a^*}{a^* + K_{ar}} \, \frac{c}{c + K_r}, \tag{131}$$

$$v_{mu} \quad = k_{mu} \, m_u \, \frac{a^*}{a^* + K_{amu}} \, \frac{c}{c + K_{mu}}. \tag{132}$$

The above reactions consume central metabolites ($c$) and charged energy cofactors (ATP) ($a^*$).

Very little information is available on the in vivo values of half-saturation constants occurring in the kinetic expressions of the macroreactions. However, previous metabolomics assays have yielded general observations on enzyme saturation (the ratio of reaction substrates and half-saturation constants) that will be exploited here (**Bennett et al., 2009**). These will be refined by combining available measurements with a recent compilation of $K_m$ values for *E. coli* (**Dourado et al., 2021**; **Park et al., 2016**).

First, in the case of central carbon metabolism, 'substrate concentrations are close to $K_m$ for many reactions' (**Bennett et al., 2009**). We have computed, for metabolites in central carbon metabolism of *E. coli* quantified by **Gerosa et al., 2015**, the ratio of metabolite concentrations and values of the half-saturation constants of the reactions in which the metabolites participate (**Dourado et al., 2021**). Taking the geometric mean of the ratios, we found an average value of substrate saturation of 1.2 for glucose and 0.72 for glycerol (**Supplementary file 3**). Assuming that this value is approximately valid for all reactions consuming central carbon metabolites in our model, we estimate for glucose

$$\hat{K}_r = \hat{K}_{mu} \approx \frac{\hat{c}}{1.2} = 0.29 \ \mathrm{Cmmol \ gDW^{-1}}, \tag{133}$$

and for glycerol

$$\hat{K}_r = \hat{K}_{mu} \approx \frac{\hat{c}}{0.72} = 0.28 \ \mathrm{Cmmol \ gDW^{-1}}, \tag{134}$$

Note that we deal with apparent half-saturation constants that account for possible metabolic regulation.

Second, ATP and NAD+ were found to saturate their enzymes with 'cofactor concentration typically exceeding their $K_m$ value by more than 10-fold' (**Bennett et al., 2009**). This motivates the following approximate values for the half-saturation constants occurring in the energy terms of the biosynthesis rate equations:

$$\hat{K}_{ar} = \hat{K}_{amu} \approx \hat{a}^*/10 \ \mathrm{mmol \ gDW^{-1}}, \tag{135}$$

with different values for growth on glucose and glycerol (0.0009 vs 0.0005 mmol gDW⁻¹).

Together with the values for the fluxes and enzyme concentrations, we can now derive values for the unknown catalytic constants $k_r$ and $k_{mu}$ from **Equations 131 and 132**. In the case of growth on glucose, we have

$$\hat{k}_r = 2.9 \text{ hr}^{-1}, \quad \hat{k}_{mu} = 1.2 \text{ hr}^{-1}, \tag{136}$$

whereas for growth on glycerol we find

$$\hat{k}_r = 3.6 \text{ hr}^{-1}, \quad \hat{k}_{mu} = 1.3 \text{ hr}^{-1}. \tag{137}$$

Note that the estimates for $k_r$ are comparable to values used for the maximum translation capacity in previous work (5.9 hr[-1] in **Scott et al., 2010**; 3.6 hr[-1] in **Giordano et al., 2016**).

## Parameters in the rate equations for energy metabolism $K_{mer}$, $K_{mef}$, $K_{amer}$, $K_{amef}$, $k_{mer}$, and $k_{mef}$

We repeat the rate equations for energy metabolism, for the two macroreactions (respiration and fermentation):

$$v_{mer} = k_{mer} m_{er} \frac{a_0 - a^*}{a_0 - a^* + K_{amer}} \frac{c}{c + K_{mer}}, \tag{138}$$

$$v_{mef} = k_{mef} m_{ef} \frac{a_0 - a^*}{a_0 - a^* + K_{amef}} \frac{c}{c + K_{mef}}. \tag{139}$$

The arguments given in the previous section for fixing the values of the half-saturation constants also apply in this case, so that we obtain

$$\hat{K}_{mer} = \hat{K}_{mef} = 0.29 \text{ Cmmol gDW}^{-1}, \tag{140}$$

$$\hat{K}_{amer} = \hat{K}_{amef} = 0.0011 \text{ mmol gDW}^{-1}, \tag{141}$$

for growth on glucose, and

$$\hat{K}_{mer} = \hat{K}_{mef} = 0.28 \text{ Cmmol gDW}^{-1}, \tag{142}$$

$$\hat{K}_{amer} = \hat{K}_{amef} = 0.001 \text{ mmol gDW}^{-1}, \tag{143}$$

for growth on glycerol.

In the previous section, we were only able to reconstruct the total concentration of enzymes involved in energy metabolism (**Appendix 2—table 1**), but not the fractions involved in aerobic respiration or fermentation. Let $\hat{m}_e = \hat{m}_{er} + \hat{m}_{ef}$. In order to derive the concentrations $m_{er}$ and $m_{ef}$, we follow approximately the same procedure as **Basan et al., 2015b**, but for the proteomics data of **Schmidt et al., 2016**. We divide the proteins labeled as taking part in energy metabolism into enzymes only playing a role in respiration (pyruvate decarboxylation, TCA cycle), enzymes only playing a role in fermentation (acetate pathway), and other enzymes, notably those constituting the electron transport chain and ATP synthases using the proton gradient for ATP production. The latter category is involved in both (aerobic) respiration and fermentation, and we divide the protein mass according to the ratio of the respiration and fermentation fluxes. For growth on glucose, we find fractions 0.45, 0.01, and 0.54 for the three protein categories, whereas for glycerol we find 0.37, 0.01, and 0.62, respectively (**Supplementary file 4**). This gives rise to the following estimates for glucose,

$$\hat{m}_{er} = (0.45 + 0.54 \frac{\hat{v}_{mer}}{\hat{v}_{mer} + \hat{v}_{mef}}) \hat{m}_e = 1.9 \text{ Cmmol gDW}^{-1}, \tag{144}$$

$$\hat{m}_{ef} = (0.01 + 0.54 \frac{\hat{v}_{mef}}{\hat{v}_{mer} + \hat{v}_{mef}}) \hat{m}_e = 1.1 \text{ Cmmol gDW}^{-1}, \tag{145}$$

and for glycerol

$$\hat{m}_{er} = (0.37 + 0.62 \frac{\hat{v}_{mer}}{\hat{v}_{mer} + \hat{v}_{mef}})\, \hat{m}_e = 4.4 \text{ Cmmol gDW}^{-1},$$

(146)

$$\hat{m}_{ef} = (0.01 + 0.62 \frac{\hat{v}_{mef}}{\hat{v}_{mer} + \hat{v}_{mef}})\, \hat{m}_e = 0.47 \text{ Cmmol gDW}^{-1}.$$

(147)

Together with the values for the fluxes and metabolite concentrations, we can now estimate values for the unknown apparent catalytic constants $k_{mer}$ and $k_{mef}$ from *Equations 138 and 139*. In the case of growth on glucose, we have

$$\hat{k}_{mer} = 5.0 \text{ hr}^{-1}, \quad \hat{k}_{mef} = 17.4 \text{ hr}^{-1},$$

(148)

and for growth on glycerol,

$$\hat{k}_{mer} = 4.4 \text{ hr}^{-1}, \quad \hat{k}_{mef} = 6.7 \text{ hr}^{-1}.$$

(149)

All parameter values derived in this and the previous sections are summarized in *Appendix 2—table 2*.

## Data and parameter estimates for continuous growth

The model calibration procedure for the other conditions considered, continuous growth in a chemostat, in minimal medium with glucose at dilution rates of 0.2 hr$^{-1}$, 0.35 hr$^{-1}$, and 0.5 hr$^{-1}$, is the same as for batch growth. Not all source data used above are available for continuous growth. In their absence, we use the corresponding data for batch growth as a proxy. In particular, total protein and metabolite concentrations were obtained from *Gerosa et al., 2015*, and *Basan et al., 2015b*, by selecting the (interpolated) values for batch growth at rates corresponding to the dilution rates (*Appendix 2—figure 1*). In addition, for the case of growth at a dilution rate of 0.2 hr$^{-1}$, where no significant acetate overflow is detected, we set the acetate secretion rate to 5% of the acetate secretion rate during continuous growth at 0.35 hr$^{-1}$, that is, a value below the detection limit. This allows the same model with respiration and fermentation to be used over all conditions.

The data used for calibration is shown in *Appendix 2—table 3* and the values for the parameters obtained after calibration are listed in *Appendix 2—table 4*.

**Appendix 2—table 3.** Reconstruction of growth and degradation rates, fluxes, and concentrations from published datasets for the case of continuous growth of *E. coli* in minimal medium with glucose at different dilution rates (D0.2: 0.2 hr$^{-1}$, D0.35: 0.35 hr$^{-1}$, D0.5: 0.5 hr$^{-1}$).
For the error bars, see *Appendix 2—table 1*. References: [a] *Peebo et al., 2015*, [b] *Esquerré et al., 2014*, [c] *Morin et al., 2016*, [d] *Basan et al., 2015b*, [e] *Schmidt et al., 2016*, [f] *Gerosa et al., 2015*, [g] *Park et al., 2016*.

| Rates | Unit | D0.2 | D0.35 | D0.5 | Reference |
|---|---|---|---|---|---|
| $\hat{\mu}$ | hr$^{-1}$ | o.2 | 0.35 | 0.5 | *a* |
| $\hat{\gamma}$ | hr$^{-1}$ | 0.027 | 0.027 | 0.027 | *b* |
| Uptake, secretion, and biosynthesis fluxes | | | | | |
| $\hat{v}_{mc}$ | Cmmol gDW$^{-1}$ hr$^{-1}$ | 16.0 | 26.2 | 37.4 | *a* |
| $\hat{v}_{mer}$ | Cmmol gDW$^{-1}$ hr$^{-1}$ | 5.3 | 8.1 | 9.4 | Derived |
| $\hat{v}_{mef}$ | Cmmol gDW$^{-1}$ hr$^{-1}$ | 0.02 | 0.16 | 2.0 | *a* |
| $\hat{v}_{mu}$ | Cmmol gDW$^{-1}$ hr$^{-1}$ | 1.9 | 3.4 | 5.2 | Derived |

*Appendix 2—table 3 Continued on next page*

*Appendix 2—table 3 Continued*

| Rates | Unit | D0.2 | D0.35 | D0.5 | Reference |
|---|---|---|---|---|---|
| $\hat{v}_r$ | Cmmol gDW$^{-1}$ hr$^{-1}$ | 7.3 | 11.8 | 16.1 | Derived |
| Total biomass concentration | | | | | |
| $1/\hat{\beta}$ | Cmmol gDW$^{-1}$ | $40.65 \pm 2.0$ | $40.65 \pm 2.0$ | $40.65 \pm 2.0$ | $c$ |
| Protein concentrations | | | | | |
| $\hat{m}_u$ | Cmmol gDW$^{-1}$ | $11.2 \pm 0.6$ | $11.2 \pm 0.6$ | $10.4 \pm 0.5$ | $c, d, e$ |
| $\hat{r}$ | Cmmol gDW$^{-1}$ | $9.3 \pm 0.5$ | $9.4 \pm 0.5$ | $11.0 \pm 0.5$ | $c, d, e$ |
| $\hat{m}_c$ | Cmmol gDW$^{-1}$ | $3.5 \pm 0.2$ | $3.4 \pm 0.2$ | $3.3 \pm 0.2$ | $c, d, e$ |
| $\hat{m}_{er} + \hat{m}_{ef}$ | Cmmol gDW$^{-1}$ | $8.0 \pm 0.4$ | $7.2 \pm 0.4$ | $5.8 \pm 0.3$ | $c, d, e$ |
| $\hat{m}_{er}$ | Cmmol gDW$^{-1}$ | 7.9 | 7.1 | 5.2 | Derived |
| $\hat{m}_{ef}$ | Cmmol gDW$^{-1}$ | 0.05 | 0.1 | 0.6 | Derived |
| Metabolite concentrations | | | | | |
| $\hat{c}$ | Cmmol gDW$^{-1}$ | $0.35 \pm 0.002$ | $0.35 \pm 0.002$ | $0.35 \pm 0.002$ | $f, g$ |
| $\hat{a}^*$ | mmol gDW$^{-1}$ | 0.005 | 0.006 | 0.008 | $f$ |
| $\hat{a}$ | mmol gDW$^{-1}$ | 0.011 | 0.015 | 0.016 | $f$ |
| $\hat{a}_0$ | mmol gDW$^{-1}$ | 0.016 | 0.021 | 0.024 | $f$ |
| Concentration of other biomass | | | | | |
| $\hat{u}$ | Cmmol gDW$^{-1}$ | 8.2 | 9.0 | 9.8 | Derived |

**Appendix 2—table 4.** Estimation of the values of the kinetic parameters in the model, in the case of continuous growth of *E. coli* in minimal medium with glucose at different dilution rates (D0.2: 0.2 hr$^{-1}$, D0.35: 0.35 hr$^{-1}$, D0.5: 0.5 hr$^{-1}$), as explained in the text.

| Parameter | D0.2 | D0.35 | D0.5 | Unit |
|---|---|---|---|---|
| $\hat{\rho}_{ru}$ | 1.16 | 1.17 | 1.17 | – |
| $\hat{\rho}_{mef}$ | 1.5 | 1.5 | 1.5 | – |
| $\hat{k}_r$ | 1.6 | 2.5 | 2.9 | hr$^{-1}$ |
| $\hat{k}_{mu}$ | 0.33 | 0.61 | 1.0 | hr$^{-1}$ |
| $\hat{e}_s$ | 4.5 | 7.6 | 11.2 | hr$^{-1}$ |
| $\hat{k}_{mer}$ | 1.3 | 2.3 | 3.6 | hr$^{-1}$ |
| $\hat{k}_{mef}$ | 0.77 | 2.98 | 6.8 | hr$^{-1}$ |
| $\hat{k}_a$ | 3203 | 4001 | 3633 | hr$^{-1}$ |
| $\hat{K}_r$ | 0.29 | 0.29 | 0.29 | Cmmol gDW$^{-1}$ |
| $\hat{K}_{mu}$ | 0.29 | 0.29 | 0.29 | Cmmol gDW$^{-1}$ |
| $\hat{K}_{mer}$ | 0.29 | 0.29 | 0.29 | Cmmol gDW$^{-1}$ |

*Appendix 2—table 4 Continued on next page*

Appendix 2—table 4 Continued

| Parameter | D0.2 | D0.35 | D0.5 | Unit |
|---|---|---|---|---|
| $\hat{K}_{mef}$ | 0.29 | 0.29 | 0.29 | $\mathrm{Cmmol\,gDW}^{-1}$ |
| $\hat{K}_{ar}$ | 0.0005 | 0.0006 | 0.0008 | $\mathrm{mmol\,gDW}^{-1}$ |
| $\hat{K}_{amer}$ | 0.0011 | 0.0015 | 0.0016 | $\mathrm{mmol\,gDW}^{-1}$ |
| $\hat{K}_{amef}$ | 0.0011 | 0.0015 | 0.0016 | $\mathrm{mmol\,gDW}^{-1}$ |
| $\hat{K}_{amu}$ | 0.0005 | 0.0006 | 0.0008 | $\mathrm{mmol\,gDW}^{-1}$ |
| $\hat{n}_{mer}$ | 4.3 | 4.3 | 4.3 | $\mathrm{mmol\,Cmmol}^{-1}$ |
| $\hat{n}_{mef}$ | 2.0 | 2.0 | 2.0 | $\mathrm{mmol\,Cmmol}^{-1}$ |
| $\hat{n}_{r}$ | 0.77 | 0.77 | 0.77 | $\mathrm{mmol\,Cmmol}^{-1}$ |
| $\hat{n}_{mu}$ | 0.65 | 0.65 | 0.65 | $\mathrm{mmol\,Cmmol}^{-1}$ |

## Data and parameter estimates for MG1655 and NCM3722 strains

In order to test the robustness of our results with respect to the calibration procedure, we calibrated the model for a different *E. coli* strain, MG1655, in the same way as for the reference strain. To this aim, we used published measurements on batch growth of MG1655 in minimal medium with glucose, including metabolite concentrations (*McCloskey et al., 2018*), proteomics data (*Schmidt et al., 2016*), and metabolic fluxes (*Monk et al., 2017*).

The total biomass concentration is the same as for the reference strain (*Equation 81*). The total metabolite concentration is obtained by *McCloskey et al., 2018*, who reported a value of 3.7 Cmmol gDW$^{-1}$, equivalent to 9.1% of the total cellular biomass. The fraction of central metabolites is estimated to be 14% of the total metabolic concentration. The total protein concentration is obtained from *Basan et al., 2015b*, who report a protein fraction of 0.71 for the MG1655 strain, to which we add the fraction of free amino acids, estimated as 50% of the total metabolite concentration (*Bennett et al., 2009*). This gives a total protein biomass fraction of 0.76.

Proteins are then distributed over our protein categories, following the mass fraction values reported by *Schmidt et al., 2016*, for the MG1655 strain. Accordingly, we estimate

$$\hat{m}_u \quad = 0.37 \cdot 0.76 \cdot 1/\hat{\beta} = 11.4 \ \mathrm{Cmmol\,gDW}^{-1}, \tag{150}$$

$$\hat{r} \quad = 0.45 \cdot 0.76 \cdot 1/\hat{\beta} = 13.8 \ \mathrm{Cmmol\,gDW}^{-1}, \tag{151}$$

$$\hat{m}_c \quad = 0.08 \cdot 0.76 \cdot 1/\hat{\beta} = 2.4 \ \mathrm{Cmmol\,gDW}^{-1}, \tag{152}$$

$$\hat{m}_{er} + \hat{m}_{ef} \quad = 0.10 \cdot 0.76 \cdot 1/\hat{\beta} = 3.1 \ \mathrm{Cmmol\,gDW}^{-1}. \tag{153}$$

Uptake and secretion rates were taken from *Monk et al., 2017*. Comparison of metabolite concentration measurements of *McCloskey et al., 2018*, with $K_m$ values collected by *Dourado et al., 2021*, shows that reactions in central carbon metabolism are more saturated in MG1655 than in the reference strain (2.2 vs 1.2), in agreement with its higher growth rate (*Supplementary file 3*). Accordingly, the half-saturation constant of reactions consuming central metabolites are estimated as

$$\hat{K}_r = \hat{K}_{mu} = \hat{K}_{mer} = \hat{K}_{mef} \approx \tfrac{\hat{c}}{2.2} = 0.24 \ \mathrm{Cmmol\,gDW}^{-1}. \tag{154}$$

The data used for calibration are summarized in *Appendix 2—table 5* and the values for the parameters obtained after calibration are listed in *Appendix 2—table 6*.

**Appendix 2—table 5.** Reconstruction of growth and degradation rates, uptake and secretion fluxes, and protein and metabolite concentrations from published datasets for *E. coli* MG1655 and NCM3722 strains for the case of batch growth in glucose minimal medium.

The uncertainty intervals for the rates, fluxes, and metabolite concentrations are standard deviations reported in the source publications, after unit conversion. For the NCM3722 strain, as an example of a fast-growing strain with a higher growth yield than the BW25113 reference strain, we only use a subset of observed values in the main text. References: [a]*Cheng et al., 2019*, [b]*Basan et al., 2015a*, [c]*Esquerré et al., 2014*, [d]*Farmer and Jones, 1976*, [e]*Monk et al., 2017*, [f]*Basan et al., 2015b*, [g]*Schmidt et al., 2016*, [h]*Park et al., 2016*, [i]*McCloskey et al., 2018*.

| Rates | Unit | MG1655 | NCM3722 | Reference |
|---|---|---|---|---|
| $\hat{\mu}$ | $hr^{-1}$ | $0.69 \pm 0.02$ | $0.97 \pm 0.05$ | $a, b$ |
| $\hat{\gamma}$ | $hr^{-1}$ | $0.027$ | – | $c, d$ |
| Uptake, secretion, and biosynthesis fluxes | | | | |
| $\hat{v}_{mc}$ | $Cmmol\,gDW^{-1}\,hr^{-1}$ | $51.5 \pm 8.5$ | $66.1 \pm 4$ | $a, b, e$ |
| $\hat{v}_{mer}$ | $Cmmol\,gDW^{-1}\,hr^{-1}$ | $5.7$ | – | Derived |
| $\hat{v}_{mef}$ | $Cmmol\,gDW^{-1}\,hr^{-1}$ | $7.8 \pm 2.3$ | $10.3 \pm 1.8$ | $a, b, e$ |
| $\hat{v}_{mu}$ | $Cmmol\,gDW^{-1}\,hr^{-1}$ | $7.0$ | – | Derived |
| $\hat{v}_{r}$ | $Cmmol\,gDW^{-1}\,hr^{-1}$ | $21.7$ | – | Derived |
| Total biomass concentration | | | | |
| $1/\hat{\beta}$ | $Cmmol\,gDW^{-1}$ | $40.65 \pm 2.0$ | – | $e$ |
| Protein concentrations | | | | |
| $\hat{p}$ | $Cmmol\,gDW^{-1}$ | $30.7 \pm 2.0$ | $29.7 \pm 1.9$ | $e, f$ |
| $\hat{m}_{u}$ | $Cmmol\,gDW^{-1}$ | $11.4 \pm 0.74$ | – | $e, f, g$ |
| $\hat{r}$ | $Cmmol\,gDW^{-1}$ | $13.8 \pm 0.9$ | – | $e, f, g$ |
| $\hat{m}_{c}$ | $Cmmol\,gDW^{-1}$ | $2.4 \pm 0.2$ | – | $e, f, g$ |
| $\hat{m}_{er} + \hat{m}_{ef}$ | $Cmmol\,gDW^{-1}$ | $3.1 \pm 0.2$ | – | $e, f, g$ |
| $\hat{m}_{er}$ | $Cmmol\,gDW^{-1}$ | $2.2 \pm 0.1$ | – | Derived |
| $\hat{m}_{ef}$ | $Cmmol\,gDW^{-1}$ | $0.9 \pm 0.04$ | – | Derived |
| Metabolite concentrations | | | | |
| $\hat{c}$ | $Cmmol\,gDW^{-1}$ | $0.5 \pm 0.09$ | $0.8 \pm 0.03$ | $h, i$ |
| $\hat{a}^*$ | $mmol\,gDW^{-1}$ | $0.046$ | – | $h, i$ |
| $\hat{a}$ | $mmol\,gDW^{-1}$ | $0.008$ | – | $h, i$ |
| $\hat{a}_0$ | $mmol\,gDW^{-1}$ | $0.054$ | – | $h, i$ |
| Concentration of other biomass | | | | |
| $\hat{u}$ | $Cmmol\,gDW^{-1}$ | $9.4$ | – | Derived |

**Appendix 2—table 6.** Estimation of the values of the kinetic parameters in the model for the *E. coli* MG1655 strain during batch growth in glucose minimal medium from data in Appendix 2—table 5, as explained in the text.
Idem for a model variant with an additional category of growth-rate-independent proteins (*Q*), using data for the BW25113 strain from *Appendix 2—table 1*.

| Parameter | MG1655 | Model variant with Q | Unit |
|---|---|---|---|
| $\hat{\rho}_{ru}$ | 1.17 | 1.17 | – |
| $\hat{\rho}_{mef}$ | 1.5 | 1.5 | – |
| $\hat{k}_r$ | 2.5 | 6.1 | $hr^{-1}$ |
| $\hat{k}_{mu}$ | 0.9 | 2.5 | $hr^{-1}$ |
| $\hat{e}_s$ | 21.0 | 38.0 | $hr^{-1}$ |
| $\hat{k}_{mer}$ | 4.1 | 10.3 | $hr^{-1}$ |
| $\hat{k}_{mef}$ | 20.4 | 36.1 | $hr^{-1}$ |
| $\hat{k}_a$ | 412 | 2278 | $hr^{-1}$ |
| $\hat{K}_r$ | 0.24 | 0.29 | $Cmmol\,gDW^{-1}$ |
| $\hat{K}_{mu}$ | 0.24 | 0.29 | $Cmmol\,gDW^{-1}$ |
| $\hat{K}_{mer}$ | 0.24 | 0.29 | $Cmmol\,gDW^{-1}$ |
| $\hat{K}_{mef}$ | 0.24 | 0.29 | $Cmmol\,gDW^{-1}$ |
| $\hat{K}_{ar}$ | 0.005 | 0.0009 | $mmol\,gDW^{-1}$ |
| $\hat{K}_{amer}$ | 0.0008 | 0.0011 | $mmol\,gDW^{-1}$ |
| $\hat{K}_{amef}$ | 0.0008 | 0.0011 | $mmol\,gDW^{-1}$ |
| $\hat{K}_{amu}$ | 0.005 | 0.0009 | $mmol\,gDW^{-1}$ |
| $\hat{n}_{mer}$ | 4.3 | 4.3 | $mmol\,Cmmol^{-1}$ |
| $\hat{n}_{mef}$ | 2.0 | 2.0 | $mmol\,Cmmol^{-1}$ |
| $\hat{n}_r$ | 0.77 | 0.77 | $mmol\,Cmmol^{-1}$ |
| $\hat{n}_{mu}$ | 0.65 | 0.65 | $mmol\,Cmmol^{-1}$ |

We also collect in *Appendix 2—table 5* the data for batch growth of the NCM3722 strain in minimal medium with glucose, used in the Results section of the main paper. The data concern the growth rate and growth yield (*Cheng et al., 2019*), the glucose uptake, and acetate secretion rates reported by *Cheng et al., 2019*, from experiments carried out by *Basan et al., 2015a*, the total protein concentration (*Basan et al., 2015a*), and the total metabolite concentration (*Park et al., 2016*).

## Calibration of model variant with an additional growth-rate-independent protein category

In Appendix 1, we introduced a model variant with an additional growth-rate-independent protein category, referred to as *Q* (*Scott et al., 2010*). Estimation of the parameters for this model variant requires the estimation, for every protein category, of the offset of the linear relation between growth rate and proteome fraction (*Hui et al., 2015*). In order to obtain results comparable to those for the reference model, we have used proteomics data for the BW25113 strain (*Schmidt et al., 2016*). We considered 22 different growth conditions, excluding stationary phase (no balanced growth) and LB medium (addition of amino acids).

For the $R$ category, the proteome fraction increases with the growth rate and the offset can be computed as $\chi_r^0 = 0.23$ (**Appendix 2—figure 2**). Unfortunately, in the case of $M_c$, $M_e$, and $M_u$, the data show a decreasing or constant pattern with growth rate, which makes it impossible to determine the offset fraction for these protein categories (**Appendix 2—figure 2**, panels B–D). We therefore followed a different approach to estimate the growth-rate-independent protein fraction. Assuming a total fraction of growth-rate-independent proteins $\chi_q = 0.52$, as reported for the MG1655 strain by **Mori et al., 2016**, we split the fraction $\chi_q - \chi_r^0 = 0.29$ over the $M_c$, $M_u$, and $M_e$ categories proportionally to their size:

$$\chi_u^0 = 0.29 \cdot \frac{0.37}{0.56} = 0.19,$$
$$\chi_c^0 = 0.29 \cdot \frac{0.09}{0.56} = 0.05,$$
$$\chi_e^0 = 0.29 \cdot \frac{0.10}{0.56} = 0.05.$$

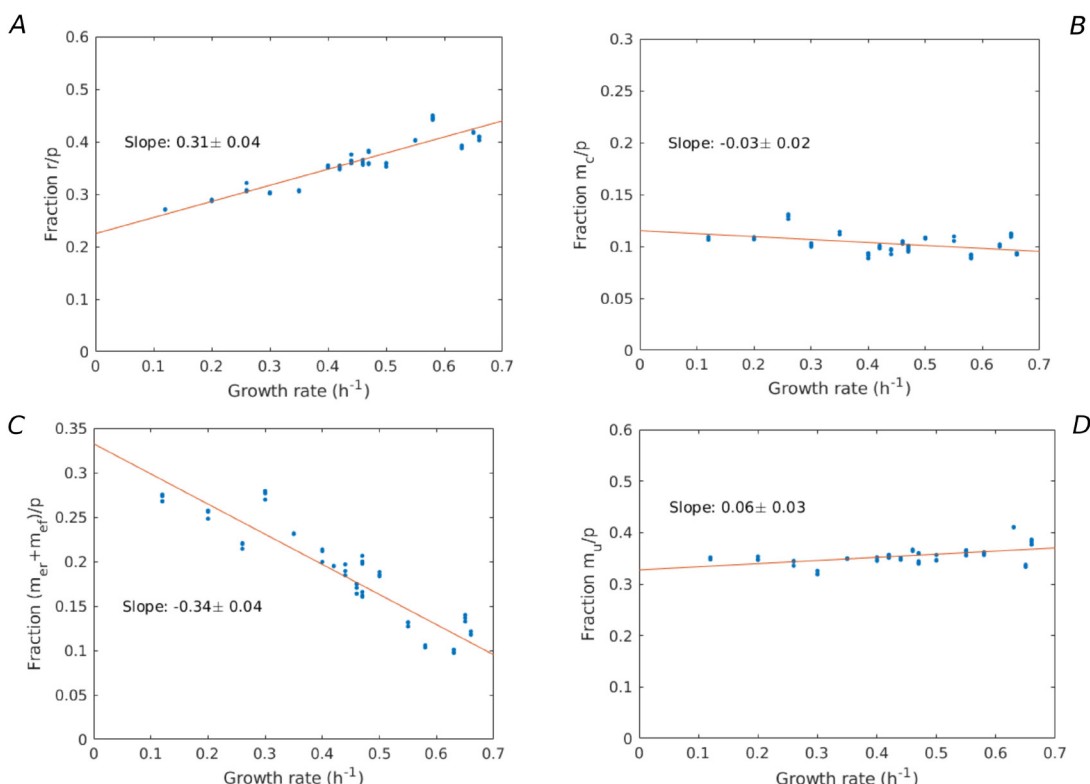

**Appendix 2—figure 2.** Growth-rate dependency of proteome fractions. Using the data from **Schmidt et al., 2016**, the proteome fractions over a large variety of growth conditions (growth on different limiting carbon sources, different temperatures, different pH, etc.) are plotted for the categories: (**A**) ribosomes and translation-affiliated proteins ($r/p$), (**B**) enzymes in central carbon metabolism ($m_c/p$), (**C**) enzymes in energy metabolism (($m_{er} + m_{ef}$)/$p$), and (**D**) other proteins ($m_u/p$). A linear regression is performed, giving rise to slopes (**A**) 0.31 ± 0.04, (**B**) –0.03 ± 0.02, (**C**) –0.34 ± 0.04, and (**D**) 0.06 ± 0.03, showing that only the fraction $r/p$ significantly increases with the growth rate.

Notice that the above partitioning is equivalent to assuming that all enzyme categories have the same proportion of growth-rate-independent proteins.

The growth-rate-dependent fractions of the protein categories are then simply obtained from the difference between the total proteome fractions (**Schmidt et al., 2016**) and the growth-rate-independent fractions:

$$\chi_u = 0.37 - 0.19 = 0.18,$$
$$\chi_r = 0.44 - 0.23 = 0.21,$$
$$\chi_c = 0.09 - 0.05 = 0.04,$$
$$\chi_e = 0.10 - 0.05 = 0.05.$$

Further calibration of the model is then identical to the calibration of the reference model, using published data for batch growth of BW25113 in glucose minimal medium (*Appendix 2—table 1*). In particular, from the total biomass concentration (40.65 Cmmol gDW$^{-1}$) and the protein mass fraction (0.74), we can estimate the following growth-rate-dependent protein concentrations:

$$\hat{q} = 0.52 \cdot 0.74 \cdot 1/\hat{\beta} = 15.9 \text{ Cmmol}, \text{gDW}^{-1}, \tag{155}$$

$$\hat{r} = 0.21 \cdot 0.74 \cdot 1/\hat{\beta} = 6.3 \text{ Cmmol gDW}^{-1}, \tag{156}$$

$$\hat{m}_u = 0.18 \cdot 0.74 \cdot 1/\hat{\beta} = 5.4 \text{ Cmmol gDW}^{-1}, \tag{157}$$

$$\hat{m}_c = 0.04 \cdot 0.74 \cdot 1/\hat{\beta} = 1.2 \text{ Cmmol gDW}^{-1}, \tag{158}$$

$$\hat{m}_{er} + \hat{m}_{ef} = 0.05 \cdot 0.74 \cdot 1/\hat{\beta} = 1.5 \text{ Cmmol gDW}^{-1}. \tag{159}$$

Parameter values derived for this model are summarized in *Appendix 2—table 6*.

