## [Editor Report]

This study develops a rigorous resource allocation model for *E. coli* growing under steady-state conditions. Validated by comparison with a compiled data set, the model highlights the complex nature of the relationship between metabolites, growth rate, and yield which is significantly more complex than the one-to-one-one relationship that has generally been assumed. The work will be of interest not only to investigators interested in basic questions of bacterial physiology but also to those working on applied problems in biotechnology.

---

## [Decision Letter]

**Decision letter after peer review:**

Thank you for submitting your article "Resource allocation accounts for the large variability of rate-yield phenotypes across bacterial strains" for consideration by *eLife*. Our sincere apologies for the delay in returning the decision.

Your article has been reviewed by 2 peer reviewers, and the evaluation has been overseen by a Reviewing Editor and Naama Barkai as the Senior Editor. The reviewers have opted to remain anonymous.

Essential revisions:

While other reviewer concerns as detailed below should be considered, the authors need to address different assumptions for ϕq in order for the study to be complete.

*Reviewer #1 (Recommendations for the authors):*

Proposed points of improvement.

Rationalize bounds. In Figure 2 I would like to better understand what are the reasons for the bounds. What gives the "P" shape? What are the trends with different allocation trade-offs? etc. Possibly some analytical insight is possible here [maybe with a simplified version of the model], leading to more transparent theoretical insight.

Rationalize (and show) trends. Figure 3 seems particularly uninformative. It would be much more instructive to see trends of uptake and secretion rates vs the other variables as 2D plots, compared with the model predictions (particularly for panels AB, panels CD show a complex trend, but this is precisely what I would like to get more insight on). It is not clear how the prediction of Figure 3C is produced by the model since the parameters are not fixed as the allocation changes.

Figure 4. I got lost with this figure, which was particularly uninformative to me (and graphically lacks proper labeling). Looking at the plots, I only see different degrees of agreement between the model and data. Reading the connected Results paragraph, there are a lot of qualitative considerations "under the hood" that seem very interesting but are not accessible/transparent. This could be my own limitation, and it's possible that this paragraph is accessible to a different audience (e.g. more expert than me on metabolic models). However, my impression was that this paragraph/figure could be made more accessible, although I did not gain enough access to give specific recommendations, other than giving the reader some insight on the model predicted trends that we are discussing here.

Where are the optima? In Figure 2 one can explore what the model gives if one tries to optimize (1) growth rate at fixed yield (2) yield at fixed growth rate or (3) looks at Pareto optima of both. I agree that optimization of one or both quantities may not be the goal, but still, it is important to understand where optimization would bring theoretically, and how the data points cluster with respect to these theoretical optima.

Comparison with other frameworks.

A more detailed comparison with other "reference" frameworks would be useful here.

I would propose: Erickson 2017, Basan 2015, Maitra 2015 [but other choices are possible]

[see below]

The definition of yield should be explained much more clearly in the main text, both in the model and in the data. Model: Explain why Equation 2 represents the fraction of carbon going into biomass. Data: explain how the quantity is measured and how the measured quantity relates to the model.

I am confused by some sort of implicit identification that the authors make between allocation (e.g. the fraction of ribosome making ribosomes) and partitioning (e.g. the fraction of proteins or total mass that is ribosomes). In particular, for ribosomes, I am not sure that their equations (e.g. Eq (6) in SI regarding ribosomes) are equivalent to the framework of Erickson 2017 (which I use as a reference). At steady state (the condition that is relevant for this study), this might be irrelevant, since allocation and partitioning coincide (Scott 2010), but then for clarity, it might be better to present the framework as steady-state relations (as in Scott 2010) and not by ODE.

Related to this point, or this may be the same point, I think the notation is confusing for the parameters v_x, m_x. These are extensive quantities and I am not clear how they are set. For example, v_r ~ R, which is also a necessary condition to get exponential growth (see above). I found this mentioned only on line 642 of the appendix.

Another related point, I did not understand if the model makes more or less implicit flux-balance assumptions (or more in general whether at some point it assumes relationships between fluxes). It should not but at some point, I had this impression. In general, it would be interesting to have some insight into the relationships between the different fluxes (in particular those in consecutive chains) for different values of the resource allocation vector.

Around line 240, the authors discuss that the trend in Figure 3AB is a consequence (through Equation 2) of (population average) density homeostasis (in this case across different strains growing in the same conditions, which is perhaps not the usual way this parameter is considered). Do we then need to think that the model prediction is trivial in this case [as pointed out above, seeing this section of the data and the model prediction would be very instructive here]?

Figure S1 could be presented with Figure 3 (although, see above, probably more is needed). Here one sees the points that do not agree with the model and the authors can comment on those. In particular, those outliers laying near the x-axis of Figure S1B seem potentially interesting to explain/rationalize.

Technical point: How do the predictions depend on the data point used for calibration of the model?

Other points raised after discussion with our group

It seems that the interpretation of the C sector might be different from the canonical one. c → ** Central carbon metabolites **, that is, catabolic products of the carbon source substrate (glucose, glycerol, …) taken up from the medium. What about catabolic enzymes?. Also, enzymes in amino acid metabolism, that are necessary for protein synthesis seem to end up in the R sector (?).

Not clear what the ρ are in dc/dt, and why they must be > 1.

The main result statements of the study are either quite generic or cannot be understood from the main figures. This can probably be improved by reengineering both figures and statements (from abstract):

- very good quantitative agreement between the predicted and observed variability in rates and yields, acetate flow does not correlate with the growth rate.

- resource allocation is a major explanatory factor of the observed variety of growth rates and growth yields across different bacterial strains.

- differences in enzyme activity need to be taken into account to explain variations in protein abundance.

Cmmol seems like a very unintuitive and non-standard unit. Has this been used before? Can a better solution be proposed? Does this hide something related to protein length in the different sectors?

[Editors' note: further revisions were suggested prior to acceptance, as described below.]

Thank you for resubmitting your work entitled "Resource allocation accounts for the large variability of rate-yield phenotypes across bacterial strains" for further consideration by *eLife*. Your revised article has been evaluated by Michael Eisen (Senior Editor) and a Reviewing Editor.

The manuscript has been improved but there are some remaining issues that need to be addressed. In particular:

1. Please address the reviewer's request that some predictions based on the model be added to the text so the reader can better understand how the model works. (i.e. make it less of a black box).

2. To clarify the model and results and how they differ from those of Basan, 2015 please revise the text to address the differences between the results of this study and those of the Basan study in detail. (The reviewer included a list of questions that should ideally be answered in any such comparison in their review below.)

*Reviewer #1 (Recommendations for the authors):*

The authors made considerable revisions and provided a detailed and clear reply to all the points raised. I maintain my opinion on the fact that the work is timely and the theoretical framework is very interesting.

Having said this, I also have to say that the results remain somewhat non-transparent, as the authors were not able to derive a mathematical or qualitative rationale for the main results or analyze the model in terms of simpler one-dimensional relationships, and the comparisons with data remain non-stringent. However, they have provided additional figures and analyses that do contribute towards clarity, as well as clarifying many of the model assumptions and definitions.

I think the manuscript should appear on *eLife*, in view of the contribution towards rationalizing a more complex relationship between growth and yield than the simple trade-off assumed by most. If this could be the central point of this study (I find it interesting and I think it might have some impact), then I have some remarks to clarify the message.

First, the message could emerge more clearly in the abstract.

Second, it might be possible to characterize (without comparing to data, making some simple assumptions on the parameters) the variation of mu_max, y_max (and maybe also some "central values") across conditions, to study and visualize their relationship with resource allocation parameters. Perhaps these predictions are not verified or directly comparable to data (and I am not asking to perform any comparison), but they might help the reader understand how the model works (as I said I think the weakest point of this study remains the "black box" feeling about all the main results).

Third, a more stringent comparison with the data/model of Basan et al. 2015 seems important to clarify the results. What brings those authors to conclude towards a trade-off between protein cost and energy efficiency? Would the model in this study describe the Basan et al. data and how? Would this comparison lead to different conclusions? Are there crucial differences in the modeling choices of the two studies? Are there (according to the authors' model) regimes with/without strong trade-offs and how can they be characterized? These seem like questions worth addressing.

---

## [Author Response]

Essential revisions:While other reviewer concerns as detailed below should be considered, the authors need to address different assumptions for ϕq in order for the study to be complete.

We have developed model variants based on different assumptions for *φ_q_*. Depending on the case considered, the resource allocation parameter for housekeeping proteins is called *χ_q_* or *χ_u_* in the revised manuscript. We have analyzed the models for fixed values of *χ_q_* or *χ_u_*, or for values varying within bounds given by the proteomics data. We show, as explained in detail below, that the predictions of the variability of rate-yield phenotypes are robust with respect to the different assumptions. In addition, we have addressed all other points raised by the reviewers.

Reviewer #1 (Recommendations for the authors):Proposed points of improvement.Rationalize bounds. In Figure 2 I would like to better understand what are the reasons for the bounds. What gives the "P" shape? What are the trends with different allocation trade-offs? etc. Possibly some analytical insight is possible here [maybe with a simplified version of the model], leading to more transparent theoretical insight.

In the revised version of the paper, we have better explained the mapping from resource allocation strategies to rate-yield phenotypes, which defines the bounds in Figure 2, using additional plots. We notably argue that insights into the physiological consequences of a strategy can be gained by means of a pictogram showing (i) the biomass composition over (different categories of) proteins, other macromolecules, and metabolites, (ii) the flux map, and (iii) the energy charge. The pictogram is used, for example, to compare growth of the reference strain used for calibration with growth at maximum rate or maximum yield. This provides a qualitatively understanding of the origin of the trade-off between growth rate and maximum growth yield, which is one of the striking feature of the shape of the cloud of predicted rate-yield phenotypes.

The mapping from resource allocation strategies to rate-yield phenotypes defined by the model is complex, due to the multiple feedback loops between metabolism, protein synthesis, and growth. Finding simplified models that facilitate mathematical analysis and, at the same time, preserve the main features of the mapping is a non-trivial challenge and raises new questions. We therefore think that a full mathematical analysis of the model is beyond the scope of this study.

We have added a new Results section, Predicted rate-yield phenotypes for *Escherichia coli*, to better explain how the rate-yield phenotypes follow from the resource allocation strategies and the macro reactions included in the model. The arguments in this section are supported by the new Figure 2.

Rationalize (and show) trends. Figure 3 seems particularly uninformative. It would be much more instructive to see trends of uptake and secretion rates vs the other variables as 2D plots, compared with the model predictions (particularly for panels AB, panels CD show a complex trend, but this is precisely what I would like to get more insight on). It is not clear how the prediction of Figure 3C is produced by the model since the parameters are not fixed as the allocation changes.

Following the suggestion of the reviewer, we have plotted the glucose uptake and acetate secretion rates against the growth rate and the growth yield in separate 2D plots. The predicted bounds of rate-yield and uptake-secretion phenotypes capture the observed variability very well. This new representation allows a clearer statement of a number of conclusions from the analysis, such as the correlation between glucose uptake rate and growth rate, the absence of correlation between growth rate and acetate secretion rate, and the inverse correlation between growth yield and acetate secretion rate.

We have replaced Figure 3 in the original manuscript by a new figure with the plots suggested by the reviewer (Figure 4 in the revised manuscript). We have accordingly revised the discussion in the section Predicted and observed uptake-secretion phenotypes for *Escherichia coli*. The new Figure 4 makes Supplementary Figures 1 and 2 in the previous version of the manuscript redundant, so these have been removed.

Figure 4. I got lost with this figure, which was particularly uninformative to me (and graphically lacks proper labeling). Looking at the plots, I only see different degrees of agreement between the model and data. Reading the connected Results paragraph, there are a lot of qualitative considerations "under the hood" that seem very interesting but are not accessible/transparent. This could be my own limitation, and it's possible that this paragraph is accessible to a different audience (e.g. more expert than me on metabolic models). However, my impression was that this paragraph/figure could be made more accessible, although I did not gain enough access to give specific recommendations, other than giving the reader some insight on the model predicted trends that we are discussing here.

The main question we want to answer in this section is how *E. coli* can grow both fast and efficiently on glucose. We agree with the reviewer that in the previous version of the manuscript our answer to this question was not clearly formulated and that Figure 4 can be improved. In the revised manuscript, we have streamlined the argument by explaining (i) that we focus on the well-characterized NCM3722 strain as a prototype for high-rate, high-yield growth, (ii) that we need to revise the model assumption of fixed catalytic constants for glycolytic enzymes to quantitatively account for the NCM growth phenotype by means of the observed resource allocation strategy, and (iii) that after doing this, we can attribute high-rate, high-yield growth to the more efficient utilization of proteomic resources. Figure 4 (Figure 5 in the revised manuscript) has been completely revised to better bring out the comparison of the predicted and observed resource allocation strategies and growth phenotypes of NCM3722.

The section Predicted and observed strategies enabling fast and efficient growth of *Escherichia coli* has been rewritten and Figure 4 (Figure 5 in the revised manuscript) has been changed accordingly.

Where are the optima? In Figure 2 one can explore what the model gives if one tries to optimize (1) growth rate at fixed yield (2) yield at fixed growth rate or (3) looks at Pareto optima of both. I agree that optimization of one or both quantities may not be the goal, but still, it is important to understand where optimization would bring theoretically, and how the data points cluster with respect to these theoretical optima.

Figure 2 indeed shows that the global optima of the growth rate and the growth yield are located on both ends of a Pareto front. In the revised version of the manuscript, we discuss the resource allocation strategies underlying the points of the predicted maximum growth rate and maximum growth yield, and the corresponding growth physiology. We also intuitively explain, as mentioned in response to the first comment of this reviewer, which resource allocation trade-off underlies the trade-off between growth rate and growth yield along the Pareto front. Whereas no experimental data points are located in the vicinity of the maximum yield, some strains grow at a rate approaching the predicted maximum.

In the revised manuscript, we discuss the points of maximum rate and yield in the new Results section, Predicted rate-yield phenotypes for *Escherichia coli*, supported by the new Figure 2. In the next section, we explain that no experimental data are available for comparison with the point of maximum yield, but that some strains have a high-rate, high-yield phenotype not far from the point of maximum growth rate. The analysis of high-rate, high-yield is pursued in the section Predicted and observed strategies enabling fast and efficient growth of *Escherichia coli*.

Comparison with other frameworks.A more detailed comparison with other "reference" frameworks would be useful here.I would propose: Erickson 2017, Basan 2015, Maitra 2015 [but other choices are possible][see below]

We have added a new section to *Appendix 1* to compare our model with other existing “reference” models. This section essentially develops the short discussion in the beginning of the section Coarse-grained model with coupled carbon and energy fluxes.

The definition of yield should be explained much more clearly in the main text, both in the model and in the data. Model: Explain why Equation 2 represents the fraction of carbon going into biomass. Data: explain how the quantity is measured and how the measured quantity relates to the model.

The definition of yield was motivated in *Appendix 1* in the discussion leading up to Equation 34 of the revised manuscript. We have repeated this explanation in the main text, as suggested by the reviewer. We have also written a new Methods subsection on the measurement of growth yields and the conversion of measured values to the dimensionless unit adopted in this work. We refer to this new subsection in the *Results section* Predicted and observed rate-yield phenotypes for *Escherichia coli*. The previous version of the manuscript had a subsection in Appendix 1 called Consistency with empirical calculations of rate and yield. This subsection has become redundant after the above modifications and has been removed from the revised manuscript.

I am confused by some sort of implicit identification that the authors make between allocation (e.g. the fraction of ribosome making ribosomes) and partitioning (e.g. the fraction of proteins or total mass that is ribosomes). In particular, for ribosomes, I am not sure that their equations (e.g. Eq (6) in SI regarding ribosomes) are equivalent to the framework of Erickson 2017 (which I use as a reference). At steady state (the condition that is relevant for this study), this might be irrelevant, since allocation and partitioning coincide (Scott 2010), but then for clarity, it might be better to present the framework as steady-state relations (as in Scott 2010) and not by ODE.

Our Equation 6 in Appendix 1 of the original manuscript, *dR/dt* = *φ_r_ V_r_* − *γ R*, corresponds to Equation 3A of Erickson *et al.* [3], *dM_Rb_/dt* = *χ_Rb_*(*t*)*J_R_*, with two differences. First, Erickson *et al.* do not take into account biomass degradation. Second, more important for the question of the reviewer, Erickson *et al.* denote the ribosomal resource allocation parameter by *χ_Rb_* (and allow it to be time-varying), whereas we call the ribosomal resource allocation parameter *φ_r_* (and set it to a constant value defined by resource allocation at steady state). Erickson *et al.* also use symbols *φ*, but these denote proteome fractions rather than resource allocation strategies. In particular, *φ_Rb_*(*t*) is defined as the ribosomal proteome fraction *M_Rb_*(*t*)*/M_P_*(*t*), which at steady state (^∗^) equals the resource allocation parameter: <inline-graphic mimetype="image" mime-subtype="png" xlink:href="media/image1.png" />([3], p. 16 of SI). In our framework, the resource allocation parameter *φ_r_* and proteome fraction *r/p* also coincide at steady state. From the steady state equation for ribosomes, *φ_r_ v_r_* = (*µ* + *γ*)*r*, and the steady-state equation for total proteins, *v_r_* = (*µ* + *γ*)*p*, it follows that *φ_r_* = *r/p*.

In conclusion, our ribosomal resource allocation parameter *φ_r_* has the same symbol as the ribosomal proteome fraction *φ_Rb_* of Erickson *et al.* In order to remove this source of confusion between resource allocation and proteome partitioning, we relabeled our resource allocation strategies to *χ* instead of *φ*, following Erickson *et al.* Even though we study the system at steady state, we prefer to keep the ODE representation of the model. First, the ODE representation is used for finding the steady state. Second, the left-hand side of the ODEs indicates the variable for which the mass or energy balance holds, and thus establishes an explicit correspondence with the graphical representation of the model in Figure 1.

We changed the symbols *φ* to *χ* throughout the text, and made a more explicit distinction between the notions of allocation and partitioning when necessary. The comparison of our model with the model of Erickson *et al.* is carried out in a new subsection of Appendix 1 (Comparison with other coarse-grained resource allocation models).

Related to this point, or this may be the same point, I think the notation is confusing for the parameters v_x, m_x. These are extensive quantities and I am not clear how they are set. For example, v_r ~ R, which is also a necessary condition to get exponential growth (see above). I found this mentioned only on line 642 of the appendix.

*v_x_* and *m_x_* are intensive and not extensive quantities; they have units Cmmol gDW^−1^ h^−1^ and Cmmol gDW^−1^, respectively (*Appendix 1*, after Equations 14 and 22). Their corresponding extensive quantities are denoted as *V_x_* and *M_x_*, with units Cmmol h^−1^ and Cmmol, respectively (Appendix 1, at the point indicated by the reviewer). Accordingly, it does not hold that *v_r_* ∼ *R*, but rather *V_r_* ∼ *R* and *v_r_* ∼ *r*. The model in Figure 1 does not include extensive quantities, only intensive quantities. The extensive quantities are used to construct the model in a principled way from basic assumptions in Appendix 1. The shift from extensive to intensive quantities introduces the growth dilution term in the model. It also allows the definition of the growth rate and growth yield in terms of reaction rates.

As an aside, and as a follow-up of the previous point, note that the model of Erickson *et al.* consists of extensive quantities, contrary to our model. The rate equations used in our model, expressing the dependency of a reaction rate on ATP and metabolite concentrations with the help of a half-saturation constant, require the use of intensive variables.

We have explicitly stated in the section Coarse-grained model with coupled carbon and energy fluxes that the model consists of intensive variables.

Another related point, I did not understand if the model makes more or less implicit flux-balance assumptions (or more in general whether at some point it assumes relationships between fluxes). It should not but at some point, I had this impression. In general, it would be interesting to have some insight into the relationships between the different fluxes (in particular those in consecutive chains) for different values of the resource allocation vector.

At steady state, the different fluxes in the model including growth dilution must be balanced, in the sense that the right-hand side of the equations in Figure 1 must equal 0. We do not make any implicit or explicit assumptions on relations between fluxes though: every flux is defined by a separate rate equation, given by Equations 35-40 in Appendix I.

In order to get a better insight into the relations between the fluxes, we have developed a visual representation of the fluxes at steady state for a given resource allocation strategy (Figure 2 in the revised manuscript, see also the first comment of the reviewer). With the help of this pictogram, we explain how the incoming carbon flux is distributed over the other fluxes (protein synthesis, respiration, fermentation, *…*).

We added a new Results section, Predicted rate-yield phenotypes for *Escherichia coli*, to better explain the working of the model, including the flux balance at steady state.

Around line 240, the authors discuss that the trend in Figure 3AB is a consequence (through Equation 2) of (population average) density homeostasis (in this case across different strains growing in the same conditions, which is perhaps not the usual way this parameter is considered). Do we then need to think that the model prediction is trivial in this case [as pointed out above, seeing this section of the data and the model prediction would be very instructive here]?

The expression *Y* = *µ/*(*β v_mc_*) (Equation 2) relates the growth yield to the growth rate and the glucose uptake rate, as explained in *Appendix 1* in the discussion leading up to Equation 34. The point we wanted to make is that this relation between the three quantities holds by definition, and that we also expect it to hold for experimental measurements of *Y*, *µ*, and *v_mc_*. As a consequence, for similar values of 1*/β* in the model and in experiments, a given pair of growth rate and growth yield returns a similar value of *v_mc_*.

In the new discussion of the comparison of the predicted and observed variability of uptake secretion phenotypes, structured around the new Figure 4, the relation of Equation 2 is used somewhat differently. We explain that, given a glucose uptake rate, the bacteria can grow at different growth rates depending on the growth yield, which is a consequence of the resource allocation strategy adopted by the cell (see also a comment of Reviewer 2). The determination of growth rate and growth yield by resource allocation is not trivial though.

This point has been reformulated in the revised version of the manuscript (Predicted and observed uptake-secretion phenotypes for *Escherichia coli*).

Figure S1 could be presented with Figure 3 (although, see above, probably more is needed). Here one sees the points that do not agree with the model and the authors can comment on those. In particular, those outliers laying near the x-axis of Figure S1B seem potentially interesting to explain/rationalize.

In the revised manuscript, following a suggestion of this reviewer, we have structured the discussion of the measured and predicted uptake and secretion rates in a different way. In particular, we compare the predicted and observed variability of uptake and secretion rates as related to the growth rate and growth yield by a series of 2D plots (Figure 4 in the revised manuscript). Some outliers occur in these new projections of the model predictions and are discussed in the text.

Supplementary Figure 1 has been removed from the revised manuscript. We discuss the outliers in the section Predicted and observed uptake-secretion phenotypes for *Escherichia coli*.

Technical point: How do the predictions depend on the data point used for calibration of the model?

We also calibrated the model for another commonly-used *E. coli* laboratory strain, MG1655 instead of BW25113. The clouds of predicted rate-yield phenotypes for the BW and MG strains are quantitatively very similar. This shows the robustness of the rate-yield predictions for calibration with an alternative dataset.

We discuss the results of the calibration of the model for another *E. coli* strain in the *Discussion section* and in Figure 3—figure supplement 1. The details of the calibration are included in a new subsection of Appendix 2, Data and parameter estimates for an alternative *E. coli* strain.

Other points raised after discussion with our groupIt seems that the interpretation of the C sector might be different from the canonical one. c → ** Central carbon metabolites **, that is, catabolic products of the carbon source substrate (glucose, glycerol, …) taken up from the medium. What about catabolic enzymes? Also, enzymes in amino acid metabolism, that are necessary for protein synthesis seem to end up in the R sector (?).

Indeed, central carbon metabolites (variable *c* in the model) are defined as consisting of the catabolic products of the carbon source (glucose, glycerol, …) taken up from the medium. They include intermediates of the glycolysis pathway, the tricarboxylic acid cycle, and the pentose phosphate pathway, notably the thirteen precursor metabolites from which the building blocks for macromolecules (amino acids, nucleotides, …) are produced ([11], Ch. 5). The catabolic enzymes are included in the protein category “Enzymes in central carbon metabolism”, which take up substrates from the environment and break them down to central carbon metabolites (variable *m_c_* in the model).

In other models, the precursor pool is often defined as consisting of amino acids, *e.g.*, the models of Erickson *et al.* [3] and Giordano *et al.* [6]. Here we needed a different definition, because our model includes macro reactions for the production of other macromolecules (RNA, DNA, …) and the secretion of acetate. The synthesis of other macromolecules and the secretion of acetate can also be traced back to the precursor metabolites mentioned above, which has motivated our definition of *c* as the pool of central carbon metabolites. As an aside, this conceptualization corresponds to the core model of *E. coli* metabolism in flux balance analysis, where the biomass function is defined in terms of a dozen precursor metabolites [4].

Following the above logic, the enzymes in amino acid metabolism are included with the ribosomes in the category “Ribosomes and translation-affiliated proteins, including enzymes in amino acid metabolism” (variable *r* in the model, Appendix 1). Enzymes in amino acid metabolism convert central metabolites into amino acids, which ribosomes assemble into proteins. Another reason for including the enzymes in amino acid metabolism with ribosomal proteins is that their proteome fractions have the same linear dependence on the growth rate, contrary to the enzymes in central carbon metabolism (Figure S11 in [12]). In previous stages of this work, we developed model versions including an amino acid pool as a separate variable, but given that this complicated the model without substantially changing its predictions, we have preferred the more parsimonious model presented here.

We briefly summarize the arguments above in the section Coarse grained model with coupled carbon and energy fluxes and in Appendix 1.

Not clear what the ρ are in dc/dt, and why they must be > 1.

The *ρ* factors account for the (additional) loss of carbon during the synthesis of macromolecules and the secretion of acetate. For example, when converting central metabolites into proteins during growth on glucose, for every 1 Cmmol of protein produced, 0.17 Cmmol of CO_2_ is generated ([1] and Appendix 2). Therefore, in order to preserve the carbon balance, when writing the rate of consumption of central carbon metabolites during protein synthesis, we need to multiply *v_r_* with the factor 1.17. This factor, accounting for the loss of CO_2_, is expressed as the constant *ρ_ru_* in the model. Because it expresses an additional consumption of carbon, beyond that included in the proteins, *ρ_ru_ >* 1 by definition. A similar explanation can be given for the loss of CO_2_ associated with the synthesis of acetate, giving rise to the term *ρ_mef_ v_mef_*.

We have explained the origin of the *ρ* factors in more detail in the main text (in the section Coarse-grained model with coupled carbon and energy fluxes) and in Appendix 1 (in the section Derivation of model equations).

The main result statements of the study are either quite generic or cannot be understood from the main figures. This can probably be improved by reengineering both figures and statements (from abstract):– very good quantitative agreement between the predicted and observed variability in rates and yields, acetate flow does not correlate with the growth rate.– resource allocation is a major explanatory factor of the observed variety of growth rates and growth yields across different bacterial strains.– differences in enzyme activity need to be taken into account to explain variations in protein abundance.

We have verified that the statements of the main results in the abstract are explicitly matched by statements in the text, in the context of the discussion of Figures 3-5 and in the Discussion section.

Cmmol seems like a very unintuitive and non-standard unit. Has this been used before? Can a better solution be proposed? Does this hide something related to protein length in the different sectors?

Cmmol is a unit often used in biotechnology (*e.g.*, [8]) and in ecology (*e.g.*, [10]). It is notably used for expressing yields: Cmmol_biomass_/Cmmol_glucose_ indicates the fraction of carbon taken up by the cells that is included in the biomass [8]. The advantage of adopting this unit in our coarse-grained model is that it enables a rigorous statement of the carbon balance, by making explicit the carbon contents of the cellular components and the fluxes. This is critical for estimating the variations in growth rate and growth yield when changing the resource allocation strategy.

We have provided an explicit motivation for the use of Cmmol units in the section Coarse-grained model with coupled carbon and energy fluxes.

[Editors' note: further revisions were suggested prior to acceptance, as described below.]

The manuscript has been improved but there are some remaining issues that need to be addressed. In particular:1. Please address the reviewer's request that some predictions based on the model be added to the text so the reader can better understand how the model works. (i.e. make it less of a black box).2. To clarify the model and results and how they differ from those of Basan, 2015 please revise the text to address the differences between the results of this study and those of the Basan study in detail. (The reviewer included a list of questions that should ideally be answered in any such comparison in their review below.)

In order to address the first point, we have added a new paragraph in the section Predicted rate-yield phenotypes for *Escherichia coli*. The text is based on the analysis of a simplified version of the model described in a new subsection of Appendix 1, Simplified coarse-grained resource allocation models. This simplified model allows to trace back the decrease in growth yield and increase in growth rate occurring along the Pareto frontier in Figure 2 to underlying changes in the resource allocation strategy. This opens up the “black box” of the model, as Reviewer 1 suggests, for a specifically striking prediction of our model. This extension comes with an additional supplementary figure, Figure 2—figure supplement 4.

The second point is addressed in a new paragraph in the *Discussion section*, which answers the questions posed by the reviewer. This paragraph is based on the new subsection Simplified coarse-grained resource allocation models in Appendix 1, where it is shown that the model of this manuscript can be reduced to the model of Basan *et al.* [1] when a number of additional assumptions are made. In particular, the possibility of a trade-off between investment in proteins and metabolites, which our model admits contrary to the model of Basan *et al.*, is shown to be critical for the differences in predictions of the models.

Reviewer #1 (Recommendations for the authors):The authors made considerable revisions and provided a detailed and clear reply to all the points raised. I maintain my opinion on the fact that the work is timely and the theoretical framework is very interesting.Having said this, I also have to say that the results remain somewhat non-transparent, as the authors were not able to derive a mathematical or qualitative rationale for the main results or analyze the model in terms of simpler one-dimensional relationships, and the comparisons with data remain non-stringent. However, they have provided additional figures and analyses that do contribute towards clarity, as well as clarifying many of the model assumptions and definitions.I think the manuscript should appear on eLife, in view of the contribution towards rationalizing a more complex relationship between growth and yield than the simple trade-off assumed by most. If this could be the central point of this study (I find it interesting and I think it might have some impact), then I have some remarks to clarify the message.First, the message could emerge more clearly in the abstract.

We have reformulated the second part of the abstract, and added a phrase at the end of the *Introduction*, to better bring out this message.

Second, it might be possible to characterize (without comparing to data, making some simple assumptions on the parameters) the variation of mu_max, y_max (and maybe also some "central values") across conditions, to study and visualize their relationship with resource allocation parameters. Perhaps these predictions are not verified or directly comparable to data (and I am not asking to perform any comparison), but they might help the reader understand how the model works (as I said I think the weakest point of this study remains the "black box" feeling about all the main results).

We have found a way to mathematically analyze the model along the predicted Pareto frontier running from *Y_max_* to *µ_max_* in the rate-yield phenotype space. This has required making a number of simplifications that are justified in this specific region. The analysis of the simplified model allows the decrease in growth yield with the increase in growth rate along the Pareto frontier to be traced back to qualitative changes in the underlying resource allocation strategy, by taking into account the constraints on carbon and energy flows, biomass composition, and resource allocation. In particular, this analysis supports the observation made in the main text that the rate-yield trade-off corresponds to a trade-off between investment in proteins *vs* metabolites on the physiological level. The analysis thus opens the “black box” for this particularly striking prediction of a trade-off between growth rate and (maximum) growth yield.

We have added a new paragraph in the section Predicted rate-yield phenotypes for *Escherichia coli*. The text is based on the analysis of a simplified version of the model described in a new subsection in Appendix 1, Simplified coarse-grained resource allocation models, which is accompanied by an additional supplementary figure (Figure 2—figure supplement 4).

Third, a more stringent comparison with the data/model of Basan et al. 2015 seems important to clarify the results. What brings those authors to conclude towards a trade-off between protein cost and energy efficiency? Would the model in this study describe the Basan et al. data and how? Would this comparison lead to different conclusions? Are there crucial differences in the modeling choices of the two studies? Are there (according to the authors' model) regimes with/without strong trade-offs and how can they be characterized? These seem like questions worth addressing.

We have shown how the model of Basan *et al.* [1] can be derived from our model when making a number of simplifying assumptions. That is, under these additional assumptions, our model and the model of Basan *et al.* make the same predictions. The major simplifying assumption is that the concentrations of central carbon metabolites, energy cofactors, and other macromolecules are constant and that their contribution to the mass balance can be ignored. As a consequence, a trade-off between investment in proteins and metabolites is no longer possible. This notably rules out the strategy of more efficiently utilizing available proteomic resources, which underlies high-rate, high-yield growth predicted by our model and observed in the data.

The comparison with the model of Basan *et al.* is addressed in a new paragraph in the Discussion section, which explicitly answers the questions posed by the reviewer. This paragraph is based on a new subsection in Appendix 1, Simplified coarse-grained resource allocation models, which shows under which conditions the model of this manuscript reduces to the model of Basan *et al.*

References

[1] M. Basan, S. Hui, H. Okano, Z. Zhang, Y. Shen, J.R. Williamson, and T. Hwa. Overflow metabolism in *Escherichia coli* results from efficient proteome allocation. *Nature*, 528(7580):99– 104, 2015.